_Article_

EMBO _reports_

# RNA polymerase II promotes the organization of chromatin following DNA replication

Susanne Bandau [ID][1], Vanesa Alvarez [ID][1], Hao Jiang[2], Sarah Graff[3], Ramasubramanian Sundaramoorthy[4], Marek Gierlinski [ID][5], Matt Toman [ID][4], Tom Owen-Hughes [ID][4], Simone Sidoli[3], Angus Lamond [ID][2] & Constance Alabert [ID][1][✉]

## Abstract

**Understanding how chromatin organisation is duplicated on the two daughter strands is a central question in epigenetics. In mammals, following the passage of the replisome, nucleosomes lose their defined positioning and transcription contributes to their re-organisation. However, whether transcription plays a greater role in the organization of chromatin following DNA replication remains unclear. Here we analysed protein re-association with newly replicated DNA upon inhibition of transcription using iPOND coupled to quantitative mass spectrometry. We show that nucleosome assembly and the re-establishment of most histone modifications are uncoupled from transcription. However, RNAPII acts to promote the re-association of hundreds of proteins with newly replicated chromatin via pathways that are not observed in steady-state chromatin. These include ATP-dependent remodellers, transcription factors and histone methyltransferases. We also identify a set of DNA repair factors that may handle transcription-replication conflicts during normal transcription in human non-transformed cells. Our study reveals that transcription plays a greater role in the organization of chromatin post-replication than previously anticipated.**

**Keywords** DNA Replication; Transcription; ATP-dependent Chromatin Remodellers; Transcription Factor; DNA Repair
**Subject Categories** Chromatin, Transcription & Genomics; DNA Replication, Recombination & Repair

## Introduction

Chromatin structures contain multiple layers of information that sustain transcription programs. These structures must be plastic enough to permit the establishment of distinct transcription programs in distinct cells during development and robust enough to ensure the maintenance of established programs in proliferating cells. In cycling cells chromatin is profoundly modified twice. In mitosis, chromatin undergoes a rapid cycle of compaction and decompaction, allowing the distribution of the copied genetic material between the two daughter cells. In S phase, at each site of DNA synthesis, chromatin undergoes a cycle of disruption ahead of the replisome and of reassembly on the two daughter strands. Despite major recent advances in our understanding of this dynamic process, the mechanisms that ensure the reassembly of chromatin structures on newly replicated DNA and their coordination with other DNA-based processes remain unclear. Understanding how chromatin-based information propagates from cell to cell, and how this information can be altered in a multitude of disease contexts such as cancers, is a central question in biology.

Following DNA replication, nucleosomes are first assembled with less well-defined positioning and transcription factor occupancy at gene regulatory sequences is reduced (Owens et al, 2019; Ramachandran and Henikoff, 2016; Stewart-Morgan et al, 2019). RNA polymerase II (RNAPII) is recruited to replicated DNA (Bruno et al, 2023; Fenstermaker et al, 2023; Stewart-Morgan et al, 2019), and although there is no direct measure of de novo transcriptional activity on newly replicated chromatin, it is believed that similar to mitosis (Palozola et al, 2017), transcription is transiently reduced behind replisomes. Indeed, the initiation competent form of RNAPII, RNAPII phosphorylated on serine 5 (RNAPII-pS5), progressively accumulates and within 2 h, pre-replicated RNAPII-pS5 levels and DNA accessibility are re-established (Stewart-Morgan et al, 2019). This restoration is not homogenous, as regions such as super-enhancers and CTCF binding sites are restored prior to promoters (Owens et al, 2019; Stewart-Morgan et al, 2019). Mechanistically, recent work has shown that in mouse ES cells, transcription is driving nucleosome repositioning within gene regulatory regions (Stewart-Morgan et al, 2019). Although several chromatin remodelling enzymes, that have the ability to reposition or evict nucleosomes, are present on newly replicated DNA (Alvarez et al, 2023; Ramachandran and Henikoff, 2016), how transcription may promote nucleosome positioning following the passage of the fork remains unclear.

Beyond nucleosome positioning, transcription has been suggested to promote other processes taking place on newly replicated

[1]MCDB, School of Life Sciences, University of Dundee, DD15EH Dundee, UK. [2]Laboratory of Quantitative Proteomics, MCDB, School of Life Sciences, University of Dundee, DD15EH Dundee, UK. [3]Department of Biochemistry at the Albert Einstein College of Medicine, New York, NY, USA. [4]Laboratory of Chromatin Structure and Function, MCDB, School of Life Sciences, University of Dundee, DD15EH Dundee, UK. [5]Data Analysis Group, Division of Computational Biology, School of Life Sciences, University of Dundee, Dow Street, DD1 5EH Dundee, UK. ✉E-mail: calabert@dundee.ac.uk

DNA such as the restoration of histone modifications. Following DNA replication, many histone methylations are diluted by the addition of newly synthesised histones that in mammals are largely unmethylated (Alabert et al, 2015; Reveron-Gomez et al, 2018). Several mechanisms have been proposed to overcome this dilution, such as the direct recruitment of H3K9me3 writers to the replisome (Padeken et al, 2022). H3K4me3 restoration following DNA replication has been suggested to rely on transcription (Reveron-Gomez et al, 2018; Serra-Cardona et al, 2022). A probable other candidate is H3K36me3, as its writer NSD2 directly bind RNAPII-pS2-S5, and this interaction promotes H3K36me3 deposition on chromatin (Neri et al, 2017; Venkatesh et al, 2012). More recently, H2BK120ub1 has been shown to be coupled with transcription (Flury et al, 2023). Yet, it remains unknown whether transcription is required for the restoration of other histone modifications. More generally, it is uncertain whether the mechanisms that maintain proteins and histone modifications in steady-state chromatin are applicable on newly replicated chromatin. These questions have gained a novel interest following the development of inhibitors and rapid degradation-based approaches. These studies revealed that continuous action of chromatin remodelling enzymes is required to maintain chromatin accessibility at enhancers (Blumli et al, 2021; Iurlaro et al, 2021; Schick et al, 2021). Similarly, the continuous presence of PRC1 at Polycomb target genes is required to maintain silencing (Dobrinic et al, 2021). Therefore, the finding that chromatin states are much more dynamic than anticipated further highlights the need to better understand the immediate response to chromatin disruption including following DNA replication.

Here, we examined the role play by transcription in chromatin organisation on newly replicated DNA. To do so we use inhibitors to block RNAPII binding and elongation and analysed protein-binding kinetics and the establishment of histone modifications behind replisomes using isolation of Proteins On Newly replicated DNA (iPOND) coupled to mass spectrometry (Sirbu et al, 2012). We show that the binding of RNAPII promotes the recruitment of ATP-dependent chromatin remodellers and of several transcription factors (TFs) to newly replicated DNA and this dependency is not observed in steady-state chromatin. Moreover, transcription promotes the restoration of H3.3K36me2, while simultaneously provoking the recruitment of several DNA repair factors. Altogether, this study revealed that transcription promotes chromatin re-establishment post-replication but is also a source of genotoxic stress.

## Results

### Isolation of newly replicated chromatin under conditions where transcription is inhibited

We aimed to investigate the role of transcription in the organisation of chromatin following DNA replication. To do so the composition of newly replicated chromatin was examined in the presence or absence of transcription (Fig. 1A). Non-transformed foetal human lung fibroblasts (TIG-3) cells were treated with transcription inhibitors and newly replicated chromatin was isolated by iPOND and analysed by TMT mass spectrometry (Fig. 1B). Two small-molecule inhibitors targeting two different transcription steps were selected. Triptolide (TPL) inhibits the

ATPase activity of XPB, a subunit of TFIIH, which is part of the pre-initiation complex. This drug induces the rapid proteasomal degradation of RBP1 and therefore blocks RNAPII loading (Vispe et al, 2009). 5,6-dichlorobenzimidazone-1-β-D-ribofuranoside (DRB) inhibits the kinase activity of CDK9, a component of the transcription elongation factor pTEFb, thereby initially increasing RNAPII pausing followed by a reduction of the fraction of chromatin-bound RNAPII (Steurer et al, 2018). To establish the drug treatment conditions required to impair transcription in TIG-3 cells, EU levels were quantified by Quantitative Image-Based Cytometry (QIBC) (Toledo et al, 2013). Within 2 h, 1 μM of TPL and 50 μM of DRB significantly reduced transcription rates (Fig. 1C). TPL treatment also reduced the level of chromatin-bound RNAPII (Fig. EV1A), which is consistent with the previously described rapid induction of RNAPII proteasomal degradation (Vispe et al, 2009). Importantly, in these conditions of transcription inhibition, DNA synthesis was generally unaffected (Fig. 1D), making it possible to EdU label and examine nascent chromatin.

Newly replicated DNA was labelled using an 11-minute pulse of EdU, which is incorporated genome-wide as the cells are growing asynchronously (Reveron-Gomez et al, 2018). At the end of the EdU pulse, cells were either processed according to the iPOND protocol (Nascent sample) or EdU was chased using thymidine for an additional 1 h (+1 h sample) or 2 h (+2 h sample) (Fig. EV1B,C). The thymidine chase stops EdU incorporation, enabling to examine the composition of EdU-labelled chromatin as it matures (Sirbu et al, 2011). Importantly, the drug treatments did not affect the chase efficiency (Fig. EV1B,C), nor the distribution of cells within S phase (Fig. EV1D). Moreover, neither TPL nor DRB induced detectable DNA damage in the cells within the time frame of the experiments (Fig. EV1E). Proteins associating with EdU-labelled DNA were recovered in four independent biological iPOND replicates, resulting in the identification of 3156 proteins (Fig. EV1F; Datasets EV1–3). The relative intensities between time points and treatments were calculated for each protein identified and expressed as log2-fold changes relative to the mean intensity across all samples (see 'Methods' for data processing). Core replisome components were enriched at the nascent time point compared to +1 and 2 h time points in DMSO, TPL and DRB, indicating that newly replicated chromatin had been successfully isolated in all three conditions (Figs. 1E and EV1G).

In DMSO, RNAPII abundance only moderately increased between 11 and 120 min after the passage of replisomes (Fig. 1E). To further assess the status of RNAPII on replicated DNA, we repeated the experiment and probed for RNAPII serine 5 phosphorylation (RNAPII-pS5), a phosphorylation deposited on RNAPII upon its activation. RNAPII-pS5 levels were low in the 11-min sample (nascent) compared to 2 h after the passage of the fork (Fig. EV1H). Together with the mass spectrometry results, these data confirm that behind replisomes RNAPII binds within the first 11 min and is activated within the following hour (Stewart-Morgan et al, 2019). Upon TPL treatment, the abundance of RNAPII components were significantly reduced compared to DMSO, indicating that in these conditions RNAPII binding is impaired (Fig. 1E,F). Upon DRB treatment, although the level of phosphorylated RNAPII decreases on nascent chromatin (Fenstermaker et al, 2023), RNAPII abundance is only moderately affected (Fig. 1E,G). The latter is consistent with in vitro FRAP experiments (Steurer

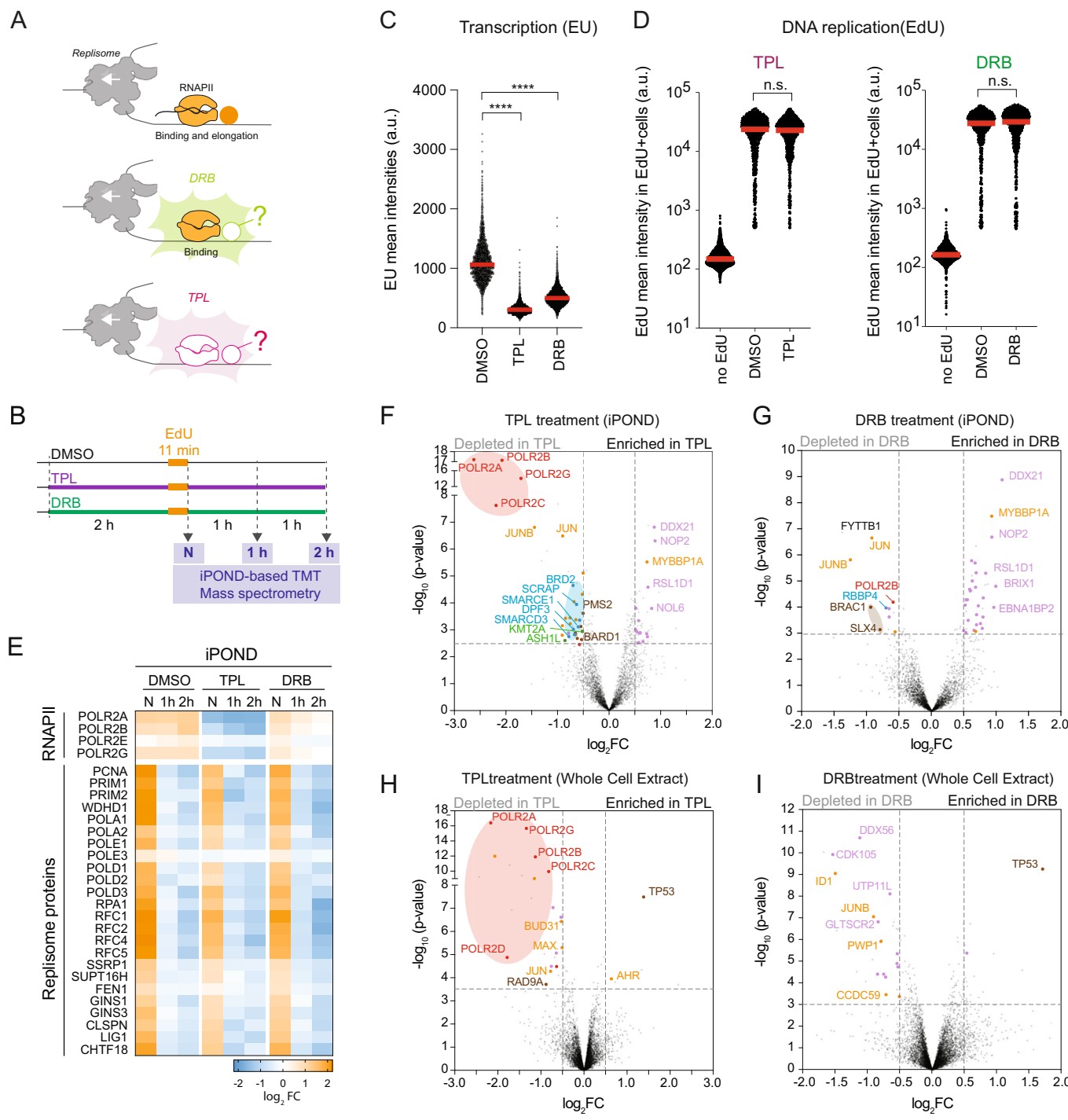

**Figure 1. Proteomic profiling of chromatin behind replisomes upon transcription inhibition.**

(A) Behind replisomes, histones and chromatin components are reassembled onto newly replicated DNA. How transcription contributes to this process is unclear. (B) Experimental design of the iPOND-TMT time course experiment in TIG-3 cells in the presence of DMSO, TPL or DRB. (C, D) QIBC-based analysis of EU level (C) and EdU level (D) in DMSO, TPL and DRB treated cells. Graphs show the mean intensity per nuclei, >539 nuclei were analysed per sample. Red line: median; unpaired Mann–Whitney *t* test; ****$P$ value < 0.0001; n.s. non-significant. $N = 3$, one biological replicate is shown. (E) Heatmap of replisome components and RNAPII subunits generated by using the log2-fold change of batch-corrected abundance with respect to the protein mean ($n = 4$ biological replicates). Each column represents a time point (N: Nascent, 1 h, and 2 h) and each row corresponds to the protein indicated on the left. Colour scale is indicated. (F) Volcano plot generated using the iPOND-TMT-MS data showing protein fold changes based on the full model for TPL treatment. Limma *t* test, significantly changing proteins with a magnitude fold change ≥0.5 (*x* axis) and an FDR ≤ 0.05 (*y* axis) are highlighted. RNAPII, red; DNA repair, brown; Chromatin remodellers, blue; Transcription regulators, orange; Histone modifiers, green. The dashed horizontal line shows the $P$ value cut-off, and the two vertical dashed lines indicate proteins enriched or depleted upon TPL treatment. (G). Same as in (F) for DRB treatment. (H, I) Same as in (F) using the whole-cell extract from the iPOND-TMT time course experiment showing the protein fold changes based on the full model for TPL treatment (H) or DRB treatment (I). Source data are available online for this figure.

et al, 2018). Of note, RNAPI and RNAPIII levels were affected by TPL and DRB, but the effects were dampened compared to RNAPII (Fig. EV1I). The association of RNAPII following these treatments was confirmed using an orthogonal approach, by Proximity Ligation Assay (PLA) (Fig. EV1J). Therefore, newly replicated chromatin has been successfully isolated in conditions where RNAPII binding and elongation was permitted (DMSO), RNAPII binding had been allowed but transcription elongation prevented (DRB), or RNAPII binding was prevented (TPL).

## RNAPII's binding contributes to protein binding to newly replicated DNA

In addition to RNAPII, the binding of 220/3156 proteins (~7%) were significantly affected by inhibition of transcription (158 in TPL only, 24 in DRB only, and 38 in DRB and TPL) (Figs. 1F,G and EV2A) (see 'Methods' for significance determination). Upon TPL treatment, newly replicated chromatin was depleted in chromatin remodellers, transcription factors, histone modifiers, and DNA repair proteins (Figs. 1F and EV2B,C). Upon DRB treatment, a distinct set of proteins were depleted, including TFs and proteins involved in DNA repair (Figs. 1G and EV2B,D). On the other hand, the association of histone chaperones, canonical histones and histone variants were generally not affected by TPL or DRB treatments (Fig. EV3A,B). Taken together, these findings revealed that transcription promotes the binding of hundreds of proteins to newly replicated DNA. On the other hand, transcription does not interfere with the replication-coupled assembly of nucleosomes.

Only 11 of the 220 proteins affected by DRB or TPL are known RNAPII interactors (Ebmeier et al, 2017) (Fig. EV3C). This indicates (i) that our data capture direct and indirect dependencies on RNAPII and (ii) that most of the RNAPII interactors bind to nascent chromatin independently of RNAPII. Next, as RNA production was purposely blocked using transcription inhibitors, we examined whether the changes detected on replicated chromatin could be indirect, simply mirroring changes in the abundance of proteins in the cells. To do so, we measured the abundance of proteins in whole-cell extracts by TMT quantitative mass spectrometry (Datases EV4 and 5). As expected from previous studies (Vispe et al, 2009), the cellular abundance of several proteins was significantly reduced upon TPL (52 proteins) and DRB (150 proteins) treatments (Figs. 1H,I and EV3D). These include RNAPII itself and additional proteins known to be degraded in TPL-treated cells (Vispe et al, 2009) (Fig. EV3E). Yet only 16 of these proteins were also depleted on replicated chromatin (Fig. EV3D). Moreover, only 21 proteins out of 220 depleted on nascent chromatin belonged to the lowest abundant proteins (using iBAQ as a proxy) and 6 belonged to the shortest half-life proteins (Li et al, 2021) (Fig. EV3C,F–H). Altogether these results reveal that the inhibition of transcription affects the binding of hundreds of proteins to newly replicated chromatin, and for most of them (87%) this is not an indirect consequence of changes to protein abundance.

## Transcription promotes H3.3K36me2 restoration following the passage of the fork

From the dozens of histone methyltransferases and acetyltransferases identified, only two were significantly depleted from newly replicated DNA upon TPL treatment, KMT2A and ASH1L (Figs. 1F,

2A and EV4A). Both catalyse histone modifications linked to transcription, H3K4me and H3K36me, respectively. It is therefore possible that following the passage of the replisome, the methylation of newly synthesised histones (that do not carry these two methylations) relies on transcription itself (Fig. 2B). To test this possibility, we examined histone modifications on newly replicated chromatin by iPOND label-free mass spectrometry (Sidoli et al, 2016) (Fig. 2C). The level of each histone modification identified on a residue is expressed as a percentage of all the possible modified states detected for this residue as in Alabert et al 2015 (Dataset EV6).

H4K5 and K12 acetylation levels decrease within two hours, consistent with the removal of this modification from newly assembled histones within 30 mins (Benson et al, 2006) (Fig. 2D). Interestingly, neither TPL nor DRB treatments affected the rate of histone H4K5K12 deacetylation, indicating that transcription and the rapid deacetylation of histone H4K5K12 are independent. Intriguingly, from the dozens of lysine acetylations monitored in this analysis, only H4K16ac showed a response to transcription inhibition, a small but significant increase upon DRB treatment (Fig. 2D).

Regarding histone methylations, within 2 h following the passage of the fork, most modifications do not change dramatically (Figs. 2E and EV4B,C, DMSO) (Alabert et al, 2015; Reveron-Gomez et al, 2018). The exception is H3.3K36me2 which increases substantially over two hours (Fig. 2E). This increase in inhibited by DRB and TPL treatments. Furthermore, association of ASH1L, a methyltransferase that generates this modification, was also significantly impaired by TPL treatment (Fig. 2A). Taken together, these results reveal that the restoration of histone modifications following the passage of the fork are generally uncoupled from transcription. The exceptions were two transcription-associated modifications, H3K36me2 and H4K16ac, potentially due to the impaired recruitment of ASH1L for H3K36me2, and a mechanism yet to be elucidated for H4K16ac.

## Blocking transcription reduced the recruitment of DNA repair proteins on nascent chromatin

The abundance of several DNA repair factors was significantly reduced on replicated chromatin upon TPL and DRB treatments (Fig. 3A). This reduction was validated for one of them, BARD1, using PLA (Reduction in TPL, non-significant; DRB, significant) (Fig. EV5A). To estimate the effect of each drug treatment on each repair pathway, the effect size was calculated as the mean log2-fold change for the treatment, and the statistical significance was estimated by gene set enrichment using a bootstrap (see 'Methods'). All DNA repair pathways were significantly depleted from replicated chromatin upon TPL treatment (Fig. 3B). In addition, Mismatch Repair (MMR), Homologous Recombination (HR), Fork Quality control (FQC), and the Fanconi Anaemia (FA) pathways were affected upon DRB treatment. This result suggests that endogenous levels of transcriptional elongation provoke the recruitment of multiple DNA repair factors on replicated chromatin. This is not surprising as transcription, in particular head-on collision between the replisome and RNAPII, is a major source of genomic instability (Berti et al, 2020; Hamperl and Cimprich, 2016; Stoy et al, 2023), and our data identify a set of DNA repair factors that are likely to contend with transcription-

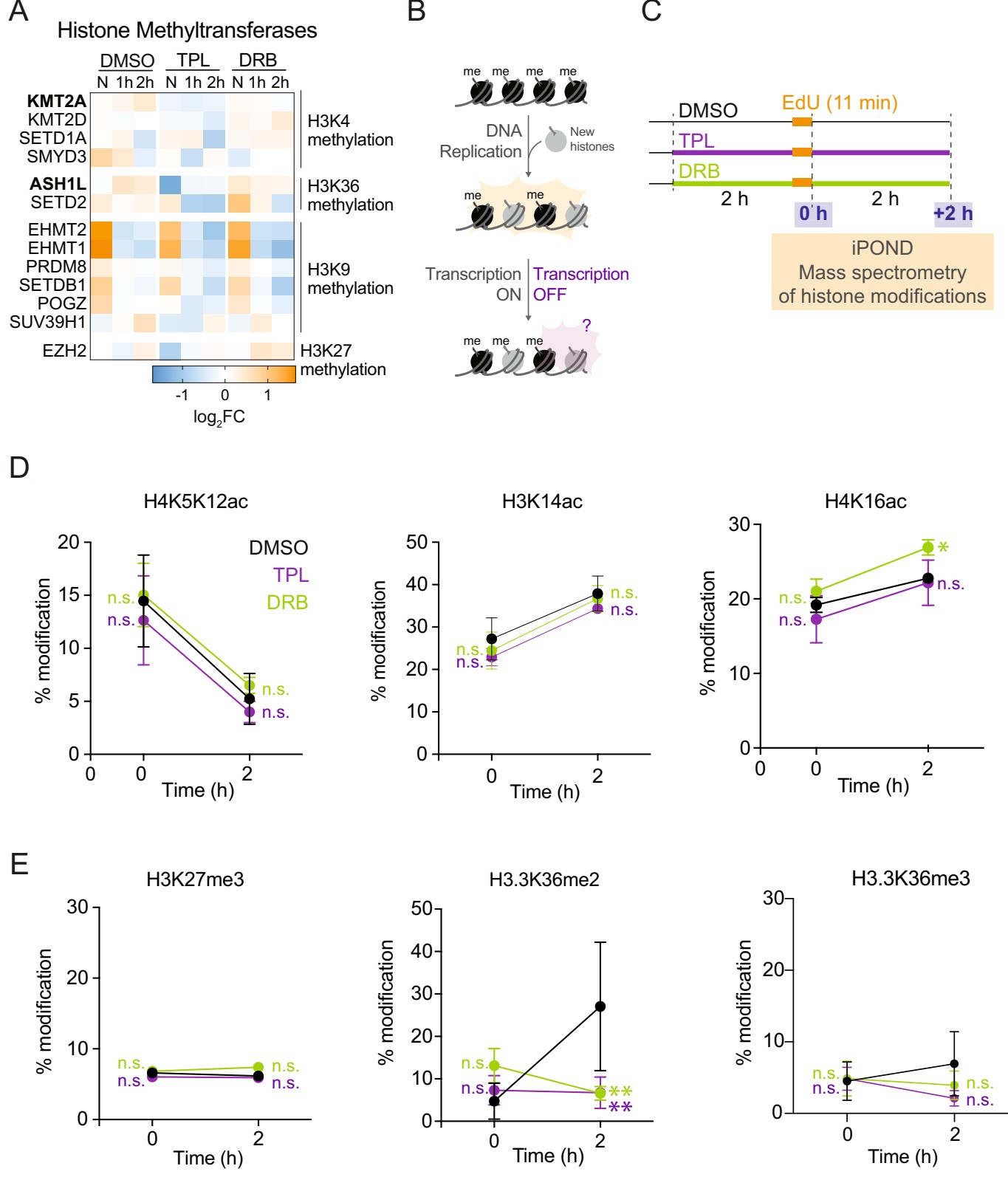

**A** Histone Methyltransferases

**B** DNA Replication / New histones / Transcription ON / Transcription OFF

**C** DMSO, TPL, DRB / EdU (11 min) / iPOND Mass spectrometry of histone modifications

**D** H4K5K12ac / H3K14ac / H4K16ac

**E** H3K27me3 / H3.3K36me2 / H3.3K36me3

Figure 2.  Transcription promotes H3.3K36me2 re-establishment on replicated chromatin.

(A) Heatmap of Histone Methyltransferases generated using the log2-fold change of batch-corrected abundance with respect to the protein mean ($n = 4$ biological replicates). Each column represents a time point (N: Nascent, 1 h, and 2 h) and each row corresponds to the protein indicated on the left. Colour scale is indicated below. (B) Behind the replication fork, histone methylation levels are diluted and must be fully restored on newly replicated chromatin. To which extent transcription contributes to the re-establishment of histone modifications following DNA replication is unknown. (C) Experimental design of the iPOND-TMT time course experiment in TIG-3 cells upon DMSO, TPL or DRB treatment. (D) Proportion of H4K5K12ac (left), H3K14ac (middle), and H4K16ac (left) on newly replicated chromatin for each time point in DMSO (black), DRB (green) and TPL (purple) treated cells. The standard error of the mean is shown. $N = 3$ biological replicates. Paired $t$ test, **$P$ value < 0.01; *$P$ value < 0.05; n.s., non-significant. (E) Same as in (D) for H3K27me3 (left), H3.3K36me2 (middle) and H3.3K36me3 (right). $N = 3$ biological replicates.

replication conflicts. Importantly, although DNA repair factors were less abundant on replicated chromatin upon transcription inhibition, the level of the DNA damage associated histone modification, γH2AX were not (Figs. 3C and EV5B,C). This may reflect that these factors are recruited at the fork due to problems arising ahead of the replisome. Alternatively, these factors may be handling another type of transcription-related stress that is not marked by γH2AX.

## TF access to newly replicated DNA can be stimulated by RNAPII binding

The third category of proteins affected by DRB or TPL were TFs (Fig. 1F,G). Chromatin accessibility studies suggested that TFs can exhibit increased or reduced association with newly replicated DNA (Stewart-Morgan et al, 2019). From the 193 TFs identified in our study, in DMSO, 69 TFs increased association (36%) and 26 TFs (13%) reduced association (Fig. 4A,B). Of note, 34 of the 193 TF identified are pioneer factors (Lambert et al, 2018; Sherwood et al, 2014), and consistent with previous work (Alvarez et al, 2023), they bind to replicated DNA as other TFs (Fig. EV6A). Upon TPL and DRB treatments, generally, TF binding kinetics were unchanged (Fig. 4C,D). However, for 13 TFs (~7%) their abundance dropped, either in TPL only (purple) or TPL and DRB (Brown). Therefore, only a subset of TFs relies on transcription to access replicated DNA. Moreover, transcription is not critical for the eviction of TFs from replicated DNA. This result was surprising as chromatin accessibility studies suggested that transcription restart would promote the eviction of TFs from gene bodies (Stewart-Morgan et al, 2019). Therefore, to independently verify our observations, we performed a PLA between EdU and ZNF462, a TF whose eviction from replicated DNA should not be affected by transcription inhibition (Fig. 4D). Quantification of the PLA signal using QIBC confirms that ZNF462 gains transient access to newly replicated DNA (Fig. 4F, DMSO). Moreover, its eviction is not affected by TPL or DRB treatments (Fig. 4F, TPL and DRB). Altogether these results support that only a small fraction of TFs rely on RNAPII binding to access replicated DNA. Moreover, binding, retention, or eviction of TFs from replicated DNA is overall not dictated by transcription activity.

## RNAPII stabilises the association of many chromatin remodelling enzymes on newly replicated chromatin

The last group of proteins significantly affected by TPL and DRB were the chromatin remodellers (Fig. EV2B). Members from the four main families of remodellers were identified on replicated chromatin, SWI/SNF (BAF and pBAF), ISWI, CHD and INO80/

SWR1 (Fig. 5A). Upon TPL or DRB treatment, several components within each complex were significantly affected (Fig. 5B–E, in bold). To estimate the effect of each drug treatment on each family, statistical significance was estimated by gene set enrichment for each remodeller family using a bootstrap (see 'Methods' for significance determination). This analysis revealed that all four families were significantly depleted from replicated chromatin upon TPL treatments (Fig. 5F), and that the BAF complex is depleted upon DRB treatment. As a control, as TPL blocks the ATPase activity of XPB, we tested whether TPL could be blocking the ATPase activity of the remodelling enzymes, leading to changes in their association with chromatin (Kim et al, 2021). To do so, we selected SMARCA4 (BRG1) and examined its ability to slide nucleosomes upon TPL treatment in vitro. In this assay, TPL treatment did not impair the ability of SMARCA4 to slide nucleosomes (Fig. EV6B). Taken together, these results suggest a role for RNAPII in stabilising the association of chromatin remodellers with newly replicated chromatin. The sensitivity of the BAF complex to DRB treatment suggests this complex is also stabilised by transcriptional elongation.

## Many RNAPII-dependent effects are specific to nascent chromatin

To test if the dissociation of several chromatin remodellers from newly replicated chromatin upon TPL and DRB treatment was specific to newly replicated chromatin or could be applicable to steady-state chromatin, we used two independent approaches. First, using QIBC, we measured the level of the SWI/SNF component SMARCA4 on steady-state chromatin upon TPL and DRB treatments. To do so, cells were pre-extracted with 0.5% triton prior to fixation, and chromatin-bound SMARCA4 levels measured. As controls, EU, RNAPII and RNAPII-pS5 were included in the analysis. Upon TPL treatment as expected, EU intensities and chromatin-bound RNAPII and RNAPII-pS5 levels decreased (Fig. EV6C–E). The loss of SMARCA4 on the other hand was modest (Fig. EV6F, less than a 2% reduction).

As a second approach, to enable systematic comparison of proteins bound to nascent and steady-state chromatin, EdU-labelled chromatin was examined after a 24 h thymidine chase period (Fig. 6A). In this set-up, the whole-cell population and the chased EdU-positive cells exhibit a similar cell cycle distribution (Fig. 6B). This supports that the EdU-labelled chromatin analysed by TMT-iPOND is no longer newly replicated, and instead has fully reverted to the steady state (Alabert et al, 2015). Experiments were performed in triplicate and 2491 proteins were identified (Fig. EV6G; Dataset EV7-8). In these conditions, RNAPII was significantly depleted from chromatin upon TPL treatment (Fig. 6C). However, no individual remodeller subunits (Fig. 6C,D),

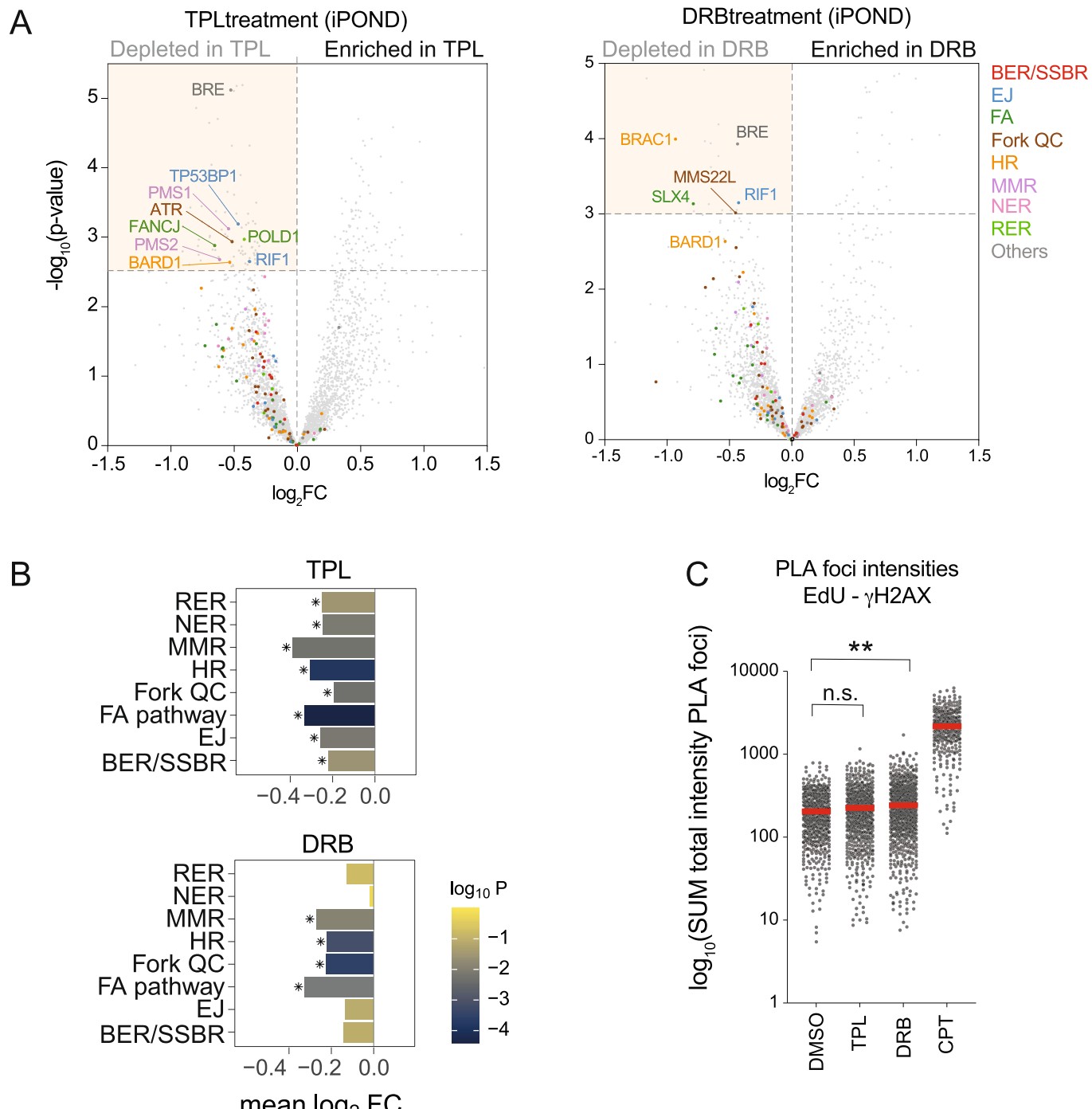

**Figure 3. Blocking transcription reduced the recruitment of DNA repair proteins on replicated chromatin.**

(A) Volcano plot generated using the iPOND-TMT-MS data displaying protein fold changes based on the full model for TPL treatment (left) or DRB treatment (right). Limma *t* test, the dashed horizontal line shows the *P* value cut-off (FDR 0.05). DNA repair proteins of different pathways are highlighted. The light red square highlights proteins that are significantly depleted upon TPL or DRB treatment. (B) Bar plot showing the mean log2-fold changes for each DNA repair pathway determined based on the full model, for DRB and TPL treatments using a bootstrap approach (see details in 'Methods'). The *P* value was derived from the proportion of bootstrap means that were lower than the observed group mean. Significant mean log2-fold changes are labelled with an asterisk, FDR < 0.05. (C) PLA between EdU and γH2AX in cells treated with DMSO, TPL and DRB. CPT is used as a positive control for the assay. PLA signal was calculated as the SUM of the total intensity of PLA foci per nucleus and normalised according to EdU intensity per cell. >750 nuclei were analysed per condition. Unpaired Mann–Whitney *t* test; **P* value = 0.0031; n.s. non-significant. Source data are available online for this figure.

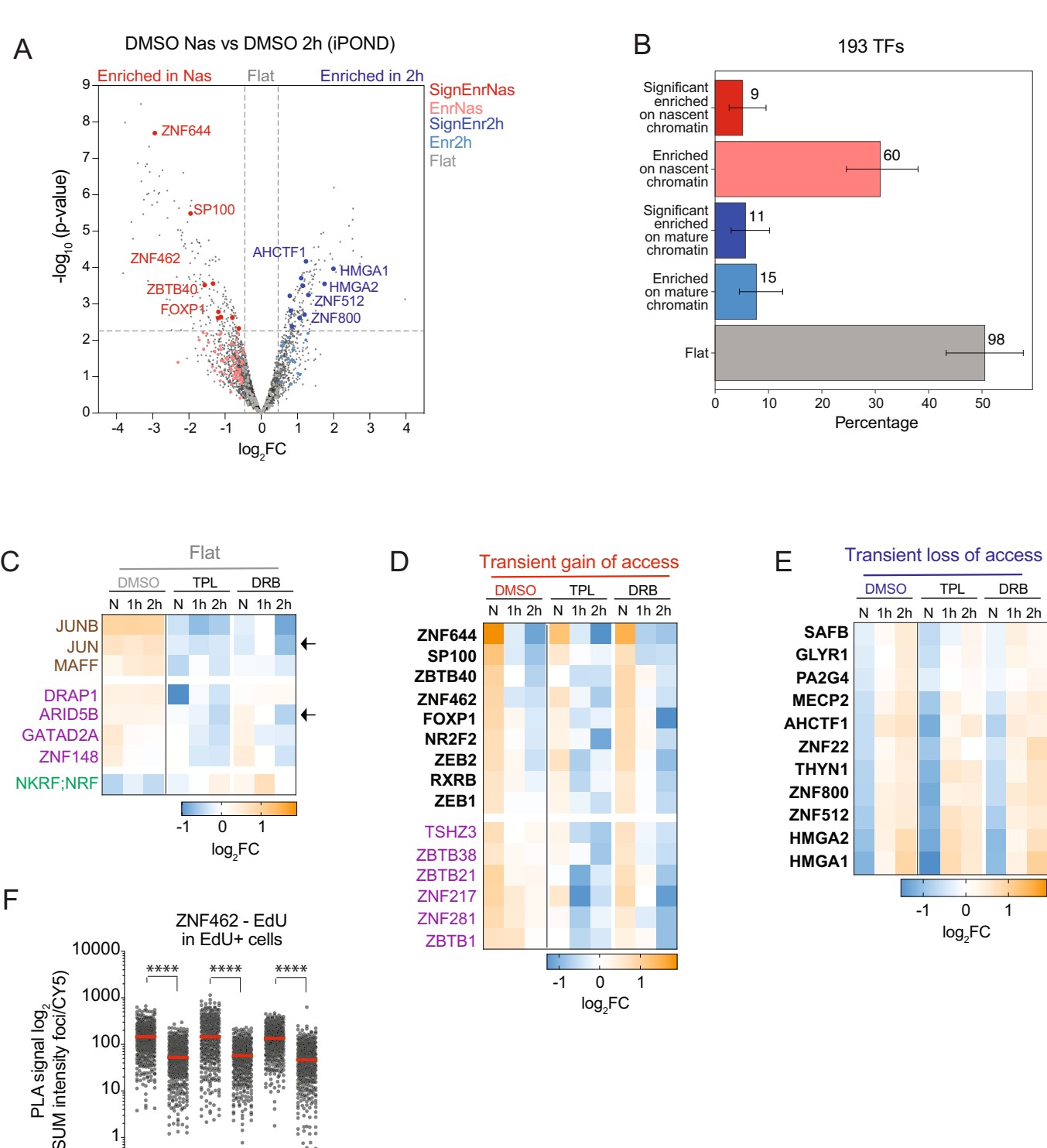

and none of the gene set enrichments for families were significantly affected by TPL or DRB treatments (Fig. 5F, steady-state chromatin). Of note, although non-significant, DRB affect the stability of pBAF and ISWI remodelers which would be consistent with promoter-proximal functions for these enzymes.

The identification of TPL and DRB-dependent proteins in the 24 h mature chromatin provides a means of establishing whether the association of TFs and DNA repair factors is also specific for nascent chromatin. Of the 13 RNAPII-dependent TFs identified in Fig. 1F,G, none were significantly affected in steady-state chromatin.

**Figure 4.  TF access to newly replicated DNA can be stimulated by RNAPII binding.**

(A) Volcano plot generated using the iPOND-TMT-MS data comparing protein fold changes between nascent chromatin and 2 h mature chromatin in DMSO-treated cells. Limma *t* test, significantly changing transcription factors with a magnitude fold change ≥0.5 (*x* axis) and an FDR ≤ 0.05 (*y* axis) are highlighted in dark red (left) and dark blue (right). TFs with a non-significant fold change of ≥0.5 are highlighted in light red and dark blue. TFs with a fold change below 0.5 are shown in grey. The dashed horizontal line shows the *P* value cut-off, and the two vertical dashed lines indicate transcription factors enriched either on nascent chromatin (left) or mature chromatin (right). The list of human TFs was generated based on two studies (Lambert et al, 2018; Lovering et al, 2021). (B) Bar graph showing the number of TFs identified in the five categories stated in Fig. 3A. *N* = 4 biological replicates. Errors shown as 95% confidence intervals of a proportion. (C) Heatmap of TFs that are categorised as "Flat" in D/E but display significant fold changes based on the full model for TPL (purple) or DRB (green) treatments (see Fig. 1F,G). The heatmap was generated using the log2-fold change of batch-corrected abundance with respect to the protein mean (*n* = 4 biological replicates). Each column represents a time point (N: Nascent, 1 h, and 2 h) and each row corresponds to the protein indicated on the left. Proteins that also show significant lower abundance in the whole-cell extract (Fig. 1H,I, full model) are highlighted with an arrow. Colour scale is indicated below. (D, E) Same as in (C) for TFs that transiently gain (D) or lose (E) access to nascent chromatin. Factors that displayed significant fold changes between nascent and mature chromatin in DMSO-treated cells are shown in bold. (F) Same as in Fig. 3C. PLA signal of EdU-ZNF462 interaction shown for nascent chromatin (N) and 2 h mature chromatin (2 h) in DMSO vs. TPL or DRB treated cells. >488 nuclei were analysed per sample. Unpaired Mann–Whitney *t* test; ****P* value < 0.0001. *N* = 2 biological replicates, one representative experiment is shown. Source data are available online for this figure.

This indicates that as with remodellers, RNAPII has a distinct effect on TF recruitment to nascent chromatin. Of the DNA repair pathways identified as TPL and DRB sensitive in nascent chromatin, only HR is sensitive to nascent and 24-h mature chromatin (Fig. 6E). MMR is sensitive to DRB in steady-state chromatin and FA is sensitive to TPL. This indicates that repair pathways also exhibit distinct dependencies in nascent chromatin. Taken together, these observations revealed that RNAPII-mediated stabilisation of TF and chromatin remodellers is specific to newly replicated chromatin.

## Discussion

We show that on replicated chromatin, RNAPII mediates the association of chromatin remodellers, transcription factors and DNA repair factors (Fig. 6F). In contrast, we find that in nascent chromatin RNAPII does not influence the restoration of most histone modifications or the cognate enzymes that generate them. Moreover, in most cases RNAPII-mediated effects are distinct from effects observed in steady-state chromatin. How RNAPII exhibits these distinct effects in newly replicated chromatin is not clear. However, the c-terminally S5 phosphorylated form of RNAPII is reduced in nascent chromatin (Fig. EV1H) (Fenstermaker et al, 2023; Stewart-Morgan et al, 2019). This indicates that RNAPII associated with nascent chromatin is likely to have a substantially distinct spectrum of c-terminal modifications. As the RNAPII carboxy-terminal domain (CTD) serves a platform for the recruitment of cofactors (Harlen and Churchman, 2017), this would be expected to affect the downstream consequences of RNAPII recruitment.

Proteins identified on nascent chromatin in this study are in good accordance with previous mass spectrometry-based analysis of nascent chromatin by iPOND or NCC (Alabert et al, 2014; Alvarez et al, 2023; Cortez, 2017; Nakamura et al, 2021; Sirbu et al, 2012). One noted difference is the abundance in histone. Histone abundance on newly replicated DNA increases only slightly upon maturation by NCC (Alabert et al, 2015; Alabert et al, 2014; Nakamura et al, 2021), while by iPOND histone abundance increases more strikingly (Dungrawala and Cortez, 2015; Sirbu et al, 2011). One possible explanation is the duration of the nucleoside labelling, 15–20 min by NCC compared to 11 min by iPOND. Shorter nucleoside labelling pulse enables to isolate a higher proportion of nucleosome-free DNA fragments, and therefore lead to a more pronounce histone accumulation upon

maturation during the chase period. Alternatively, it may reflect differences between the two technologies such as the efficacy of the nucleoside labelling (Biotin-dUTP vs EdU) or the required Click-it reaction in iPOND that may shear the nascent DNA or affect the nucleosomal density.

The stabilisation of remodellers by RNAPII may result in the generation of a transient chromatin environment in which TF's and other DNA-binding proteins associate transiently as an intermediate step in attaining the binding profiles observed in steady-state chromatin. Analogous to this, active transcription promotes nucleosome organisation at promoters on newly replicated DNA (Stewart-Morgan et al, 2019; Vasseur et al, 2016). Interestingly, our data showed that the BAF complex is not retained on replicated chromatin upon prolonged promoter-proximal RNAPII pausing in the presence of DRB (Fig. 4F, DRB). This suggests that the association of BAF complexes with nascent chromatin is dependent on transcriptional elongation. This could be a result of direct association with elongating RNAPII or as a result of chromatin transitions occurring as a consequence of transcription. For example, BAF might be recruited to sites of transcription-replication conflicts (Bayona-Feliu et al, 2023). Alternatively, transcriptional elongation might generate disrupted nucleosomes that recruit BAF complexes (Brahma and Henikoff, 2024).

Finally, in this study, we find evidence that the eviction of TFs from newly replicated chromatin is not dependent on transcription. The observation of this transient overloading is interesting as it resonates with previous ideas that replication provides a window of opportunity for the establishment of new gene regulatory architectures in daughter cells (Stewart-Morgan et al, 2020). It is also consistent with previous observations that sites of spurious chromatin accessibility may arise in newly replicated chromatin over coding genes (Stewart-Morgan et al, 2019). In this case, transcription was observed to restore coding gene chromatin. As the majority of the genome is not transcribed at high frequency alternative mechanisms are likely to regulate where TFs remain stably associated within intergenic chromatin. One possibility is that where TFs bind and can form multivalent interactions, they are more likely to remain associated. Many TFs can form direct or indirect interactions with RNAPII. In this respect RNAPII may act as a hub stabilising via looping interactions the association of TFs with adjacent loci. The effects of transcriptional inhibitors on factor loading are broadly consistent with this. Prevention of RNAPII loading with TPL results in reduced association of several TFs, chromatin remodelling enzymes and histone modifying enzymes that may be capable of forming direct or indirect interactions with RNAPII. Obtaining further

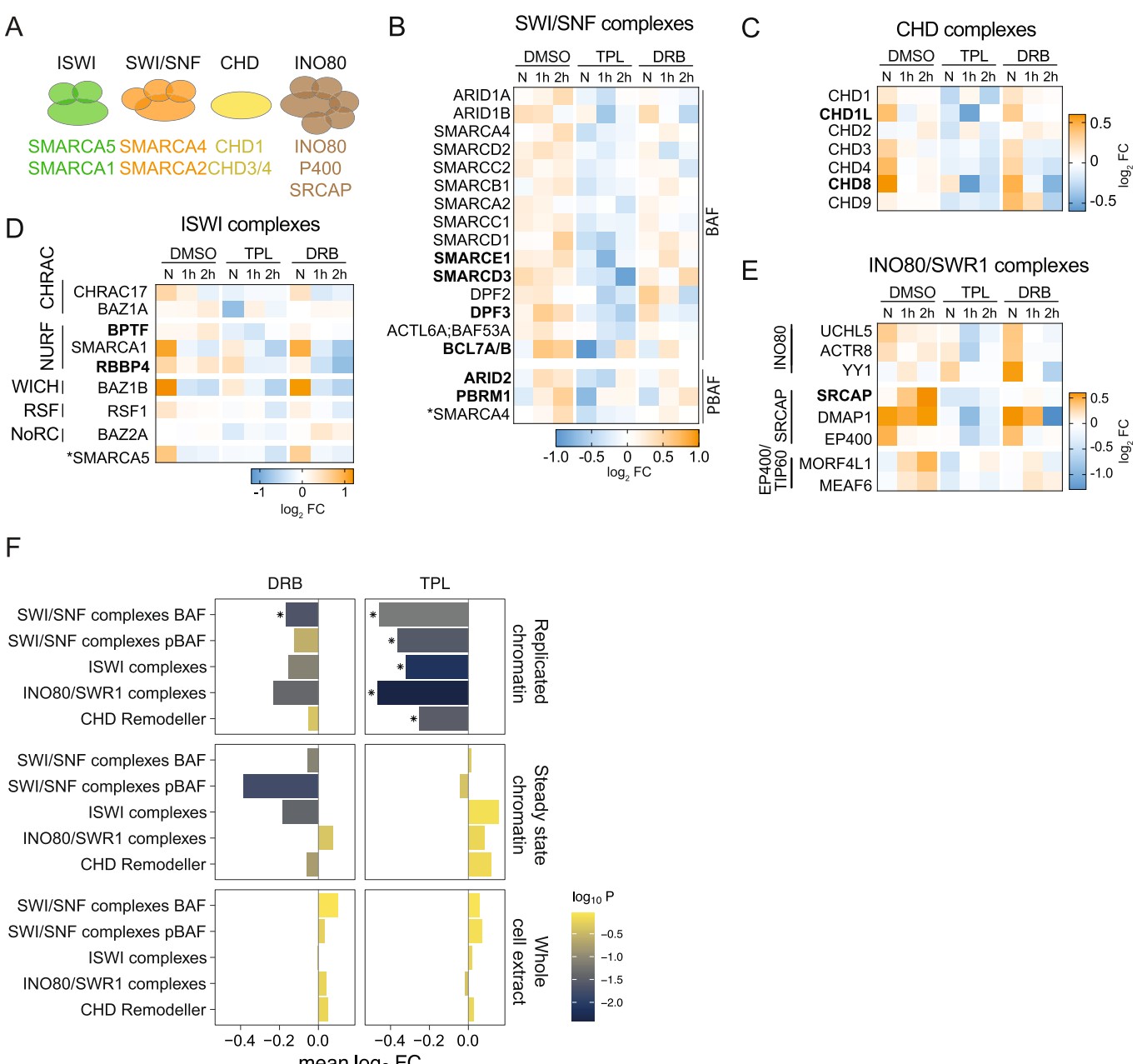

**Figure 5. The abundance of chromatin remodeller on replicated chromatin is impaired upon TPL treatment.**

(A) The four families of ATP-dependent chromatin remodelling complexes and their catalytic subunits below (ATPases). (B–E) Heatmaps of subunits belonging to the four families of ATP-dependent chromatin remodelling complexes: SWI/SNF (B) CHD (C), ISWI (D), and INO80/SWR1 (E). The heatmaps are generate using the log2-fold change of batch-corrected abundance with respect to the protein mean ($n = 4$ biological replicates). Each column represents a time point (N: Nascent, 1 h, and 2 h) and each row corresponds to the protein indicated on the left. Proteins present in more than one complex are highlighted with a star. Colour scale is indicated. Remodellers with significant fold changes (Limma $t$ test, FDR < 0.05) based on the full model for TPL and DRB treatment (Fig. 1F/G) are shown in bold. (F) Bar plot showing the mean log2-fold changes for the for families of remodeller complexes determined based on the full model for TPL and DRB treatment using a bootstrap approach (see details in 'Methods'). The $P$ value was derived from the proportion of bootstrap means that were lower than the observed group mean. Significant mean log2-fold changes are labelled with an asterisk, FDR < 0.05. The analysis was performed for the replicated chromatin (top), steady-state chromatin (middle), and whole-cell lysate (bottom).

support for this model may be challenging as the association of many factors in S phase may be more distributed than at other stages of the cell cycle and as a result more difficult to detect by genomics approaches. Yet, if true, from a cell fate point of view, the access on newly replicated DNA to previously prohibited gene regulatory regions,

or the replication-coupled "window of opportunity" hypothesis, may be narrower in time than anticipated (Stewart-Morgan et al, 2020; Yadav et al, 2018). Studies in systems with defined transcriptional changes will allow to further explore this process, and to identify the functional differences between the different groups of TFs.

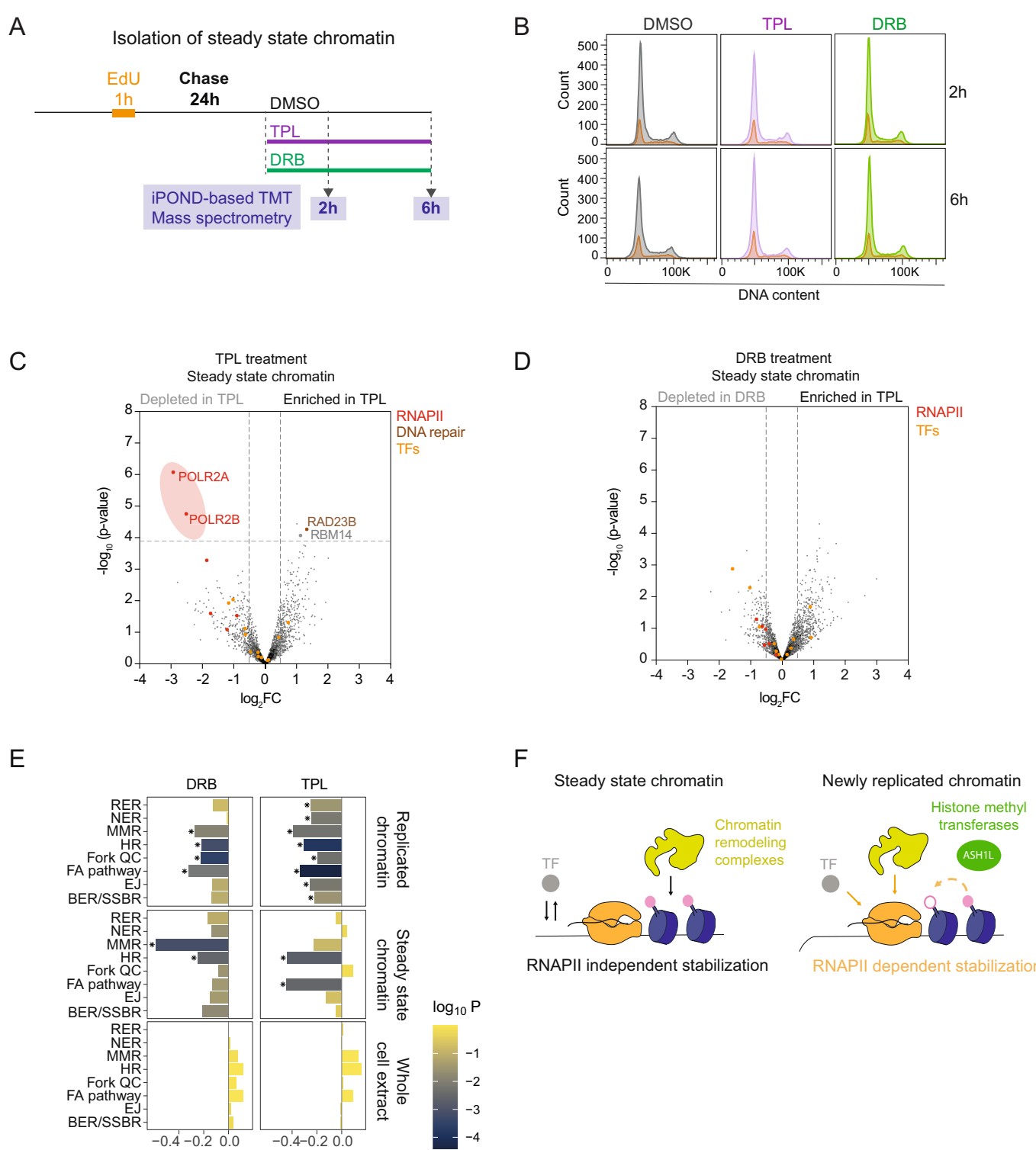

## Limitations of the study

The proteomics approach we have taken is not locus-specific meaning that we cannot distinguish the association of proteins to transcriptionally active or inactive regions. As a small proportion of the genome is transcriptionally active, it is possible our study underestimates transcription dependencies. On the other hand, it enables to avoid cell cycle synchronisation that is known to introduce changes on chromatin (Ly et al, 2015). In line with this, TPL/DRB may have a partially adverse effect on genome replication. Although we did not detect major defects in EdU incorporation rates (Fig. 1D), and of EdU pattern distributions

Figure 6.  Many RNAPII-dependent effects are specific to nascent chromatin.

(A) Experimental design of the iPOND-TMT 24 h chase experiment in TIG-3 cells upon DMSO, TPL or DRB treatment. (B) FACS histograms showing an overlay of the DNA content of the whole-cell population or EdU-positive cells only (orange) for cells treated for 2 h or 6 h with DMSO, TPL or DRB. (C, D) Volcano plots generated using the data from the iPOND-TMT 24 hr chase experiment, displaying protein fold changes based on the full model for TPL (C) and DRB (D) treatment on total chromatin. Limma t test, significantly changing proteins with a magnitude fold change ≥0.5 (x axis) and an FDR ≤ 0.05 (y axis) are highlighted. RNAPII, red; DNA repair, brown; Transcription factors, orange. The dashed horizontal line shows the P value cut-off, and the two vertical dashed lines indicate proteins enriched or depleted upon TPL treatment. N = 3 biological replicates. (E) Bar plot showing the mean log2-fold changes for the DNA repair pathways determined based on the full model for TPL and DRB treatment using a bootstrap approach (see details in 'Methods'). The P value was derived from the proportion of bootstrap means that were lower than the observed group mean. Significant mean log2-fold changes are labelled with an asterisk, FDR < 0.05. The analysis was performed for the replicated chromatin (top), steady-state chromatin (middle), and whole-cell lysate (bottom). (F) Model summarising the key findings from this study.

(Fig. EV1D), ChOR-seq-based experiments will allow to further investigate the site-specific effect of transcription inhibition in chromatin restoration. Moreover, the formaldehyde cross-linking used in iPOND makes it difficult to distinguish what is DNA bound versus what is enriched due to protein-protein cross-linking. Furthermore, subunits of the same complex often exhibit different enrichments by iPOND, including subunits of the replisome (Cortez, 2017). This has been attributed to the different abundance in tryptic peptides between different subunits, limiting the ability of using iPOND enrichment to identify potential new members of a complex. Finally, the usage of transcription inhibitors could be avoided with the recent development of RNAPII degron cell lines, allowing to explore the role of RNAPII on newly replicated chromatin with greater precision.

# Methods

## Cell culture

The normal fibroblast cell line TIG-3-20, derived from Japanese foetal lung (JCRB0506, JCRB Cell Bank) was cultured in Dulbecco's Modified Eagle's Medium (DMEM, Gibco, 61965059) supplemented with 10% FBS foetal bovine serum (FBS, Gibco, 10270106) and 1% penicillin/streptomycin at 37 °C in 5% $CO_2$ and humid environment. For all large-scale iPOND experiments, $4 \times 10^6$ cells were seeded in 15-cm dishes 2 days prior to EdU labelling and processing (10 dishes per sample).

## DNA labelling

All samples were labelled with medium containing 5-ethynyl-20-deoxyuridine (EdU; Thermo Fisher scientific, E10187) at a final concentration of 20 μM for 11 min. Following labelling, nascent samples were immediately taken and processed. All mature samples were washed twice in 1× PBS and further incubated in fresh media containing 20 μM thymidine (Sigma, T1895) for the appropriate time interval before collection.

## Transcription inhibition

To inhibit transcription initiation, 2 h prior EdU labelling, media was changed for media containing triptolide (Sigma, T3652) at a final concentration of 1 μM. To block transcription elongation, 2 h prior EdU labelling, media was changed for media containing 5,6-dichlorobenzimida-zole 1-b-D-ribofuranoside (DRB; Sigma, D1916) at a final concentration of 50 μM. DMSO-treated controls

were treated 2 h prior to EdU labelling with 10 μl DMSO/plate (Sigma, D8418). Inhibitors and DMSO were included for the duration of EdU labelling and chromatin maturation in all experiments.

## Isolation of proteins in nascent DNA (iPOND)

iPOND was performed as described previously (Sirbu et al, 2012). In short, $1 \times 10^8$ asynchronised TIG-3-20 cells (per time point) were labelled with 20 μM EdU (Thermo Fisher) for 11 min. The thymidine-chased sample were washed twice and subsequently further incubated in medium containing 20 μM thymidine for the indicated times. For the iPOND-TMT 24 h chase experiment, cells were labelled with 20 μM EdU for 1 h, grown in medium containing 20 μM thymidine for another 24 h, and treated with DMSO, TPL and DRB 2 h or 6 h prior to harvesting. All samples were crosslinked with 1% formaldehyde (Sigma, F8775) for 15 min at RT followed by 5-min incubation with 0.125 M glycine to quench the formaldehyde. Afterward, cells were scraped from the plate (on ice), permeabilized in 0.25% Triton X-100 in PBS for 30 min at RT and washed with 0.5% BSA in PBS. To conjugate biotin to EdU-labelled DNA, click chemistry reactions were performed for 1.5 h in the dark (10 μM Biotin-azide (Thermo Fisher, B10184), 10 mM Sodium ascorbate, 2 mM $CuSO_4$ in 1×PBS). Cells were lysed (1% SDS in 50 mM Tris, pH 8.0, protease inhibitor), sonicated (Bioruptor, Diagenode), the lysate diluted 1:1 (v/v) with cold PBS containing protease inhibitor, and streptavidin beads (Thermo Fisher, 65002) used to capture the biotin-conjugated DNA-protein complexes. At this point, the protein lysate is collected to be analysed together with the iPOND pulldown. Captured complexes were washed extensively using lysis buffer and 1 M NaCl. The proteins were eluted under reducing conditions by boiling in 2× LSB sample buffer for 30 min.

## Tandem mass tag (TMT)-based MS for quantitative proteomics analysis (TMT MS)

iPOND samples were prepared using SP3 protocol for TMT labelling as described in (Hughes et al, 2014). In brief, protein samples were mixed with SP3 beads (1:10, protein: beads) and digested with trypsin (1: 50, trypsin:protein). For total extracts, equal amounts (100 μg) of each peptide sample were dried and dissolved in 100 μL 100 mM TEAB buffer (pH 8.0). For iPOND samples, all the peptides from each sample were dried and dissolved in 100 μL 100 mM TEAB buffer (pH 8.0). TMT labelling of each sample were followed by the TMT10plex Isobaric

Mass Tag Labelling Kit (Thermo Fisher Scientific) manual. The TMT-labelled peptide samples were further fractionated using offline high-pH reverse-phase (RP) chromatography, as previously described (Brenes et al, 2019). The 24 fractions were subsequently dried and the peptides re-dissolved in 5% formic acid and analysed by nanoLC–MS/MS system Orbitrap Fusion Tribrid Mass Spectrometer (Thermo Fisher Scientific), equipped with an UltiMate 3000 RSLCnano Liquid Chromatography System, as previously described (Brenes et al, 2019) The MS data were analysed using MaxQuant (v 1.6.7.0) and searched against Homo Sapiens database from Uniport (Swissport, downloaded at January 2020) (Cox and Mann, 2011). The TMT quantification was set to reporter ion MS3 type with 10plex TMT (LOT: UH285228). The detailed parameter file was uploaded together with the raw MS data. Protein groups output table from MaxQuant was cleaned, filtered and median normalised with Perseus (1.6.7.0) (Tyanova et al, 2016).

### Mass spectrometry data analysis

MaxQuant's output files, proteinGroups.txt, underwent analysis using R code available at https://github.com/bartongroup/MG_NCCProt. Within R, we imported the protein group files, filtering out proteins with fewer than three unique razor peptides, as well as those identified as reverse sequences or potential contaminants. For subsequent analyses, intensities were log-transformed (base 10) and normalised to the median. Both PCA and clustering plots unveiled pronounced batch effects, predominantly associated with distinct mass spectrometer runs. To mitigate these effects, we employed HarmonizR (Voss et al, 2022) version 0.0.0.9000, utilising the *limma* method. Subsequent PCA and clustering verified the substantial attenuation of these batch effects. For determining differential abundance, we used *limma* (Ritchie et al, 2015) version 3.58.0, applying a model defined as ~ treatment + time_point. Multiple test corrections were conducted using the Benjamini–Hochberg method (Benjamini and Hochberg, 1995), with proteins exhibiting significant changes between the treatment baseline (DMSO) and the Nascent time point determined by a false discovery rate of less than 0.01. A separate *limma* analysis that incorporated interactions revealed no significant proteins in interaction terms, substantiating the treatment and time point's independence. For clustering, each protein (row in the heatmap), the mean of batch-corrected logarithmic intensities was found and subtracted from the logarithmic intensities. As this was done in logarithmic space, the resulting values represent log fold changes with respect to the mean. This approach removed absolute intensities and allowed for the comparison of relative intensity variations across different conditions. The rows in the heatmap were clustered using complete-linkage hierarchical clustering with Euclidean distance metric. Heatmaps, scatter plots and volcano plots were generated in GraphPrism. Gene Ontology analysis of biological processes was performed using STRING (Szklarczyk et al, 2015).

### Immunoblotting

Total cell extracts were prepared using 2× Laemmli Sample buffer (100 mM Tris-HCl (pH 6.8), 4% SDS, Glycerol), and the protein amount was determined using the Pierce BCA Protein Assay (23227, Thermo Fisher Scientific). After measuring the protein concentration, NuPAGE® Sample reducing agents (NP0009,

Invitrogen) and NuPAGE® LDS Sample Buffer (NP0007, Invitrogen) were added to the samples. After boiling at 95 °C for 5 min, the cell extracts or de-crosslinked samples were subjected to SDS–PAGE separation on NuPAGE 4–12% gels (Invitrogen, NP0321) in MES buffer. Proteins were transferred onto 0.2-μm pore size nitrocellulose membranes (GE Healthcare, 10600001) at 18 V for 1 h using Trans-Blot SD Semi-Dry Transfer Cell (Bio-Rad). Membranes were blocked in 5% skimmed milk in tris-buffer saline (TBS, 50 mM Tris, pH 7.5, 150 mM NaCl) supplemented with 0.1% Tween20 (TBS-T) for 1 h. The following dilution for each antibody were used: Anti-PCNA (1:1000, Abcam, ab29), anti-RNA polymerase II CTD repeat YSPTSPS [phospho S5] [4H8] (1:1000, ab5408, Abcam), anti-histone H3 (1:1000, ab10799, Abcam), anti-GAPDH (1:2000, 2118, Cell Signalling Technology), rabbit-HRP (1:5000, 115-035-062, Jackson Immunoresearch), and mouse-HRP (1:5000, 711-035-152, Jackson Immunoresearch). Signals from HRP-conjugated antibodies were revealed by Super-Signal™ West Pico PLUS Chemiluminescent (Thermo Fisher Scientific, 34580). Membranes were imaged using ChemiDoc XRS+ (Bio-Rad).

## Flow cytometry

For cell cycle analysis, cells were trypsinized, fixed with ice-cold 70% ethanol overnight at 4 °C, treated with propidium iodide (PI) solution (50 μg/ml PI, 50 μg/ml RNaseA, and 0.1% Triton X-100, 1% FBS in PBS) for 30 min and acquired using BD FACSCanto. Results were analysed using FlowJo software. For EdU detection, Alexa 647 was covalently linked to EdU by click chemistry reaction for 45 min in the dark (Alexa 647-azide (Thermo Fisher, A10277), 10 mM Sodium ascorbate, 2 mM CuSO$_4$ in 1× PBS).

## Microscopy

### Immunofluorescence

TIG-3-20 cells were seeded at 10,000 cells per well in clear bottom, black 96-well plates (Corning, 3340) and grown for 24 h. To measure DNA replication and transcription rates, cells were treated either with 20 μM EdU for 20 min or 1 mM 5-ethynyluridine (EU) for 1 h, respectively, prior to performing pre-extraction with ice-cold CSK buffer (10 mM PIPES (pH 7), 100 mM NaCl, 300 mM sucrose, 3 mM MgCl$_2$). After pre-extraction cells were washed once with 1× PBS and fixed with 2% formaldehyde for 20 min. For EdU detection, Alexa 647 was covalently linked to EdU using the Click-iT EdU Imaging Kit (Thermo Fisher Scientific, C10640). For EU detection, Alexa 488 was covalently linked to EU using the Click-iT RNA Alexa Flour 488 Imaging Kit (Thermo Fisher Scientific, C10329). Samples were blocked with BSA in PBS-Tween and antibodies were incubated at the following dilutions: anti-Biotin (1:1000, Vector Labs, SP-3000), anti-RNA polymerase II CTD repeat YSPTSPS [phospho S5] [4H8] (1:1000, ab5408, Abcam), anti-RNA polymerase II CTD repeat YSPTSPS [8WG16] (1:500, ab817, Abcam), anti-SMARCA4 (1:500, ab110641, Abcam), anti-ZNF462 (1:100, HPA022283, Merck), anti-rabbit IgG AF488 (1:1000, 1910751, Invitrogen), anti-Alexa Fluor 546 goat anti-mouse IgG(H + L) (1:1000, A11030, Invitrogen), and anti-Alexa Fluor 488 donkey anti-mouse IgG (H + L) (1:1000, A21202, Invitrogen). For DNA staining, the secondary antibodies were incubated together with DAPI (Thermo Fisher Scientific, 62248).

The cells were then washed twice in PBS-Tween, one time with PBS, and left in PBS until imaging.

### Proximal ligation assay (PLA)

Cells were fixed and permeabilised as described before. EdU was then covalently linked to biotin-azide (Thermo Fisher Scientific, B10184) using the Click-iT EdU Imaging Kit (Thermo Fisher Scientific, C10340). To allow data normalisation using the EdU signal, the Click-iT reaction was performed in the presents of Alexa 647-azide that was spiked in using a 40× dilution to the Biotin-azide. PLA between biotin and the protein of interest was performed according to the manufacturer instructions, using the Duolink® Proximity Ligation Assay from Sigma (DUO92006, DUO92014, DUO92002) or the Proximity Ligation kit from Navinci (Cambridge bioscience, NF.GR.100). After performing the PLA protocol, cells were stained with DAPI (Thermo Fisher Scientific, 62248) for 20 min in PBS, washed with PBS, and left in PBS until imaging.

### QIBC

Images were taken and analysed with ScanR High Content Screening Microscopy (Olympus). Data were visualised and statistically analysed in Tableau and GraphPrism.

## Remodeler assay in vitro

Nucleosomes were reconstituted on Cy3-labelled DNA, based on the 601 sequence, with a 17 bp DNA extension on one side and a 47 bp extension on the other side of the 601 sequence. Repositioning by Brg1 subcomplex was performed in 20 mM HEPES pH 7.5, 50 mM NaCl, 3 mM MgCl$_2$, 1 mM ATP, 100 nM each nucleosome, and 20 nM Brg1 in a 20 μL reaction for each condition with the specified amount of either DMSO, TPL or DRB. The reactions were carried out on ice and stopped with the addition of 100 ng/μL competitor DNA, 120 mM NaCl, and 2% sucrose. Repositioned nucleosomes were run on 6% PAGE/0.2× TBE gels in recirculating 0.2× TBE buffer for 3–4 h at 250 V. The percent of repositioned nucleosomes was analysed using Aida image analysis software and plotted in Microsoft Excel.

## Measuring PTMs on newly replicated chromatin upon transcription inhibition

The iPOND was performed as described before with two changes. The captured complexes were washed with lysis buffer and 1 M NaCl, followed by two washes with 50 mM Tris, pH 8.

### Histone extraction and digestion

TIG-3 cells were washed twice with PBS, and the cell pellets were then resuspended in an ice-cold Triton Extraction Buffer (TEB: PBS containing 0.5% Triton X-100, AEBSF, Protease Inhibitor Cocktail) at a cell density of 10e7 cells per ml. After lysing on ice for 10 min, cells were centrifuged at 2000 rpm for 10 min at 4 °C and the supernatant was discarded. To remove the residual cytoplasmic protein, the cell pellets were washed one more time with the TEB buffer and centrifuged again as before. The cell pellets were resuspended in 0.2 N HCl at a proper density ($4 \times 10e7$ cells/ml) and vortexed overnight at 4 °C. The supernatant acid extracts were collected after 10 min centrifugation at 6500 rpm. An equal volume of 50% trichloroacetic acid (TCA) was added and mixed on ice for 30 min. After centrifugation for 10 min at 10,000 rpm the pellets were washed with ice-cold acetone and centrifuged again at 10,000 rpm for 10 min. The samples were dried in a vacuum centrifuge. The pellet was dissolved in 50 mM ammonium bicarbonate, pH 8.0, and histones were subjected to derivatization using 5 μL of propionic anhydride and 14 μL of ammonium hydroxide (all Sigma Aldrich) to balance the pH at 8.0. The mixture was incubated for 15 min and the procedure was repeated. Histones were then digested with 1 μg of sequencing grade trypsin (Promega) diluted in 50 mM ammonium bicarbonate (1:20, enzyme:sample) overnight at room temperature. Derivatization reaction was repeated to derivatize peptide N-termini. The samples were dried in a vacuum centrifuge.

### Sample desalting

Prior to mass spectrometry analysis, samples were desalted using a 96-well plate filter (Orochem) packed with 1 mg of Oasis HLB C-18 resin (Waters). Briefly, the samples were resuspended in 100 μl of 0.1% TFA and loaded onto the HLB resin, which was previously equilibrated using 100 μl of the same buffer. After washing with 100 μl of 0.1% TFA, the samples were eluted with a buffer containing 70 μl of 60% acetonitrile and 0.1% TFA and then dried in a vacuum centrifuge.

### LC-MS/MS acquisition and analysis

Samples were resuspended in 10 μl of 0.1% TFA and loaded onto a Dionex RSLC Ultimate 300 (Thermo Scientific), coupled online with an Orbitrap Fusion Lumos (Thermo Scientific). Chromatographic separation was performed with a two-column system, consisting of a C-18 trap cartridge (300 μm ID, 5 mm length) and a Picofrit analytical column (75 μm ID, 25 cm length) packed in-house with reversed-phase Repro-Sil Pur C-18-AQ 3 μm resin. Peptides were separated using a 30-min gradient from 1–30% buffer B (buffer A: 0.1% formic acid, buffer B: 80% acetonitrile + 0.1% formic acid) at a flow rate of 300 nl/min. The mass spectrometer was set to acquire spectra in a data-independent acquisition (DIA) mode. Briefly, the full MS scan was set to 300–1100 $m/z$ in the orbitrap with a resolution of 120,000 (at 200 $m/z$) and an AGC target of $5 \times 10e5$. MS/MS was performed in the orbitrap with sequential isolation windows of 50 $m/z$ with an AGC target of $2 \times 10e5$ and an HCD collision energy of 30. Histone peptides raw files were imported into EpiProfile 2.0 software (Sidoli et al, 2016). From the extracted ion chromatogram, the area under the curve was obtained and used to estimate the abundance of each peptide. In order to achieve the relative abundance of post-translational modifications (PTMs), the sum of all different modified forms of a histone peptide was considered as 100% and the area of the particular peptide was divided by the total area for that histone peptide in all of its modified forms. The relative ratio of two isobaric forms was estimated by averaging the ratio for each fragment ion with different mass between the two species. The resulting peptide lists generated by EpiProfile were exported to Microsoft Excel and further processed for a detailed analysis.

## Quantification and statistical analysis

For the panel showing IF or PLA data, as intensities from immunofluorescence reveal non-normal distribution a non-parametric Mann–Whitney test was used to compare the different treatments. Principal Component Analyses were performed using

function prcomp in R version 4.3.1 In Figs. 3B, 5F, and 6E, for each protein group, the mean log2-fold change was determined based on the full model, for DRB and TLP treatments, respectively. To approximate the sampling distribution of this mean log2-fold change for these groups and treatments, a bootstrap approach was employed. A set, identical in size to the original group, was randomly chosen from all proteins, and its mean log2-fold change for both treatments was calculated. This bootstrapping procedure was iteratively performed 100,000 times. Subsequently, the observed mean log2-fold change for each group and treatment was compared with the simulated sampling distribution, and the p value was derived from the proportion of bootstrap means that were lower than the observed group mean.

## Data availability

The TMT mass spectrometry proteomics data have been deposited to the ProteomeXchange Consortium via the PRIDE (Perez-Riverol et al, 2022) partner repository with the dataset identifier PXD046514 (iPOND-TMT time course experiment), PXD040888 (Whole-cell extracts from iPOND-TMT time course experiment), PXD046546 (iPOND-TMT 24 h chase experiment), and are publicly available as of the date of publication. Mass spectrometry raw files of the label-free analysis of the histone modifications are available on the public repository Chorus (https://chorusproject.org) under project number 1812. Any additional information required to reanalyse the data reported in this paper is available from the lead contact upon request.

## Peer review information

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

## Acknowledgements

The authors thank Giulia Saredi, Kathleen Stewart-Morgan and Jane Wright for reading the manuscript. We thank Alejandro Brenes and Sara Ten Have for fruitful discussions and support in proteomics. We thank the Imaging facility, the Flow Cytometry and Cell Sorting facility and the Proteomics facility in School of Life Sciences. SB is supported by the ERC-Stg-IDRE. VA is supported by the CRUK-CDF C57404/A21782. HJ was supported by the Wellcome Collaborator Award (Ref: 206293/Z/17/Z) and by BBSRC grant (Ref: BB/V010948/1). Research in the Lamond lab is funded by BBSRC grant (Ref: BB/V010948/1) and UKRI grant (Ref: EP/Y010655/1). Research in the Owen-Hughes lab is funded by MRC grant MR/SO21647/1. The Sidoli lab gratefully acknowledges for funding from the NIH Office of the Director (S10OD030286). Research in the Alabert lab is funded by CRUK-CDF (C57404/A21782) and the European Research Council ERC-Stg-IDRE.

## Author contributions

**Susanne Bandau:** Conceptualisation; Validation; Investigation; Visualisation; Writing—original draft. **Vanesa Alvarez:** Formal analysis; Validation;

Investigation. **Hao Jiang**: Validation; Investigation. **Sarah Graff**: Formal analysis; Investigation. **Ramasubramanian Sundaramoorthy**: Formal analysis; Investigation. **Marek Gierlinski**: Data curation; Formal analysis. **Matt Toman**: Investigation. **Tom Owen-Hughes**: Formal analysis; Investigation; Writing—original draft; Writing—review and editing. **Simone Sidoli**: Formal analysis; Investigation; Writing—original draft. **Angus Lamond**: Formal analysis; Investigation; Writing—original draft; Writing—review and editing. **Constance Alabert**: Conceptualisation; Formal analysis; Supervision; Funding acquisition; Validation; Investigation; Visualisation; Methodology; Writing—original draft; Project administration; Writing—review and editing.

## Disclosure and competing interests statement

The authors declare no competing interests.

# Expanded View Figures

**Figure EV1.  Proteomic profiling of chromatin behind replisomes upon transcription inhibition. Related to Fig. 1.**

(A) QIBC-based analysis of RNAPII level in DMSO, TPL and DRB treated cells. Graphs show the mean intensity per nuclei, > 539 nuclei were analysed per sample. Red line: median. Unpaired Mann–Whitney *t* test; ****$P$ value < 0.0001. $N = 3$ biological replicates, one representative experiment is shown. (B) EdU pulse chase quantification by flow cytometry. Cells were pulsed for 11 min with EdU (20 μM) (Nascent sample) and chased 2 h by 20 μM of thymidine (Mature sample). Left, the gating for EdU-positive and negative cells is shown. Right, quantification of cell density (expressed as %) and EdU intensities (a.u.) for 1 N and 2 N populations. (C) Flow cytometry plots of asynchronous cells treated with DMSO, TPL and DRB. Cells were EdU pulsed and chased as described in (B). Nas, Nascent sample, red; 2 h, Mature sample, black. The percentage of overlapping area between Nas and 2 h is indicated. (D) Cells were treated for 2 h with DMSO, TPL and DRB, followed by a 20 min EdU pulse and analysed by microscopy. Left: Representative images of Early/Mid and Late S-phase patterns. Early and Mid-S-phase cells have a pattern of replication foci distributed throughout the nucleus. Late S-phase cells would have a small number of large foci within the nucleus. Right: Distribution of cells in G1 and G2 phase based on EdU and DAPI intensities. Distribution of cells in Early/Mid (top section) and late S phase (bottom section) based on the distinct EdU patterns shown on the left. The mean and standard deviation (S.D.) of >1230 nuclei are shown. $N = 2$ biological replicates. One representative experiment is shown. (E) Western plot showing the γH2AX level in cells treated with DMSO, TPL or DRB. Cells were treated with inhibitors for 3 h according to the concentrations indicated. Total cell extracts were collected and GAPDH used as a loading control. (F) Principal-component analysis of the four biological replicates based on the proteins identified. (G) Volcano plot generated using the iPOND-TMT time course data showing protein fold changes based on the full model for the 2 hr mature time point. $N = 4$ biological replicates, Limma t test, significantly changing proteins with a magnitude fold change ≥0.5 (*x* axis) and an FDR ≤ 0.05 (*y* axis) are highlighted. Replisome, red; Canonical Histones, blue; Histone variants, light blue. The dashed horizontal line shows the $P$ value cut-off, and the two vertical dashed lines indicate proteins enriched or depleted from 2 hr mature chromatin. (H) Western blot analysis of iPOND samples. The western blot was probed with antibodies against RNAPII-pS5, PCNA, and Histone H3 (indicated on the right). Sample labelling shown on top. (I) Heatmap of RNA Polymerase I, II, and II. The log2-fold change of batch-corrected abundance with respect to the protein mean is shown (n = 4 biological replicates). Each column represents a time point (N: Nascent, 1 h, and 2 h) and each row corresponds to the protein indicated on the left. Colour scale is indicated. (J) Left: Scheme of the PLA analysis between EdU and protein of interest by QIBC. TIG-3 cells are EdU-labelled, and the PLA signal between EdU and the protein of interest on nascent chromatin analysed (see details in 'Methods'). Right: Single-cell PLA signal of EdU-RNAPII interaction shown for nascent chromatin in DMSO, TPL and DRB treated cells. Cells with a similar EdU signal were chosen and the PLA signal was calculated as the SUM of the total intensity of PLA foci per nucleus. >78 nuclei were analysed per sample. Red line, median; Unpaired Mann–Whitney *t* test; ****$P$ value < 0.0001; n.s., non-significant. $N = 2$ biological replicates, one representative experiment is shown. Source data are available online for this figure.

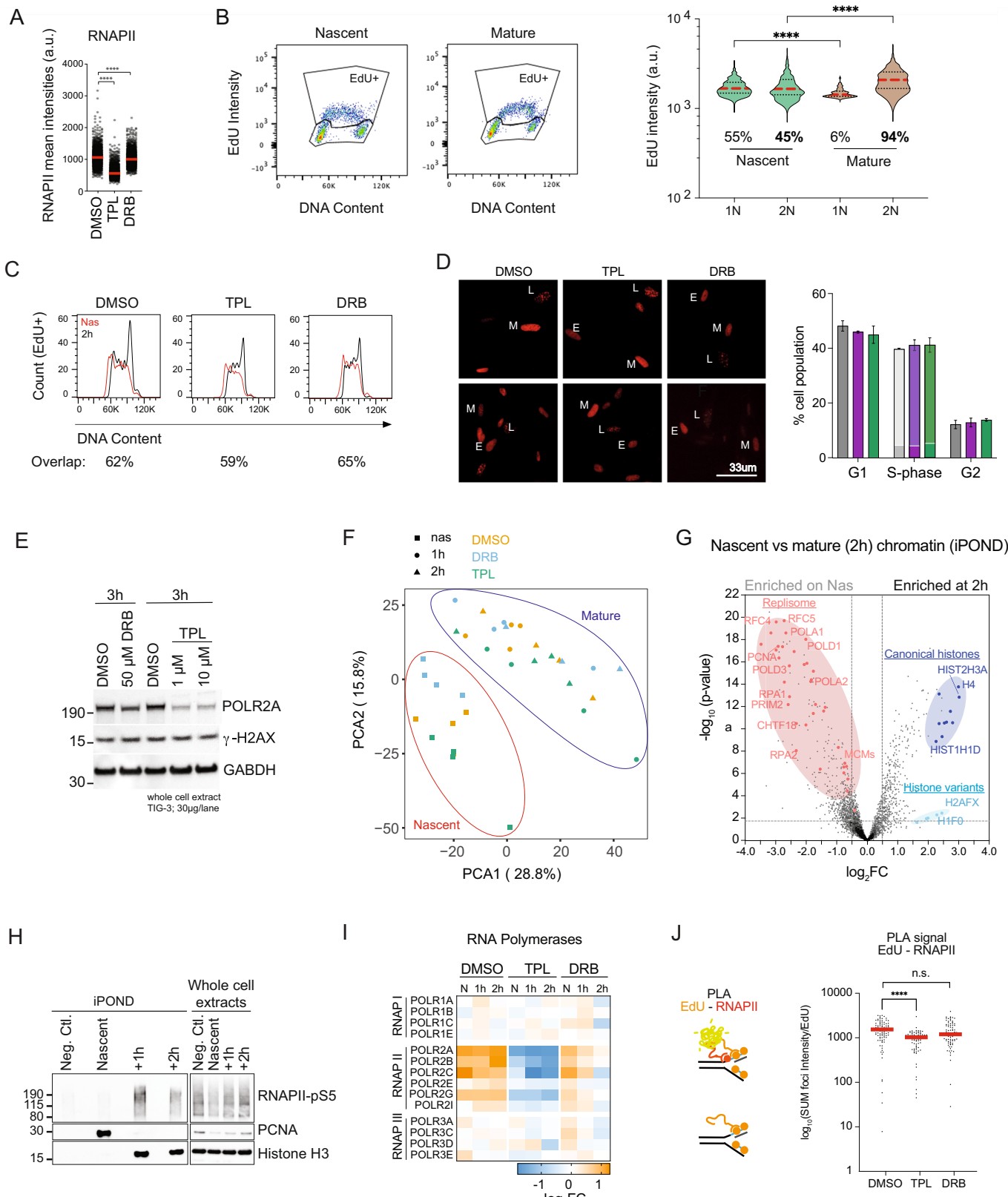

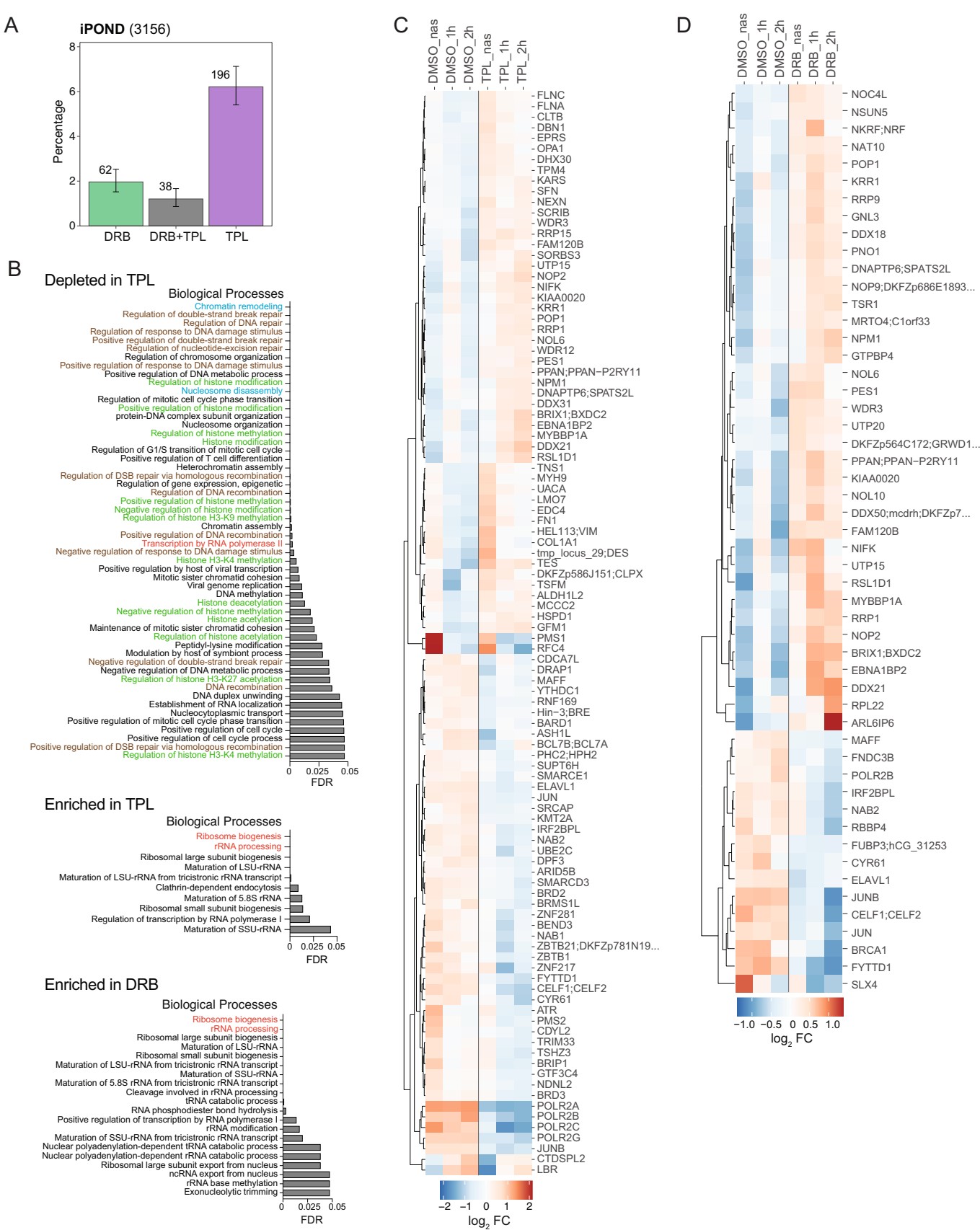

◀   **Figure EV2.   Proteomic profiling of chromatin behind replisomes upon transcription inhibition. Related to Fig. 1.**

(A) Histogram showing the number of factors that had a significant fold change based on the full model for TPL and DRB treatment shown in Fig. 1F,G. TPL + DRB: common factors between TPL and DRB treatment. $N = 4$ biological replicates. 95% confidence interval of a proportion is shown. (B) GO term analysis using STRING (Szklarczyk et al, 2015) with the significantly changing proteins based on the full model for TPL and DRB treatment shown in Fig. 1F,G. GO terms were filtered for a background gene count <400, chromatin related biological processes and an FDR below 0.05. The proteins were categorised into three groups: Depleted in TPL, enriched in TPL, and enriched in DRB. The GO terms have been colour coded based on four categories: Chromatin remodelling, blue; DNA repair, brown; Histone modification, green; Transcription, red. (C, D) The rows in the heatmaps were clustered using complete-linkage hierarchical clustering with Euclidean distance metric (see material and method for details). Source data are available online for this figure.

A
## Histone Chaperones

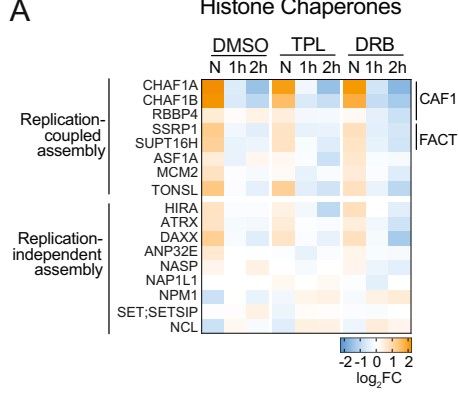

B
## Canonical Histones

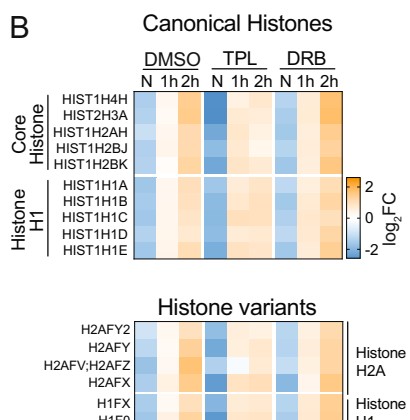

C
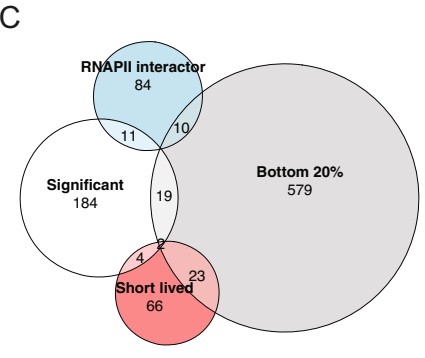

D

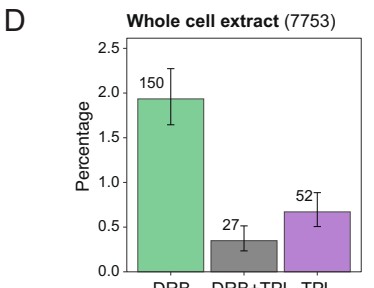

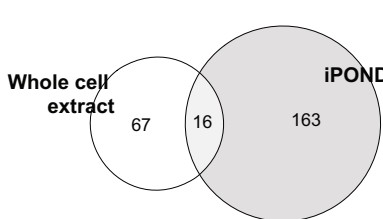

E
## Known proteins down-
## regulated upon TPL treatment

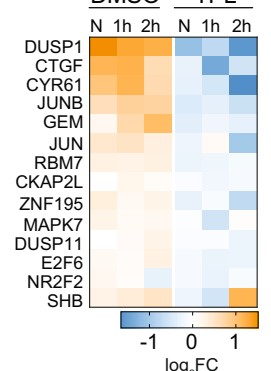

F

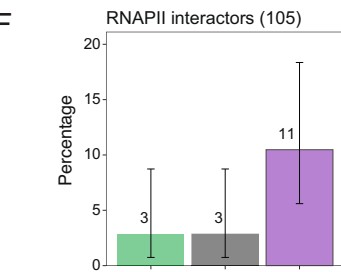

G

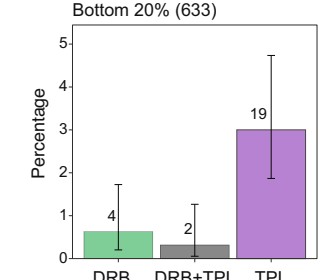

H

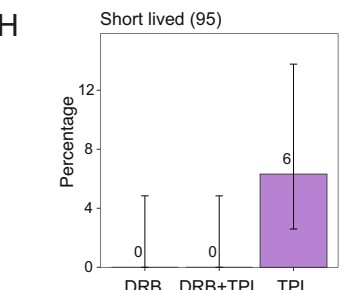

**Figure EV3.  Proteomic profiling of chromatin behind replisomes upon transcription inhibition. Related to Fig. 1.**

(A, B) Heatmaps of Histone Chaperones (A) and Canonical Histones/Histone variants (B). The log2-fold change of batch-corrected abundance with respect to the protein mean is shown ($n = 4$ biological replicates). Each column represents a time point (N: Nascent, 1 h, and 2 h) and each row corresponds to the protein indicated on the left. Colour scale is indicated. (C) Venn diagram between proteins with a significant fold change based on the full model for TPL and DRB treatment (white, Fig. 1F/G), identified RNAPII interactor (blue), identified short-lived proteins (red), and the 20% lowest abundant proteins identified (light purple). The lists of RNAPII interactor and short-lived proteins were generated based on (Ebmeier et al, 2017) and (Li et al, 2021), respectively. (D) Top: Same as in Fig. EV2A for significantly changing proteins in the whole-cell extract upon TPL or DRB treatment (shown in Fig. 1H,I). 95% confidence interval of a proportion is shown. Bottom: Venn diagram between significantly changing proteins (FDR < 0.05, full model for TPL and DRB treatment) from the whole-cell extract and the iPOND-TMT time course experiments. $N = 4$ biological replicates. (E) Same as in (A, B) for reported proteins by (Vispe et al, 2009) downregulated upon TPL treatment. $N = 4$ biological replicates. (F–H) Same as in Fig. EV2A for identified RNAPII interactor (F), the 20% lowest abundant proteins (G) and the identified short-lived proteins (H). $N = 4$ biological replicates. 95% confidence interval of a proportion is shown.

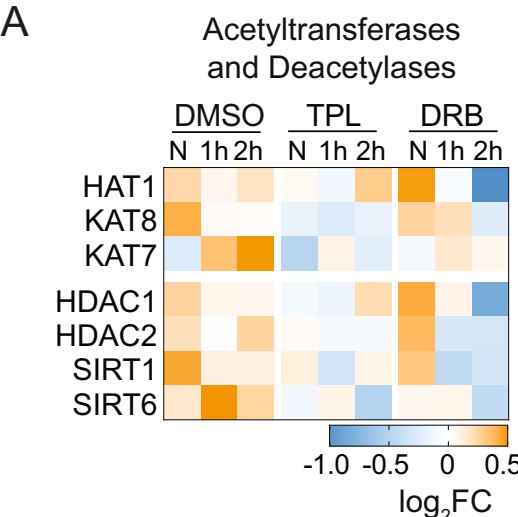

A

Acetyltransferases
and Deacetylases

B

C

◀ **Figure EV4. Transcription promotes H3.3K36me2 re-establishment on replicated chromatin. Related to Fig. 2.**

(A) Heatmap of Histone Acetyltransferases and Deacetylates ($n = 4$ biological replicates). The log2-fold change of batch-corrected abundance with respect to the protein mean is shown ($n = 4$ biological replicates). Each column represents a time point (N: Nascent, 1 h, and 2 h) and each row corresponds to the protein indicated on the left. Colour scale is indicated. (B) Proportion of H3K9me1, H3K9me2, and H3K9me3 (left to right) on newly replicated chromatin for each time point in DMSO (black), DRB (green) and TPL (purple) treated cells. Standard error of the mean is shown. $N = 3$ biological replicates. Paired $t$ test, **$P$ value < 0.01; n.s., non-significant. (C) Same as in (B) for H4K20me0, H4K20me1, and H4K20me2 (left to right).

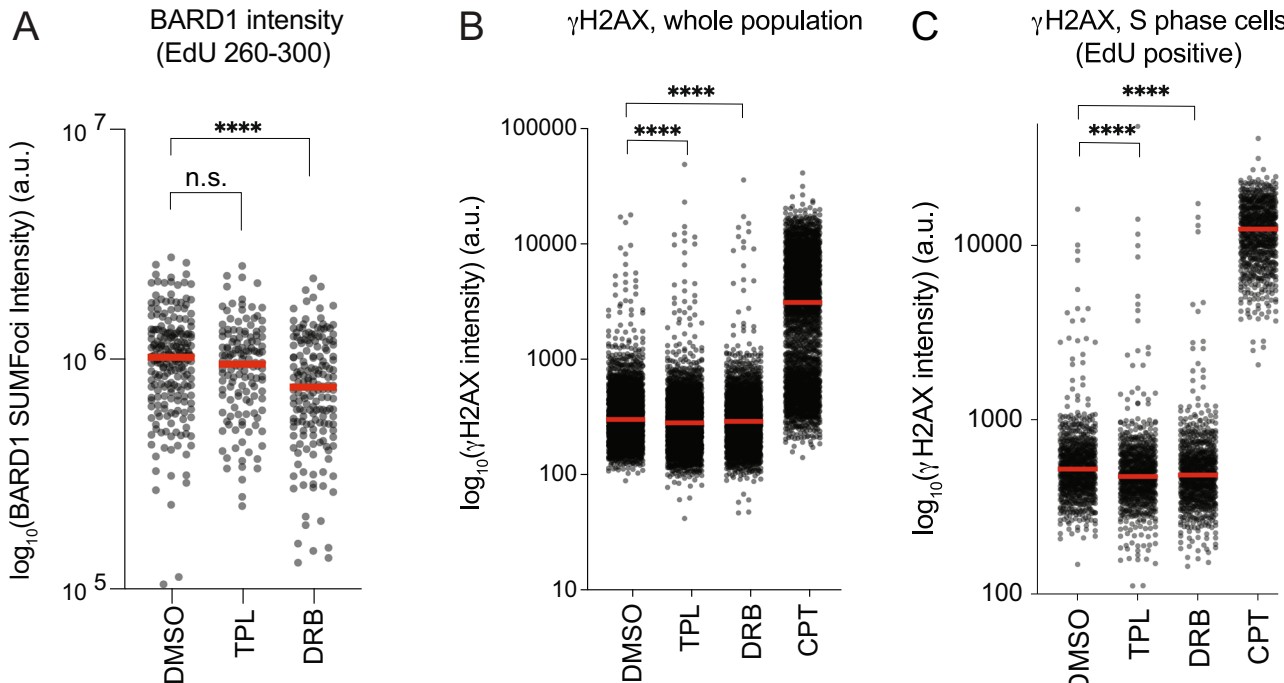

**Figure EV5. Blocking transcription reduced the recruitment of DNA repair proteins on replicated chromatin. Related to Fig. 3.**

(A) Single-cell PLA signal of EdU-BARD1 interaction shown for nascent chromatin in DMSO, TPL and DRB treated cells. Cells with a similar EdU signal were chosen and the PLA signal was calculated as the SUM of the total intensity of PLA foci per nucleus. >138 nuclei were analysed per sample. Red line, median; Unpaired Mann–Whitney $t$ test; ****$P$ value < 0.0001; n.s., non-significant. $N = 2$ biological replicates, one representative experiment is shown. (B, C) Quantification of $\gamma$H2AX signal in the whole single-cell population (B) and EdU-positive cells only (C). Quantification of $\gamma$H2AX fluorescence signals in individual cells was measured by QIBC. Cells were treated with TPL and DRB according to conditions used in the study and labelled with EdU for the last 20 min. Camptothecin treatment is used as a positive control for DNA damage. >600 nuclei were analysed per condition. Red line, median; Unpaired Mann–Whitney $t$ test; ****$P$ value < 0.0001. $N = 2$ biological replicates, one representative experiment is shown. Source data are available online for this figure.

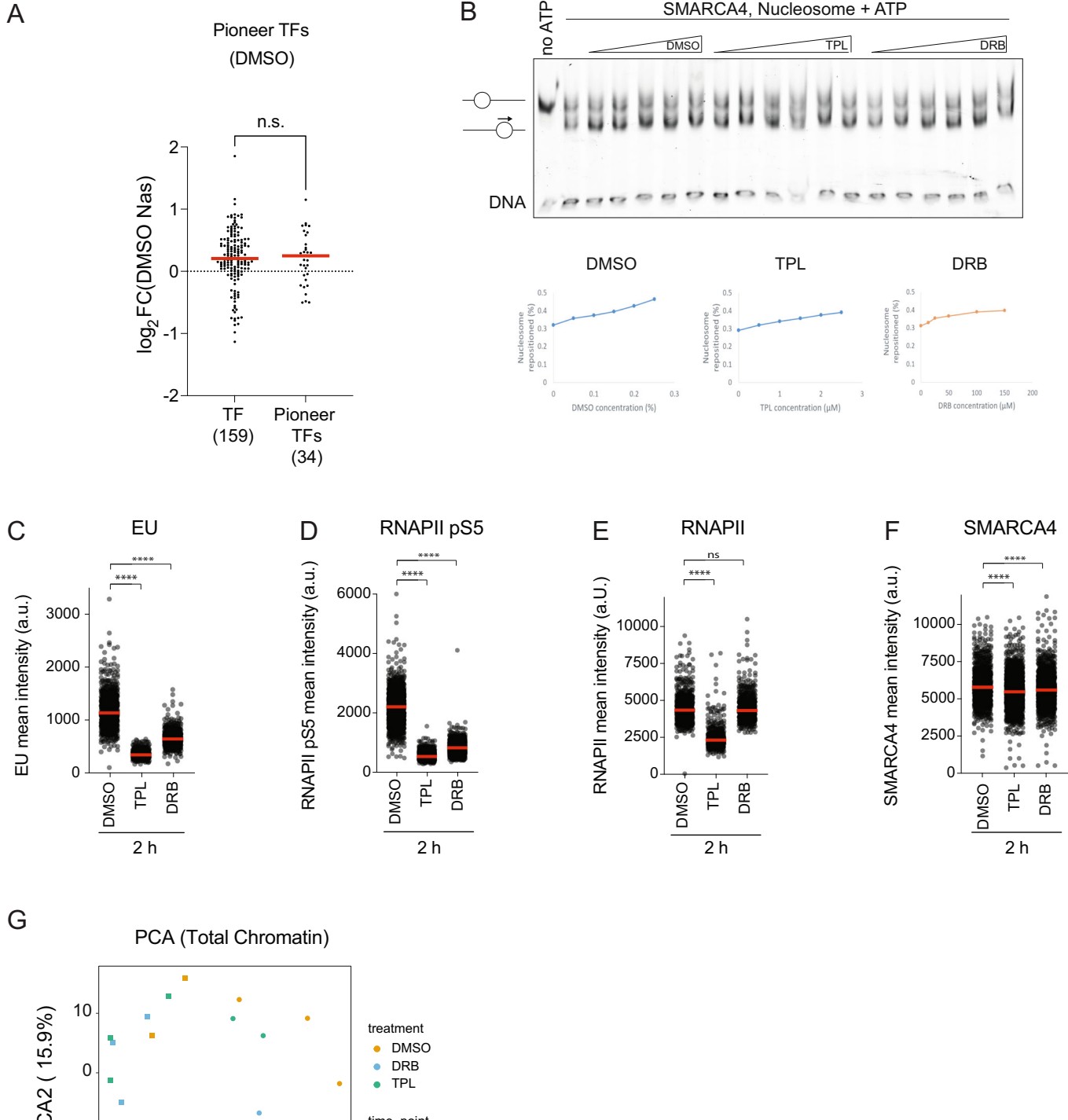

◄ **Figure EV6.   Analysis of TF and remodellers on replicated chromatin and of steady state chromatin. Related to Figs. 4, 5 and 6.**

(A) $\log_2$ fold change of batch-corrected abundance with respect to the protein mean for the nascent time point is shown ($n = 4$ biological replicates). TFs are divided into two groups, pioneer factors and non-pioneer (Sherwood et al, 2014). Red line, median; Unpaired Mann–Whitney $t$ test; n.s., non-significant. (B) Nucleosome sliding in vitro assay (left) and its quantification (right). (C–F) QIBC analysis of chromatin-bound intensities of EU, RNAPII-pS5, RNAPII and SMARCA4 in DMSO, TPL or DRB treated cells. Graphs show the mean intensity per nuclei, >291 nuclei were analysed per sample. Red line, median; Unpaired Mann–Whitney $t$ test; ****$P$ value < 0.0001; n.s., non-significant. $N = 3$ biological replicates, one representative experiment is shown. (G) Principal-component analysis using the identified proteins from the iPOND-TMT chase experiments. Source data are available online for this figure.

