## [Peer Review File · EMBO Reports]

RNA polymerase II promotes the organization of chromatin following DNA replication.

Susanne Bandau, Vanesa Alvarez, Hao Jiang, Sarah Graff, Ramasubramanian Sundaramoorthy, Matthias Toman, Tom Owen-Hughes, Simone Sidoli, Angus Lamond, Constance Alabert, and Marek Gierlinski

Corresponding author(s): Constance Alabert (calabert@dundee.ac.uk)

Review Timeline:	Transfer from Review Commons:	21st Jul 23
	Editorial Decision:	28th Jul 23
	Revision Received:	7th Nov 23
	Editorial Decision:	20th Dec 23
	Revision Received:	17th Jan 24
	Accepted:	23rd Jan 24

Editor: Esther Schnapp

Transaction Report: This manuscript was transferred to
EMBO reports following peer review at Review Commons.

Review
COMMONS

Review #1

1. Evidence, reproducibility and clarity:

Evidence, reproducibility and clarity (Required)

The goal was to characterize the changes in the composition of the proteome associated with replicated DNA in conditions of genome-wide inhibition of transcription initiation or transcription elongation. They use iPOND, a MS based technique that identifies proteins specifically associated with replicated DNA labeled with EdU. They use non-synchronized foetal human lung fibroblasts and examine time points immediately after replication (after the 11 min EdU pulse) and 1 and 2 hrs after the Thymidine "chase" later when chromatin has "matured", to assess how the inhibition of transcription influences chromatin maturation and the binding of chromatin associated proteins to replicated DNA.

The question is pertinent and is in line with the long-standing interest of the group in chromatin replication dynamics. They conclude that 1. RNAPII loading is necessary for the binding of some TFs, chromatin remodelers and DNA repair factors and 2. RNAPII elongation is needed for H2A.Z incorporation, H3K36me2 restoration and DNA repair factor binding. Transcription is on the other hand not needed for nucleosome assembly, histone acetylation and H3K9me3, H3K27me3 and H4K20me2 incorporation or restoration.

There are two main issues that make the interpretation of their results very difficult and make me question their conclusions:

1. They don't provide sufficient evidence that the treatments with TPL and DRB do not interfere with replication. The distributions of EdU intensity per EDU+ cell after treatment in Figures 1D-E and S1A are not sufficient. It is not clear why EdU incorporation is so heterogeneous in the cell population (the range of intensities goes from near 0 to 50000!), which makes me wonder if the DMSO treatment also has an effect on replication. I don't think this heterogeneity can simply be explained by the fact that the cell population is asynchronous. They need to show a -DMSO control as well. Besides since they are only using a positive EdU signal as their criteria for replicating cells, they cannot rule out that some of the EdU signal is coming from DNA repair after replication and depending on how deleterious DMSO/TPL/DRB are to replication the fraction of cells that undergo DNA repair might be significant. More importantly, they need to show that the various treatments don't interfere with the replication program, especially since replication is coupled with new nucleosome

assembly and the transcription of replication dependent histone variants is induced during S-phase. Transcription inhibition could disproportionately affect the replication of some parts of the genome more than others and since there is no evidence to the contrary the differences that they observe between the TPL/DRM treated and DMSO treated proteomes bound to replicated DNA could just be because they were isolated from different genomic loci. I am also not convinced that they are able to stop EdU incorporation after 11min with the addition of only equimolar amounts of Thymidine (20 μ M EDU and 20 μ M Thymidine). Equimolar amounts of Thymidine are not sufficient to stop EdU incorporation rapidly. They need to show the kinetics of EdU incorporation in synchronized cells +/- Thymidine.

Without these controls it is impossible to draw any meaningful conclusions from the iPOND data.

2. The normalization of iPOND and total protein MS data is problematic. It seems that each time point from each treatment was first normalized internally to the median of all protein levels in each dataset and then the relative abundances of each protein were normalized to 100% over all treatments and time points. Internal normalization makes it impossible to directly compare time points and treatments between each other. If the enrichment of a protein goes down from one time point to the next it doesn't mean that there is less of that protein on replicated DNA in absolute terms, it just means that there is less of that protein relative to the median of the whole set of proteins at that time point. Their claim that they are comparing iPOND enrichments to total protein abundance is misleading since the data from total protein extracts was also internally normalized so they are comparing relative enrichments in iPOND data to relative enrichments in total cell extracts, which unsurprisingly do not correlate. It is impossible to make any meaningful conclusions about proteome dynamics using this kind of analysis. They should have used external normalization with a "spiked in" protein to be able to directly compare time points and treatments.

Such as it is right now, their analysis produces some puzzling conclusions that I suspect will turn out to be artefacts of their normalization procedure. It is not clear for example why the appearance of histones on replicated DNA would be delayed as they claim: in yeast nucleosomes (new and old recycled ones) are assembled on replicated DNA within minutes of the passage of the replication fork, I don't see why this would not be the case in human cells since the replication machinery is essentially the same in humans and yeast. It is also puzzling why RNAP2 is enriched in the nascent and 1hr time points but then becomes depleted in the 2hr time point in the DRB treatment since global RNAPII levels don't change in the DRB treatment compared to DMSO (Figure 1C). All the conclusions for PTM restoration/incorporation are essentially meaningless: internal normalization makes it impossible to detect whether PTM levels double at the 2hr time point compared to the Nascent time point in the DMSO treatment, as would be expected for all examined PTMs except for H4K5K12Kac which are marks of new histones. Right now, relative PTM levels are all over the

place: only histone acetylations seem to increase, while H3K9me3 and H3K27me3 don't change even though they should also double since heterochromatin should also be restored on both sister chromatids. They will only be able to accurately assess the impact of transcription inhibition on PTM restoration when they are able to reliably measure the rate of increase of PTM levels during chromatin maturation.

****Referees cross-commenting****

On reviewer's 2 comment on significance:

I think a thorough descriptive analysis of a biological process is extremely valuable and unlike my colleague, I think these types of studies need to be published in high impact journals with a broad readership. Biological processes need to be described first as completely as possible before we can propose meaningful models on how they function and identify the molecular mechanisms that execute and regulate them. As my colleague is surely aware, thorough descriptive studies of any poorly characterized biological process take years (i.e. at least one grant cycle) and comprehensive follow up mechanistic studies can take even longer than the initial descriptive study and can only be done during the following grant cycle, if the authors were lucky enough to obtain funding. Funding agencies however are more likely to award grants to perform these follow up mechanistic studies if the authors (especially if they are junior PIs) have published in higher impact journals in their previous grant cycle. The kind of thinking exhibited by reviewer 2 disproportionately disadvantages junior PIs that work on understudied biological processes. It is a disservice to scientific progress to dismiss excellent descriptive studies and "downgrade" them to lower impact journals where they will be unfairly labeled as a "work of lesser importance". This kind of thinking is also a disservice to the lower impact journals that often publish works whose quality is comparable to articles published in high impact journals. I value more any comprehensive description of a biological process over what most of the time passes for mechanistic insight that is deemed worthy of publication in a high impact journal i.e. a hastily analyzed phenotype of, more often than not, one single mutant tacked on at the end of a descriptive study. This one mutant phenotype then forms the basis of a somewhat "slapdash" model that is often proven wrong by subsequent publications and that the authors would have probably dismissed themselves had they been given more time to develop and test their model in a follow up publication.

I do not think the main issue with the present study is its descriptive nature. As I said in my review, the main issues are technical: the lack of external normalization of MS data and insufficient evidence of the impact of transcription inhibitors on replication dynamics. The study should not be published in any journal (high or low impact) before those issues are resolved.

on reviewer 2's remarque 4. in major comments:
iPOND identifies proteins bound to 100-300bp fragments labeled with EdU (i.e. after replication or DNA repair). It is by definition identifying proteins bound to chromatin behind the fork, so I don't think that the isolation of RNAPolII bound in from of the fork is a major issue

2. Significance:

Significance (Required)

I am not convinced by their conclusions and I cannot recommend that the the study be published at this stage due to normalization issues and insufficient evidence that transcription inhibition does not perturb the replication program (see above). They would need to redo all the iPOND experiments using external "spike in" normalization and monitor replication genome-wide before they can make any meaningful conclusions about the transcription dependent composition of the proteome associated with replicated DNA.

Expertise keywords: Chromatin, Genomics (assay development and bioinformatics analysis) , Replication, Transcription

3. How much time do you estimate the authors will need to complete the suggested revisions:

Estimated time to Complete Revisions (Required)

(Decision Recommendation)

More than 6 months

Yes

Review #2

1. Evidence, reproducibility and clarity:

Evidence, reproducibility and clarity (Required)

In this manuscript, the authors characterized the re-establishment of chromatin after DNA replication in fibroblasts using iPOND-MS. By using a short pulse of EdU, followed by different length of thymidine-chase, the authors compare the proteome at nascent DNA (just after the EdU pulse) with the proteome on re-established chromatin (1h and 2h post EdU pulse). Moreover, by using two different transcription inhibitors, they investigate the implication of active transcription elongation and of RNAPII binding itself on the reestablishment of chromatin. They show that different transcription factors bind to newly replicated DNA with different kinetics and are affected differentially by transcription inhibition. They also show that upon transcription inhibition by DRB, certain DNA damage repair proteins are depleted, implicating transcription in the recruitment of these factors at nascent DNA. Chromatin remodelers were shown to be enriched on nascent DNA, but triptolide-transcription inhibition reduced their enrichment, implicating RNAPII in the reestablishment of chromatin structure and of steady-state chromatin accessibility. Lastly, the authors show that histone incorporation and histone modification restoration on nascent DNA is mostly uncoupled from transcription with the exceptions of H3.3K36me2 (transcription inhibition by triptolide or DRB drastically reduces restoration) and H4K16ac (DRB treatment increases its incorporation in nascent DNA).

Overall, the results and the analysis of the datasets appear robust and well executed. Nevertheless, the work provided by the authors feels mainly descriptive and does not provide further mechanistic insights beyond the current state of the art. Some follow-up experiments to study the functional impact on the different enrichment patterns on nascent DNA or the function of the dependency on RNAPII for the reestablishment of steady-state enrichment on chromatin of some factors would have greatly increased the scientific impact of the manuscript. Nevertheless, the proteome of nascent DNA, its kinetic, and the effect of transcription inhibitors will provide interesting

information and a useful resource for research groups in the DNA replication, chromatin, epigenetic, and DNA damage repair fields. Thus, in conclusion, I would recommend this manuscript to be published in its current state in a lower tier journal such as MBoC or PLOS ONE journals. If the authors can provide additional mechanistic insights by addressing at least a few of the specific points listed below, I think it would become a stronger candidate for a journal with higher impact.

****Major comments:****

1. At p.7, the authors state: "Altogether, this analysis further confirms that RNAPII's binding and elongation on newly replicated chromatin are a source of genotoxic stress, and identifies dedicated repair factors handling transcription replication conflicts." I don't think that depleted DNA repair proteins from nascent chromatin upon DRB treatment is enough to claim that the analysis confirms that transcription on nascent DNA is a source of stress. Another possibility could be that transcription helps handling prior DNA damage on nascent DNA without causing the damage. A useful experiment to clarify this point would be the direct quantification of DNA damage markers on nascent chromatin (e.g γ H2AX-EdU colocalization quantification by immunofluorescence). Has the γ H2AX variant been detected in the iPOND MS dataset? Another possible follow-up experiment could be to detect direct physical DNA damage on nascent DNA for example by using a TUNEL assay or similar DSB mapping method. Can the DNA damage be prevented by DRB or TRP addition?
2. Figure 1B-E: Can the authors also show quantifications of EU, RNAPII and EdU at the 1h and 2h timepoints after the chase?
3. The authors state in p.7 that "The other proteins are DNA repair proteins involved in fork quality control and HR as well as transcription replication conflicts (Berti et al., 2020)". I think this gives rise to the question if the effect of DRB treatments on the enrichment of certain proteins at nascent DNA is due to the inhibition of transcription elongation inhibition on nascent DNA or in front of replication forks, affecting the enrichment of proteins implicated in handling transcription-replication conflicts in front of the fork and not on nascent DNA itself. The authors should address the possibility that some of the proteins enriched in the iPOND-MS datasets could be there because they are enriched in front of the replisome instead of on the nascent DNA.
4. On this topic, transcription inhibition is performed for two hours prior to the EdU pulse and iPOND-MS procedures. For the DRB treatment, I would expect RNAPII to be paused/stalled prior to the passage of the replication fork that will replicate the analyzed EdU-labelled nascent DNA. This would mean that replication forks during the EdU pulse will encounter paused/stalled RNAPII, generating potential problems. Such interference would most probably lead to chromatin removal of RNAPII from the chromatin. Surprisingly, the authors show enrichment of RNAPII at nascent DNA.

How can the author differentiate from accumulation of RNAPII in front of the fork, leading to purification by iPOND, and RNAPII on nascent DNA. Also, if the accumulation of RNAPII is on the nascent DNA, do the authors suggest that RNAPII gets loaded more on nascent DNA while under DRB inhibition or that stalled RNAPII are mainly by-passed by replication forks, leading to their enrichment on nascent DNA?

5. At p.14, the authors state: "Because they share the same DNA template, transcription is known to challenge replisome progression at high frequency, from RNAPII constituting a roadblock to progressing replisomes, to generate RNA:DNA hybrids (Berti et al., 2020). It is therefore remarkable that behind replisomes, only a handful of DNA repair factors appear to be involved in response to RNAPII binding and elongation.". How does the fact that transcription represents a roadblock in front of the forks makes it remarkable that only a handful of DNA damage repair pathways are involved behind the fork (where they are not a roadblock to any replisome)?

6. At p.11, the authors states: "As di and tri-methylations require several hours to be re-established following DNA replication (Alabert et al., 2015; Reveron-Gomez et al., 2018), 11 minutes after the passage of the fork, such increase most probably reflects an increase of H4K20me2 and H3K9me3 on recycled parental histones.". Can the authors extend their interpretation of this result? Do the authors think that DRB treatments increase methylation of histones in G1, prior to replication, or specifically in front of the fork (due to conflicts? DNA damage?), and that those methylated histones get recycled on nascent DNA?

7. Figure 4: The authors perform the histone PTM analysis under 0h (nascent chromatin) versus 2h (re-established chromatin) timepoints. It would have been insightful to also include a 1h timepoint in this experiment. There appear to be some trends/changes but they do not show statistical significance (e.g. H4K5K12ac or H3K14ac). It might be useful to increase the number of biological replicates (including the 1h timepoint) here, which could improve the confidence in the results and/or discover additional transcription-dependent changes of histone PTM restoration.

****Minor comments:****

1. Fig3I: It would be nice to show a TF from the "Restored within 11 min" category as a comparison point.

2. In page 14, the authors state: "However, we did not detect significant signs of DNA damage in DRB treated cells (Fig. 2A, 2B)". Which signs the authors looked at?

3. In the iPOND experiment, which size of DNA fragments is achieved?

4. At p.14, the authors state: "Because they share the same DNA template, transcription is known to challenge replisome progression at high frequency, from RNAPII constituting a roadblock to progressing replisomes, to generate RNA:DNA

hybrids (Berti et al., 2020)." The paper has not addressed the role of RNA:DNA hybrids in these processes.

5. Fig3D: Is there enough datapoints to state a conclusion?

6. S1A: mistakes in the x axis labels ("no EU" in a EdU quantification graph, "no EdU" in a EU quantification graph).

7. S1F is not sufficiently described in the legend. It took me some time and additional efforts to understand what the right panel of S1F was showing.

8. S2E-F: are the axis wrong? Is it supposed to be Nascent when its comparing total extracts?

9. A lot of graphs have non-precise axis labels that needs reading of the manuscript and/or legends to understand. For example: 1K-L (distribution, %), 2L (% of the max), 3B-C-D $\log_2(\text{Nascent}/2\text{h})$, 3G IP/Input, 4C (Inhibitor treatment/DMSO (%)), S2E-F (TPL/DRB Nascent/ DMSO Nascent), S3A (IP/Input), S4A (No Y axis label).

10. FigS4: Assignment of colors in bar graphs of C-J to treatments is not shown. Heatmaps in H and K do not show if these are 0 or 2h. The heatmap in H shows H3 modification and the heatmap in K shows modification in H3.3 but the labels of the modification in K (except the first one) are the names of the modifications of H3, not H3.3. In the legend, GAPDH is written GABDH.

2. Significance:

Significance (Required)

In this manuscript, the authors characterized the re-establishment of chromatin after DNA replication in fibroblasts using iPOND-MS. As mentioned above, the work provided by the authors feels mainly descriptive and incremental and therefore does not provide further mechanistic insights beyond the current state of the art. Some follow-up experiments to study the functional impact on the different enrichment patterns on nascent DNA or the function of the dependency on RNAPII for the reestablishment of steady-state enrichment on chromatin of some factors would have greatly increased the scientific impact of the manuscript. Nevertheless, the proteome of nascent DNA, its kinetic, and the effect of transcription inhibitors will provide interesting information and a useful resource for research groups in the DNA replication, chromatin, epigenetic, and DNA damage repair fields. Thus, in conclusion, I would recommend this manuscript to be published in its current state in a lower tier journal such as MBoC or PLOS ONE journals. If the authors can provide additional mechanistic insights by addressing at least a few of the specific points, I think it would become a stronger candidate for a journal with higher impact.

3. How much time do you estimate the authors will need to complete the suggested revisions:

Estimated time to Complete Revisions (Required)

(Decision Recommendation)

Between 3 and 6 months

Yes

Review #3

1. Evidence, reproducibility and clarity:

Evidence, reproducibility and clarity (Required)

In this study, the authors used metabolic labeling of newly-replicated or nascent chromatin followed by quantitative Mass spectrometry (iPOND-MS) to characterize protein composition of nascent chromatin at time points after DNA replication: immediately after a short pulse of EdU labeling (nascent), and after 1 and 2 hours of Thymidine chase (maturing chromatin). The iPOND method was established before but in the current manuscript the authors combined this with inhibiting RNA Pol II transcription at distinct stages to determine the effects on transcription and RNA Pol II cycle on chromatin protein dynamics at the wake of DNA replication. The inhibitors they used are Triptolide, which blocks transcription initiation and induces a proteasomal degradation of all chromatin bound RNA PolIII, and DRB, which blocks transcription elongation causing an enrichment of paused RNA PolIII. The authors compared the relative enrichment of ~1200 proteins on nascent and maturing chromatin and the effects on transcription inhibition on these proteins.

The authors found that RNA PolIII does not affect the loading or retention of most histones on nascent chromatin except for the histone variant H2A.Z, which requires RNA PolIII loading. However, DRB treatment (no elongation) resulted in stabilization of all histones (which the authors do not seem to catch on). Interesting, unlike the histone, both replication-coupled and -independent histone chaperons seem to be enriched immediately behind the fork and are affected by RNA PolIII to different extents. They next look at ATP-dependent remodelers and find that most remodeler families are facilitated by RNA PolIII loading, while elongation affects some remodeler families and not others. They see the same trend looking at a wide variety of transcription factors. Interestingly, while RNA PolIII loading is required for the establishment of some histone post translational modifications (H3K36me3), some others such as H3K9me3 and H4K20me2 are negatively affected. Finally, the authors find that RNA PolIII elongation promotes binding of several DNA repair proteins, and speculate that this is because of DNA damage from replication-transcription conflict.

My main concern about this manuscript is that the relative enrichment of most factors show variability across the time points, which make the interpretation of the data difficult. This becomes more concerning when we look at protein complexes such as the ATP-dependent remodelers. Subunits of the same complex which are expected to bind together show different patterns of enrichment. This raises the concern as to how data was normalized. Furthermore, how do the replicates compare to each other? The others selected ~1200 proteins which were enriched in all three replicates, but how does their relative enrichment compare in the replicates? The authors need to show some kind of comparison across replicates to confirm that the differential relative enrichments are real and biologically meaningful.

Also, the TF data is very descriptive. Insightful analysis of similarities/differences between types of TFs would be interesting.

****Minor comment:**** The formaldehyde cross linking used in iPOND makes it difficult to interpret/distinguish what is actually chromatin bound versus what is enriched due to protein-protein cross linking. The authors should highlight that in the limitations section.

****Referees cross-commenting****

I agree with most of Reviewer 1's comments about the lack of proper controls and normalization, which make the interpretations difficult. Particularly all of the controls mentioned under point 1 should not be difficult to perform, and if included, would strengthen the study and the manuscript.

Reviewer 1 makes an important point about normalization, which I totally agree with. Ideally, a spike-in approach would help obtain a much more quantitative and reliable understanding of differential protein enrichment. However, repeating all iPOND experiments with spike-in might be a big ask. What the authors could do at minimum is show how replicates compare with each other. It looks like they pooled all three replicates for analysis, but comparing relative enrichment of all 1257 proteins across replicates would help. The point about delayed histone occupancy is a critical one and difficult to rationalize. To note, histone chaperons are enriched on nascent, but histones are not. Besides, in the current way that the data is analyzed and presented, there are a lot of fluctuations in protein enrichment across the 1-2 hour timepoints of chromatin maturation, which would be very interesting if real. For e.g., Fig. 1I, Triptolide treatment, most of the cluster I and cluster II proteins show medium-high enrichment on nascent, depleted in 1h, but recover in 2h. If the binding/recruitment of these proteins on newly-replicated chromatin is RNA Pol II dependent, why would they come back after 1h? If this real, this would be very interesting. There are several additional examples of problems with quantification/normalization. As for SWI/SNF subunits, both SMARCA4 and SMARCC1 are core subunits and based on several thorough biochemical studies, cannot be expected to bind separately. However, they show different kinetics in DMSO as well as TPL and DRB.

Another problem of the assay is that it shows genome-wide average. As Reviewer 1 rightly pointed out, transcription inhibition could disproportionately affect chromatin maturation kinetics in different genomic regions. Perhaps it would be interesting to analyze sets of genomic regions separately, such as highly transcribed and lowly transcribed genes. This might be achieved by adding a purification step using pools of DNA sequence probes before or after the streptavidin enrichment.

Additional comment: The formaldehyde cross linking used in iPOND makes it difficult to interpret/distinguish what is actually chromatin bound versus what is enriched due to protein-protein cross linking. The authors should highlight that in the limitations section.

On a positive note, it is a very important and timely study and the manuscript has a lot to consider. Addition of proper controls and normalization/analysis of replicates will make it stronger

2. Significance:

Significance (Required)

Overall, it is a very important and timely study, and the manuscript has a lot to consider. There are several recent papers on the kinetics of chromatin maturation behind the replication fork, and this study adds a very important dimension to this ongoing investigation, and will be of interest to a broad readership in the chromatin and transcription field.

3. How much time do you estimate the authors will need to complete the suggested revisions:

Estimated time to Complete Revisions (Required)

(Decision Recommendation)

Cannot tell / Not applicable

Yes

Dundee, 21.07.2023

Dear Esther,

I am happy to provide you with our manuscript “Transcription restart promotes and challenges the re-establishment of post-replicated chromatin” by Bandau et al. This is the work I have presented in Heidelberg, and you kindly mentioned that EMBO reports would potentially be interested. We have received 3 reviewer’s comments through Review Commons, and we are pleased to send you our revision plan.

In this study we have combined our expertise in chromatin biology and mass spectrometry to tackle one of the key questions in epigenetics: the mechanisms underlying inheritance of chromatin organization. Using quantitative MS-based proteomics coupled to iPOND, we provide the first comprehensive picture of how transcription restart contributes to chromatin restoration following DNA replication.

We are happy that all reviewers acknowledge the broad interest of our work. The strongest critic comes from Reviewer 1, asking for missing controls (that we provide below), and criticising the normalisation method we used (that we discuss in detail below). Reviewers 2 and 3 ask for additional biological replicates to increase the robustness of our findings, and more mechanistic experiments, such as distinguishing the effect of blocking transcription on DNA replication ahead and behind replisomes. We propose to clarify these important points by performing experiments within the time frame of a 3-month revision. Therefore, please find below our revision plan and a point-by-point response to all reviewer’s comments. We hope you will find them appropriate. If anything is unclear, please let us know.

I am looking forward to hearing from you.

Yours sincerely,

Constance

Constance Alabert
Principal investigator
School of Life Sciences (SLS), University of Dundee
Phone: +447578848288 email: calabert@dundee.ac.uk

I. REVISION PLAN

*Comments from reviewers are labelled as followed: **R1.1.1**: Reviewer 1, Comment 1, Section 1. Sections were created to address the multiple points raised by reviewers within one comment.*

Description of the planned revisions

- 1) To validate the key findings of this study: use an orthogonal approach, PLA-EdU (R1.2.4)
- 2) To strengthen the iPOND-MS conclusions: increase the number of biological replicates (R2.7, R3.2)
- 3) To distinguish the effect of TPL and DRB ahead and behind replisomes: iPOND-MS in chase experiments (24h, as in Alabert et al. 2015) (R1.2.6, R2.1)
- 4) To measure the proportion of DNA replication independent EdU signal: Quantification of PCNA and EdU foci colocalization (+/- transcription inhibitor) (R1.1.3)
- 5) Test FRAP based prediction: measure RNAPII level on chromatin after short DRB treatment by QIBC (R2.4.1)

Existing Data and information we would like to include in the revised version of the manuscript.

- 1) Effect of DMSO: Analysis of EdU incorporation, cell cycle distribution and DNA damage in +/- DMSO (R1.1.2)
- 2) EdU chase conditions: pulse chase quantification by flow cytometry (R1.1.5)
- 3) Effect of TPL/DRB on DNA repair: Quantification of γ H2AX levels in EdU positive and negative cells (R1.1.3)
- 4) Effect of TPL/DRB on the replication program: Analysis of S phase progression, cell cycle- and S phase-distribution (QIBC) (R1.1.4)
- 5) DNA damage on newly replicated chromatin: PLA experiment between EdU and γ H2AX +/- transcription inhibition as well as the quantification of the H2AFX peptide from iPOND-MS (n=3) (R2.1).
- 6) DNA damage on chromatin: Quantification of γ H2AX +/- transcription inhibition on total chromatin via QIBC and Western Blot (R1.1.2, R2.1)
- 7) Assessment of the reproducibility of the dataset: Clustering and enrichment of selected proteins in each individual biological replicate (R3.2).
- 8) Expand the limitation section (R3.1 and 3.4)

Description of analyses that authors prefer not to carry out.

At this point, we do not consider repeating all the iPOND-MS experiments using a spike-in for external normalisation due to the reasons presented in R1.2.1 (p6).

II. POINT BY POINT RESPONSE TO REVIEWER COMMENTS

Reviewer #1 (Evidence, reproducibility and clarity (Required)):

The goal was to characterize the changes in the composition of the proteome associated with replicated DNA in conditions of genome-wide inhibition of transcription initiation or transcription elongation. They use iPOND, a MS based technique that identifies proteins specifically associated with replicated DNA labeled with EdU. They use non-synchronized foetal human lung fibroblasts and examine time points immediately after replication (after the 11 min EdU pulse) and 1 and 2 hrs after the Thymidine "chase" later when chromatin has "matured", to assess how the inhibition of transcription influences chromatin maturation and the binding of chromatin associated proteins to replicated DNA. The question is pertinent and is in line with the long-standing interest of the group in chromatin replication dynamics. They conclude that 1. RNAPII loading is necessary for the binding of some TFs, chromatin remodelers and DNA repair factors and 2. RNAPII elongation is needed for H2A.Z incorporation, H3K36me2 restoration and DNA repair factor binding. Transcription is on the other hand not needed for nucleosome assembly, histone acetylation and H3K9me3, H3K27me3 and H4K20me2 incorporation or restoration.

There are two main issues that make the interpretation of their results very difficult and make me question their conclusions:

R1.1.1 They don't provide sufficient evidence that the treatments with TPL and DRB do not interfere with replication. The distributions of EdU intensity per EDU+ cell after treatment in Figures 1D-E and S1A are not sufficient. It is not clear why EdU incorporation is so heterogeneous in the cell population (the range of intensities goes from near 0 to 50000!), which makes me wonder if the DMSO treatment also has an effect on replication. I don't think this heterogeneity can simply be explained by the fact that the cell population is asynchronous.

In Figure 1D-E, a linear scale is shown instead of a logarithmic scale to facilitate comparison of EdU incorporation upon DMSO, TPL and DRB treatments. As shown below, the range of EdU intensities is in good accordance with previous publications (see examples from Massimo Lopes and Luis Toledo labs).

Range of EdU intensities. Cells were labelled with EdU and imaged by Quantitative Image-based Cytometry (QIBC). EdU intensities (a.u., arbitrary units) are shown as a function of total DAPI intensities. Each dot represents a nucleus identified based on DAPI intensities. This study, data from Fig. 1D, EdU intensities in S phase cells range from 800 to 50,000. Berti et al. 2020, from 6 to 600. Zonderland et al. 2022, from 30 to 4000.

R1.1.2 They need to show a -DMSO control as well.

A standard concentration of DMSO was used (0.06%). When comparing DMSO and no DMSO conditions, no defects in EdU incorporation, changes in cell cycle distribution or increases in DNA damage were observed (see panels below). These controls will be added in the revised version of the manuscript.

Comparison of DMSO and no DMSO conditions. Left, EdU incorporation. Cells were labelled with EdU for 20 min and imaged by QIBC. Each dot represents a single cell. Middle, cell cycle distribution. EdU and DAPI were used to gate G1, S and G2 phase cells. Right, quantification of γ -H2AX fluorescence signals in individual cells. Each dot represents a single cell. Median values are shown in red. Cells were treated 3h with Camptothecin (Campto, 1 μ M) as a positive control for DNA damage. >1464 nuclei were analysed per condition.

R1.1.3 Besides since they are only using a positive EdU signal as their criteria for replicating cells, they cannot rule out that some of the EdU signal is coming from DNA repair after replication and depending on how deleterious DMSO/TPL/DRB are to replication the fraction of cells that undergo DNA repair might be significant.

The drug treatments used in this study, DMSO, TPL and DRB, do not increase γ -H2AX signal (see right panel above). γ H2AX levels have also been measured in EdU positive and negative cells (see panels below). It seems therefore unlikely that a significant proportion of EdU signal is coming from DNA repair. These controls will be included in the revised version of the manuscript. Moreover, we propose to quantify by microscopy the percentage of EdU foci colocalising with chromatin bound PCNA in +/- transcription inhibitor conditions, to quantify the proportion of DNA replication independent EdU signal.

Quantification of γ H2AX signal in EdU positive and negative cells. Quantification of γ -H2AX fluorescence signals in individual cells measured by QIBC. Cells were treated with TPL and DRB according to conditions used in the study, and labelled with EdU for the last 20 min. Camptothecin treatment is used as a positive control for DNA damage. >600 nuclei were analysed per condition.

R1.1.4 More importantly, they need to show that the various treatments don't interfere with the replication program, especially since replication is coupled with new nucleosome assembly

and the transcription of replication dependent histone variants is induced during S-phase. Transcription inhibition could disproportionately affect the replication of some parts of the genome more than others and since there is no evidence to the contrary the differences that they observe between the TPL/DRM treated and DMSO treated proteomes bound to replicated DNA could just be because they were isolated from different genomic loci.

We have carefully examined the impact of TPL and DRB treatments on the replication program using multiple approaches. First, inhibitor treatments minimally affects DNA replication efficiency (Fig. 1D-E) and do not alter S phase progression (see panel below), which is in good accordance with previous published work (Stewart-Morgan et al. 2020).

Analysis of S phase distribution upon DMSO, TPL and DRB treatment. Flow cytometry plots of asynchronous cells treated with DMSO, TPL and DRB. Cells were EdU pulsed and chased according to the conditions defined in the manuscript. Nas, Nascent sample; 2h, Mature sample.

Second, we have inspected replication foci patterns (Dimitrova & Gilbert 1999) and quantified the distribution of the cell population between early/mid and late S phase cells. We did not observe any abnormal patterns, or changes in patterns distribution upon transcription inhibition as observed after DNA damage (Seiler et al. 2007) or upon RIF1 depletion (Yamazaki et al. 2012) (see panel below). Altogether these data support that the replication program is unaffected 2h after transcription inhibition, and to address the point raised by Reviewer 1, we propose to include these controls in the revised version of the manuscript.

Cell cycle and S phase distribution. DMSO, TPL and DRB treated cells are pulsed with EdU for 20 min and analysed by microscopy. A) Representative images of Early/Mid and Late S phase patterns. Early and Mid-S phase cells have a pattern of replication foci distributed throughout the nucleus. Late-S-phase cells would have a small number of large foci within the nucleus. B) Distribution of cells in G1 and G2 phase based on EdU and DAPI intensities. Distribution of cells in Early/Mid (top section) and late S phase (bottom section) based on the distinct EdU patterns shown in A. >1230 nuclei were analysed per condition.

R1.1.5 I am also not convinced that they are able to stop EdU incorporation after 11min with the addition of only equimolar amounts of Thymidine (20µM EDU and 20µM Thymidine). Equimolar amounts of Thymidine are not sufficient to stop EdU incorporation rapidly. They need to show the kinetics of EdU incorporation in synchronized cells +/- Thymidine.

EdU incorporation has been monitored after thymidine chase in TIG-3 cells and showed that the chase is stopping EdU incorporation (see panel below). This new panel will be added in the revised version of the manuscript to address Reviewer 1's concern.

EdU pulse chase quantification by flow cytometry. Cells were pulsed for 11 min with EdU (20 μ M) (Nascent sample) and chased 2h by 20 μ M of thymidine (Mature sample).

R1.1.6 Without these controls it is impossible to draw any meaningful conclusions from the iPOND data.

We hope that the controls provided above will satisfy Reviewer 1 concerns regarding the experimental conditions of this study.

R1.2.1 The normalization of iPOND and total protein MS data is problematic. It seems that each time point from each treatment was first normalized internally to the median of all protein levels in each dataset and then the relative abundances of each protein were normalized to 100% over all treatments and time points. Internal normalization makes it impossible to directly compare time points and treatments between each other. If the enrichment of a protein goes down from one time point to the next it doesn't mean that there is less of that protein on replicated DNA in absolute terms, it just means that there is less of that protein relative to the median of the whole set of proteins at that time point.

Reviewer 1 is raising an important point. There is no consensus in the field on how iPOND-MS and NCC-MS should be normalised. As a result, four methods have been used: 1) No normalisation (**NN**) (Lopez-Contreras et al. 2013); 2) Normalisation based on histone proteins (**NH**) (Alabert et al. 2014); 3) Each sample is median normalised (**NS**) based on the assumption that most of the proteins in the compared samples have similar abundances and only a relatively small percentage of proteins are differentially abundant (Alvarez et al. 2023); 4) The ratio between two conditions are median normalised (not assuming that proteins in compared samples have similar abundance) (**NR**) (Nakamura et al. 2021); We have tested all four methods on our dataset and selected NS for the reasons listed below:

1. When examining the abundance of the proteins identified, variations between the different batches of TMT (known as the batch effect) seems minor (Panel A-B below).
2. Variations between time points within a batch can on the other hand be observed. This can be attributed to sample preparation variation or to the fact that proteins in different samples have different abundance. In our experience, sample preparation variation is substantial in iPOND, using proteomics or western blot as a readout. We have tested multiple conditions over the years (different phases of the cell cycle, different drug treatments), and the main source of variation arises from variable efficiency of the click-

It reaction and subsequent pulldown. As a result, **NS** seems an appropriate normalisation method (Panel C-D).

4. NS is based on the assumption that most of the proteins in the compared samples have similar abundances and only a relatively small percentage of proteins are differentially abundant. As pointed by Reviewer 1, NS potentially erases biological differences between samples, and to preserve these differences, **NR** should be used instead. Yet, in our datasets such differences have never been observed across biological replicates, whether we use replication stress agents (HU, CPT, ATMi), translation inhibitor (CHX), or here transcription inhibitors (TPL, DRB, Panel A-B). Consistent with this, the overlap between the list of proteins selected following NS or NR is very high (Panel E), further supporting the validity of the assumption and that NS is an appropriate normalisation method.
5. Another possible normalisation method is to use histones as protein rulers (NH) (Alabert et al. 2014). This method was successful for NCC-MS (Alabert et al. 2014) but proven to be problematic for iPOND-MS. In NCC (20 min biotin-dUTP labelling), the amount of histone detected on nascent and mature samples moderately changes. In iPOND (11min EdU labelling) it greatly increases (Fig. 1H). Consequently, normalising the iPOND-MS dataset using histones shifted the proteins toward nascent time points (Panel F). This method was therefore considered not appropriate for this dataset.
6. Others have also used no normalisation approach (NN). In that case, whether a protein is a hit or not relies on cut-off and/or the protein been identified as a hit in 2/3 or 5/6 experiments (Lopez-Contreras et al. 2013). We have run such analysis on our data and although it reduces the number of hits, several key findings remain (Panel G).
7. Finally, Reviewer 1 is suggesting an external normalisation method using a “spike in” protein. EdU-labelled DNA from *Drosophila* S2 cells has been spiked in samples prior to the click-iT reaction and used to normalize read counts in iPOND sequencing experiments (Reveron et al. 2018). However, normalising iPOND mass spectrometry dataset using a ‘spike in’ protein, to allow to correct for sample preparation variation, requires a synthetic peptide or a protein clearly distinguishable from the human homologue and that binds EdU labelled DNA. To our knowledge, there is no known protein that could be used for this purpose, including histones due to the high degree of histone sequence conservation.

In conclusion, we have tested the four published methods to handle iPOND-MS on our dataset, and as most of our conclusions stands, we do not think the conclusions of this study are normalisation artefacts.

R1.2.2 Their claim that they are comparing iPOND enrichments to total protein abundance is misleading since the data from total protein extracts was also internally normalized so they are comparing relative enrichments in iPOND data to relative enrichments in total cell extracts, which unsurprisingly do not correlate.

In total extracts, 2 hr after TPL or DRB treatments, most proteins are expected to have similar abundances, while only a small percentage of proteins are differentially abundant (Skalska et al. 2021). As for R1.2.1, analysis of the raw intensities further supports that NS is an appropriate normalisation method (see panel A below). Consistent with this, the correlation between RNAPII change in abundance in total extract and iPOND-MS is conserved following NN, NR and NS (see panels B-D). Of note, this comparison is important to identify changes on replicated chromatin that may be mirrored by changes in total extracts.

A

Comparison of newly replicated chromatin and total extracts. A. Box plots showing the protein intensities prior to normalisation for 3 conditions, 3 biological replicates (9 in total). Whiskers show min to max. **B.** ratio of DMSO and TPL intensities, log2 transformed (NN). X axis, iPOND-MS; Y axis, total extract-MS. The mean of three experiments is shown. **C.** Same as B but following NR. **D.** Same as A but following NS.

B

C

D

R1.2.3 It is impossible to make any meaningful conclusions about proteome dynamics using this kind of analysis. They should have used external normalization with a "spiked in" protein to be able to directly compare time points and treatments.

Please see R1.2.1, point 7 above.

R1.2.4 Such as it is right now, their analysis produces some puzzling conclusions that I suspect will turn out to be artefacts of their normalization procedure.

We have tested the no normalization method and the three other normalization strategies on our dataset, and as most of the conclusions stands, we do not think the conclusions of this study are normalisation artefacts. We propose to state in the result section that various normalization strategies has been compared and why NS was chosen. In the discussion we propose to further expand on the importance of normalisation choice. Moreover, we propose

to validate key findings using an orthogonal approach, PLA-Edu (Serebryanny and Misteli, 2019, Fenstermaker et al. 2023), as previously done in Alvarez et al. 2023, and for the TF ZNF462 in this manuscript (Fig. 3I).

R1.2.5 It is not clear for example why the appearance of histones on replicated DNA would be delayed as they claim: in yeast nucleosomes (new and old recycled ones) are assembled on replicated DNA within minutes of the passage of the replication fork, I don't see why this would not be the case in human cells since the replication machinery is essentially the same in humans and yeast.

Yeast and human are indeed very similar with only ~300 bp of DNA nucleosome free behind each replisome in average. Consequently, even if nucleosome assembly is rapid, when pulling down nascent chromatin, we do expect to pulldown less histones than in mature samples. We, and others (Lopez-Contreras et al. 2013), have detected a similar drop in histone abundance on nascent chromatin compared to mature samples by iPOND-MS, and iPOND-western blot, whether the same amount of protein or number of cells is loaded per well (Fig. 1H).

R1.2.6 It is also puzzling why RNAP2 is enriched in the nascent and 1hr time points but then becomes depleted in the 2hr time point in the DRB treatment since global RNAPII levels don't change in the DRB treatment compared to DMSO (Figure 1C).

This is either an interesting biological difference between chromatin and newly replicated chromatin, or due to the different sensitivity of the two technologies used (High content microscopy for chromatin, iPOND-MS for replicated chromatin). To monitor the effect of transcription inhibition on chromatin using iPOND-MS, we propose to perform an iPOND-MS on cells EdU pulse chased for 24h, as previously done in Alabert et al. 2015 (see below and point R2.1).

Experimental design to compare newly replicated chromatin and chromatin by iPOND-MS. **A.** Experimental design of existing dataset. **B.** Experimental design of the new set of experiment proposed ($n=3$). See details in the text above.

R1.2.7 All the conclusions for PTM restoration/incorporation are essentially meaningless: internal normalization makes it impossible to detect whether PTM levels double at the 2hr time point compared to the Nascent time point in the DMSO treatment, as would be expected for all examined PTMs except for H4K5K12Kac which are marks of new histones.

The histone PTMs data are not internally normalized. For each biological replicate and each time point, for any given peptide identified, the proportion of each modified state of the peptide is calculated as in Alabert et al 2015. For instance, for the peptide histone H4 aa20-24 carrying K20, the proportion of peptides with K20me0, K20me1, K20me2 and K20me3 are calculated.

Moreover, in human, histone methylations are not expected to double within 2h, requiring instead 6h to 20h, depending on the histone residue (Alabert et al. 2015, Reveron et al. 2018). Histone modifications expected to decrease within 2h are H4K5K12ac and are behaving accordingly in this dataset (Fig. 4D). This dataset is therefore in good accordance with published MS-analysis of histone modifications.

R1.2.8 Right now, relative PTM levels are all over the place: only histone acetylations seem to increase, while H3K9me3 and H3K27me3 don't change even though they should also double since heterochromatin should also be restored on both sister chromatids.

H3K9me3 and H3K27me3 are not expected to double within 2h in human cells (see point above).

R1.2.9 They will only be able to accurately assess the impact of transcription inhibition on PTM restoration when they are able to reliably measure the rate of increase of PTM levels during chromatin maturation.

This is what is shown for the first 2h after the passage of the fork. Longer time points (12 and 20h for instance) would be required to complete the histone methylation restoration analysis. However, such experiments will be affected by the known effect of prolonged transcription inhibitors treatments on genome stability (Skalska et al. 2021).

****Referees cross-commenting****

On reviewer's 2 comment on significance:

I think a thorough descriptive analysis of a biological process is extremely valuable and unlike my colleague, I think these types of studies need to be published in high impact journals with a broad readership. Biological processes need to be described first as completely as possible before we can propose meaningful models on how they function and identify the molecular mechanisms that execute and regulate them. As my colleague is surely aware, thorough descriptive studies of any poorly characterized biological process take years (i.e. at least one grant cycle) and comprehensive follow up mechanistic studies can take even longer than the initial descriptive study and can only be done during the following grant cycle, if the authors were lucky enough to obtain funding. Funding agencies however are more likely to award grants to perform these follow up mechanistic studies if the authors (especially if they are junior PIs) have published in higher impact journals in their previous grant cycle. The kind of thinking exhibited by reviewer 2 disproportionately disadvantages junior PIs that work on understudied biological processes. It is a disservice to scientific progress to dismiss excellent descriptive studies and "downgrade" them to lower impact journals where they will be unfairly labeled as a "work of lesser importance". This kind of thinking is also a disservice to the lower impact journals that often publish works whose quality is comparable to articles published in high impact journals. I value more any comprehensive description of a biological process over what most of the time passes for mechanistic insight that is deemed worthy of publication in a high impact journal i.e. a hastily analyzed phenotype of, more often than not, one single mutant tacked on at the end of a descriptive study. This one mutant phenotype then forms the basis of a somewhat "slapdash" model that is often proven wrong by subsequent publications and that the authors would have probably dismissed themselves had they been given more time to develop and test their model in a follow up publication.

I do not think the main issue with the present study is its descriptive nature. As I said in my review, the main issues are technical: the lack of external normalization of MS data and insufficient evidence of the impact of transcription inhibitors on replication dynamics. The study

should not be published in any journal (high or low impact) before those issues are resolved.

on reviewer 2's remarque 4. in major comments:

iPOND identifies proteins bound to 100-300bp fragments labeled with EdU (i.e. after replication or DNA repair). It is by definition identifying proteins bound to chromatin behind the fork, so I don't think that the isolation of RNAPoIII bound in front of the fork is a major issue

Reviewer #1 (Significance (Required)):

I am not convinced by their conclusions and I cannot recommend that the study be published at this stage due to normalization issues and insufficient evidence that transcription inhibition does not perturb the replication program (see above). They would need to redo all the iPOND experiments using external "spike in" normalization and monitor replication genome-wide before they can make any meaningful conclusions about the transcription dependent composition of the proteome associated with replicated DNA.

Expertise keywords: Chromatin, Genomics (assay development and bioinformatics analysis) , Replication, Transcription

Reviewer #2 (Evidence, reproducibility and clarity (Required)):

In this manuscript, the authors characterized the re-establishment of chromatin after DNA replication in fibroblasts using iPOND-MS. By using a short pulse of EdU, followed by different length of thymidine-chase, the authors compare the proteome at nascent DNA (just after the EdU pulse) with the proteome on re-established chromatin (1h and 2h post EdU pulse). Moreover, by using two different transcription inhibitors, they investigate the implication of active transcription elongation and of RNAPII binding itself on the reestablishment of chromatin. They show that different transcription factors bind to newly replicated DNA with different kinetics and are affected differentially by transcription inhibition. They also show that upon transcription inhibition by DRB, certain DNA damage repair proteins are depleted, implicating transcription in the recruitment of these factors at nascent DNA. Chromatin remodelers were shown to be enriched on nascent DNA, but triptolide-transcription inhibition reduced their enrichment, implicating RNAPII in the reestablishment of chromatin structure and of steady-state chromatin accessibility. Lastly, the authors show that histone incorporation and histone modification restoration on nascent DNA is mostly uncoupled from transcription with the exceptions of H3.3K36me2 (transcription inhibition by triptolide or DRB drastically reduces restoration) and H4K16ac (DRB treatment increases its incorporation in nascent DNA). Overall, the results and the analysis of the datasets appear robust and well executed. Nevertheless, the work provided by the authors feels mainly descriptive and does not provide further mechanistic insights beyond the current state of the art. Some follow-up experiments to study the functional impact on the different enrichment patterns on nascent DNA or the function of the dependency on RNAPII for the reestablishment of steady-state enrichment on chromatin of some factors would have greatly increased the scientific impact of the manuscript. Nevertheless, the proteome of nascent DNA, its kinetic, and the effect of transcription inhibitors will provide interesting information and a useful resource for research groups in the DNA replication, chromatin, epigenetic, and DNA damage repair fields. Thus, in conclusion, I would recommend this manuscript to be published in its current state in a lower tier journal such as MBoC or PLOS ONE journals. If the authors can provide additional mechanistic insights by

addressing at least a few of the specific points listed below, I think it would become a stronger candidate for a journal with higher impact.

Major comments:

R2.1. At p.7, the authors state: "Altogether, this analysis further confirms that RNAPII's binding and elongation on newly replicated chromatin are a source of genotoxic stress, and identifies dedicated repair factors handling transcription replication conflicts.". I don't think that depleted DNA repair proteins from nascent chromatin upon DRB treatment is enough to claim that the analysis confirms that transcription on nascent DNA is a source of stress. Another possibility could be that transcription helps handling prior DNA damage on nascent DNA without causing the damage. A useful experiment to clarify this point would be the direct quantification of DNA damage markers on nascent chromatin (e.g γ H2AX-EdU colocalization quantification by immunofluorescence). Has the γ H2AX variant been detected in the iPOND MS dataset? Another possible follow-up experiment could be to detect direct physical DNA damage on nascent DNA for example by using a TUNEL assay or similar DSB mapping method. Can the DNA damage be prevented by DRB or TRP addition?

Several DNA repair factors are reduced on nascent chromatin upon DRB and TPL treatments, but as pointed out by Reviewer 2 we have no indication that the DNA damage is reduced on nascent chromatin. Indeed, the variant H2AX (gene name H2AFX) is detected by iPOND-MS, and its abundance is not reduced upon transcription inhibition (Fig. 2F and see panel A below). To further clarify this point, γ H2AX level on nascent chromatin was quantified directly by high content microscopy (QIBC), measuring PLA signal between EdU and γ H2AX. Similar to the iPOND-MS result, γ H2AX levels do not drop on newly replicated chromatin upon transcription inhibition (See panel B below).

DNA damage assessment on nascent chromatin. A. Quantification of the H2AFX peptide from iPOND-MS ($n=3$). Median and standard deviation are shown. Dots, values from each individual experiments. **B.** PLA foci from a PLA between EdU and γ H2AX in cells treated with DMSO, TPL and DRB. CPT is used as a positive control for the assay. >750 nuclei were analysed per condition.

As suggested by Reviewer 2, one possible explanation is that in unperturbed condition, these DNA repair factors are recruited to replisomes due to replicative stress present ahead of the fork. To test this possibility, we propose to isolate chromatin in G1 phase, in cells treated with DMSO, TPL or DRB. To do so we will perform a longer EdU pulse chase experiment (24h, as in Alabert et al. 2015) and collect chromatin by iPOND-MS. Other markers of DNA damage (such as 53BP1) will also be assessed (see R1.2.6).

R2.2. Figure 1B-E: Can the authors also show quantifications of EU, RNAPII and EdU at the 1h and 2h timepoints after the chase?

Quantification of EU and RNAPII 2h after the chase are shown below (Panel A-C). We do not have quantification for EU and RNAPII at the 1h chase. Quantifications of EdU in nascent and 1 or 2hr after the chase are shown below (Panel D). We are happy to repeat these experiments to collect the 1h after the chase time point, and add these controls in the revised version of the manuscript.

Quantification of EU, RNAPII and EdU in chase time points. A. Experimental design. **B.** EU incorporation quantified by QIBC after 2h and 4h of treatment. **C.** RNAPII level quantified by QIBC after 2h and 4h of treatment. **D.** EdU intensities determined by flow cytometry in EdU pulsed cells (20 μ M) and chased for 1h or 2h with thymidine (20 μ M). Left to right, 11min pulse, 1 and 2h chase. Top to bottom, DMSO, TPL and DRB. >950 nuclei were analysed per condition.

R2.3. The authors state in p.7 that "The other proteins are DNA repair proteins involved in fork quality control and HR as well as transcription replication conflicts (Berti et al., 2020)". I think this gives rise to the question if the effect of DRB treatments on the enrichment of certain proteins at nascent DNA is due to the inhibition of transcription elongation inhibition on nascent DNA or in front of replication forks, affecting the enrichment of proteins implicated in handling transcription-replication conflicts in front of the fork and not on nascent DNA itself. The authors should address the possibility that some of the proteins enriched in the iPOND-MS datasets could be there because they are enriched in front of the replisome instead of on the nascent DNA.

Because DNA fragments generated in iPOND range from 100 to 300bp, isolation of unlabelled DNA is unlikely (see below).

Comparison of genomic DNA fragment size using TapeStation. Example of electropherograms of DNA fragments generated by iPOND protocol.

R2.4.1 On this topic, transcription inhibition is performed for two hours prior to the EdU pulse and iPOND-MS procedures. For the DRB treatment, I would expect RNAPII to be paused/stalled prior to the passage of the replication fork that will replicate the analyzed EdU-labelled nascent DNA. This would mean that replication forks during the EdU pulse will encounter paused/stalled RNAPII, generating potential problems. Such interference would most probably lead to chromatin removal of RNAPII from the chromatin. Surprisingly, the authors show enrichment of RNAPII at nascent DNA.

FRAP experiments show that DRB initially increases the pool of stalled RNAPII onto chromatin, followed by RNAPII unloading (Steurer et al. 2018). By QIBC, 2h after DRB treatment, RNAPII abundance on chromatin did not change (Fig. 2N), suggesting that the cycle of unloading has started. Therefore, in these DRB conditions, at the moment of the EdU pulse, forks will not face a higher number of RNAPII. As FRAP experiments suggest that a shorter DRB treatment should increase RNAPII level on chromatin (Steurer et al. 2018), we are happy to test this prediction by QIBC to reconcile FRAP and QIBC observations.

R2.4.2 How can the author differentiate from accumulation of RNAPII in front of the fork, leading to purification by iPOND, and RNAPII on nascent DNA.

As mentioned above, because of fragment size generated in the iPOND protocol, we do not expect to isolate chromatin ahead of the fork (please see R2.3).

R2.4.3 Also, if the accumulation of RNAPII is on the nascent DNA, do the authors suggests that RNAPII gets loaded more on nascent DNA while under DRB inhibition or that stalled RNAPII are mainly by-passed by replication forks, leading to their enrichment on nascent DNA?

Our data suggest that in DMSO and DRB, RNAPII is loaded to similar extent on newly replicated chromatin. In later time points, while in DMSO, RNAPII remains on replicated chromatin, in DRB, RNAPII is unloaded. This is in good accordance with DRB blocking the elongation step and not the loading step of RNAPII.

R2.5. At p.14, the authors state: "Because they share the same DNA template, transcription is known to challenge replisome progression at high frequency, from RNAPII constituting a roadblock to progressing replisomes, to generate RNA:DNA hybrids (Berti et al., 2020). It is therefore remarkable that behind replisomes, only a handful of DNA repair factors appear to be involved in response to RNAPII binding and elongation.". How does the fact that transcription represents a roadblock in front of the forks makes it remarkable that only a handful of DNA damage repair pathways are involved behind the fork (where they are not a roadblock to any replisome)?

Because of the diverse range of issues generated by transcription - replication conflicts (such as RNA:DNA hybrid, head-on and codirectional collisions and topological stress), we were expecting to detect a greater variety of DNA repair proteins at replisomes.

R2.6. At p.11, the authors states: "As di and tri-methylations require several hours to be re-established following DNA replication (Alabert et al., 2015; Reveron-Gomez et al., 2018), 11 minutes after the passage of the fork, such increase most probably reflects an increase of H4K20me2 and H3K9me3 on recycled parental histones.". Can the authors extend their

interpretation of this result? Do the authors think that DRB treatments increase methylation of histones in G1, prior to replication, or specifically in front of the fork (due to conflicts? DNA damage?), and that those methylated histones get recycled on nascent DNA?

Our data suggest that parental histones are modified ahead of the fork upon DRB treatments and transferred on nascent chromatin. We are happy to extend this section in the revised version of the manuscript.

R2.7. Figure 4: The authors perform the histone PTM analysis under 0h (nascent chromatin) versus 2h (re-established chromatin) timepoints. It would have been insightful to also include a 1h timepoint in this experiment. There appear to be some trends/changes but they do not show statistical significance (e.g. H4K5K12ac or H3K14ac). It might be useful to increase the number of biological replicates (including the 1h timepoint) here, which could improve the confidence in the results and/or discover additional transcription-dependent changes of histone PTM restoration.

We are happy to increase the number of biological replicates to improve the confidence of the results.

Minor comments:

1. Fig3I: It would be nice to show a TF from the "Restored within 11 min" category as a comparison point.

We agree with Reviewer 3, PLA experiments between EdU and TFs from this category will be included in the revised version of the manuscript.

2. In page 14, the authors state: "However, we did not detect significant signs of DNA damage in DRB treated cells (Fig. 2A, 2B)". Which signs the authors looked at?

We have measured γ H2AX level on chromatin by QIBC (R1.1.2), on nascent chromatin by PLA between EdU and γ H2AX (R2.1) and in total protein extract by westernblot (see below).

γ -H2AX level in cells treated with DRB or TPL compared to DMSO. Cells were treated with inhibitors for 3hr according to the concentrations indicated. Total extracts were collected and GAPDH used as a loading control.

3. In the iPOND experiment, which size of DNA fragments is achieved?

DNA fragments are typically 150 to 300bp, see point R2.3.

4. At p.14, the authors state: "Because they share the same DNA template, transcription is known to challenge replisome progression at high frequency, from RNAPII constituting a roadblock to progressing replisomes, to generate RNA:DNA hybrids (Berti et al., 2020)." The paper has not addressed the role of RNA:DNA hybrids in these processes.

We used the antibody that recognize RNA:DNA hybrid (Clone S9.6), and tested by high content microscopy its specificity. In our hands, this antibody was not specific and will therefore require further optimization to explore this pathway.

5. Fig3D: Is there enough datapoints to state a conclusion?

This is something we have considered and unless the additional biological repeat increases the number of TFs included in the final analysis, we will take out this panel.

6. S1A: mistakes in the x axis labels ("no EU" in a EdU quantification graph, "no EdU" in a EU quantification graph).

Thank you for pointing out this mistake, it will be corrected in the revised version of the manuscript.

7. S1F is not sufficiently described in the legend. It took me some time and additional efforts to understand what the right panel of S1F was showing.

Thank you for pointing this out, we will modify the legend accordingly.

8. S2E-F: are the axis wrong? Is it supposed to be Nascent when its comparing total extracts?

Thank you for pointing this out. Total extracts are collected at the nascent and the mature time points, providing information 2, 3 and 4hr after drug treatments. For clarity, axis should be labelled X: DMSO (2hr), Y: TPL (2hr). This will be modified in the revised version of the manuscript.

9. A lot of graphs have non-precise axis labels that needs reading of the manuscript and/or legends to understand. For example: 1K-L (distribution, %), 2L (% of the max), 3B-C-D log2(Nascent/2h), 3G IP/Input, 4C (Inhibitor treatment/DMSO (%)), S2E-F (TPL/DRB Nascent/DMSO Nascent), S3A (IP/Input), S4A (No Y axis label).

Thank you for pointing this out, it will be corrected in the revised version of the manuscript.

10. FigS4: Assignment of colors in bar graphs of C-J to treatments is not shown. Heatmaps in H and K do not show if these are 0 or 2h. The heatmap in H shows H3 modification and the heatmap in K shows modification in H3.3 but the labels of the modification in K (except the first one) are the names of the modifications of H3, not H3.3. In the legend, GAPDH is written GABDH.

Thank you for pointing this out, it will be corrected in the revised version of the manuscript.

Reviewer #2 (Significance (Required)):

In this manuscript, the authors characterized the re-establishment of chromatin after DNA replication in fibroblasts using iPOND-MS. As mentioned above, the work provided by the authors feels mainly descriptive and incremental and therefore does not provide further mechanistic insights beyond the current state of the art. Some follow-up experiments to study the functional impact on the different enrichment patterns on nascent DNA or the function of the dependency on RNAPII for the reestablishment of steady-state enrichment on chromatin of some factors would have greatly increased the scientific impact of the manuscript. Nevertheless, the proteome of nascent DNA, its kinetic, and the effect of transcription inhibitors will provide interesting information and a useful resource for research groups in the DNA replication, chromatin, epigenetic, and DNA damage repair fields. Thus, in conclusion, I would recommend this manuscript to be published in its current state in a lower tier journal such as

MBoC or PLOS ONE journals. If the authors can provide additional mechanistic insights by addressing at least a few of the specific points, I think it would become a stronger candidate for a journal with higher impact.

Reviewer #3 (Evidence, reproducibility and clarity (Required)):

In this study, the authors used metabolic labeling of newly-replicated or nascent chromatin followed by quantitative Mass spectrometry (iPOND-MS) to characterize protein composition of nascent chromatin at time points after DNA replication: immediately after a short pulse of EdU labeling (nascent), and after 1 and 2 hours of Thymidine chase (maturing chromatin). The iPOND method was established before but in the current manuscript the authors combined this with inhibiting RNA Pol II transcription at distinct stages to determine the effects on transcription and RNA Pol II cycle on chromatin protein dynamics at the wake of DNA replication. The inhibitors they used are Triptolide, which blocks transcription initiation and induces a proteasomal degradation of all chromatin bound RNA PolII, and DRB, which blocks transcription elongation causing an enrichment of paused RNA PolII. The authors compared the relative enrichment of ~1200 proteins on nascent and maturing chromatin and the effects on transcription inhibition on these proteins.

The authors found that RNA PolII does not affect the loading or retention of most histones on nascent chromatin except for the histone variant H2A.Z, which requires RNA PolII loading. However, DRB treatment (no elongation) resulted in stabilization of all histones (which the authors do not seem to catch on). Interesting, unlike the histone, both replication-coupled and -independent histone chaperons seem to be enriched immediately behind the fork and are affected by RNA PolII to different extents. They next look at ATP-dependent remodelers and find that most remodeler families are facilitated by RNA PolII loading, while elongation affects some remodeler families and not others. They see the same trend looking at a wide variety of transcription factors. Interestingly, while RNA PolII loading is required for the establishment of some histone post translational modifications (H3K36me3), some others such as H3K9me3 and H4K20me2 are negatively affected. Finally, the authors find that RNA PolII elongation promotes binding of several DNA repair proteins, and speculate that this is because of DNA damage from replication-transcription conflict.

R3.1 My main concern about this manuscript is that the relative enrichment of most factors show variability across the time points, which make the interpretation of the data difficult. This becomes more concerning when we look at protein complexes such as the ATP-dependent remodelers. Subunits of the same complex which are expected to bind together show different patterns of enrichment. This raises the concern as to how data was normalized.

Proteins that function as part of a stoichiometric protein complex should be captured equally in an iPOND-MS experiment. However, subunits of the same complex often exhibit different enrichments, including subunits of the replisome (See below, figure from David Cortez, *Methods in Enzymology*, 2017). This has been attributed to the different abundance in tryptic peptides between different subunits. This point will be included in the limitation section of the iPOND-MS method.

Fig. 5 Comparison of the enrichment of selected replisome proteins at replication forks calculated in five proteomic datasets. A log₂ transformation of the mean enrichment comparing fork/chromatin (pulse/chase) is depicted. Larger positive values indicate increased enrichment at forks compared to bulk chromatin. Error bars were calculated as SEM where possible. (...)

R3.2 Furthermore, how do the replicates compare to each other? The others selected ~1200 proteins which were enriched in all three replicates, but how does their relative enrichment compare in the replicates? The authors need to show some kind of comparison across replicates to confirm that the differential relative enrichments are real and biologically meaningful.

Volcano plots were used to select protein of interest (Fig. 2A-B, and S2A-D), p-values providing a mean to identify statistically reproducible results. To further facilitate the assessment of reproducibility, we are happy to include panels as shown below for the proteins highlighted in the result section in the revised version of the manuscript.

Enrichment for selected proteins in individual biological replicate. Mean of the 3 biological replicates with standard deviation is shown. Dot, value from biological replicate.

For the most direct assessment of reproducibility for the iPOND-MS datasets we provide are Principal Component Analysis and hierarchical clustering (Fig. S1B and see below). In general, mature samples are more variable than nascent samples. To compensate for this variability, we propose to increase the number of biological replicates. This will require 1.5 months total of lab work and analysis.

Reproducibility assessment

A. Principal Component Analysis of the three biological replicates presented in this study (Included in Fig. S1B). **B-D.** Relative enrichments for replisome components (top) and the full dataset (bottom). 1-3 refer to the three biological replicates.

R3.3 Also, the TF data is very descriptive. Insightful analysis of similarities/differences between types of TFs would be interesting.

We did not include this analysis in the manuscript as each sub-type of TFs is formed of ~30 TFs, limiting the relevance of the findings. Unless the additional biological replicate increases the number of TFs identified, we will not include these panels.

Analysis of the different types of TF. A. Pioneer TFs. **B.** dbTF. **C.** DNA binding domains. 1, Enriched on nascent compared to mature chromatin in DMSO. 2, Restored within 11 min in DMSO. 3, Delayed restoration in DMSO.

R3.4 Minor comment: The formaldehyde cross linking used in iPOND makes it difficult to interpret/distinguish what is actually chromatin bound versus what is enriched due to protein-protein cross linking. The authors should highlight that in the limitations section.

This point will be added in the limitations section of the revised version of the manuscript.

****Referees cross-commenting****

I agree with most of Reviewer 1's comments about the lack of proper controls and normalization, which make the interpretations difficult. Particularly all of the controls mentioned

under point 1 should not be difficult to perform, and if included, would strengthen the study and the manuscript.

All the controls required by Reviewer 1 are provided page 3-6.

Reviewer 1 makes an important point about normalization, which I totally agree with. Ideally, a spike-in approach would help obtain a much more quantitative and reliable understanding of differential protein enrichment. However, repeating all iPOND experiments with spike-in might be a big ask.

We would like to thank Reviewer 3 for his/her understanding. As mentioned in R1.2.1 point 7, to our knowledge, while spike-in is routinely used to normalize EdU-seq experiments, no such spike-in exist to quantify EdU based proteomic dataset.

What the authors could do at minimum is show how replicates compare with each other. It looks like they pooled all three replicates for analysis, but comparing relative enrichment of all 1257 proteins across replicates would help.

We provide assessment of reproducibility of the dataset (R3.2) that we will include in the revised result section of the manuscript.

The point about delayed histone occupancy is a critical one and difficult to rationalize. To note, histone chaperons are enriched on nascent, but histones are not.

Although histones and histone chaperones form a complex on newly replicated chromatin, their binding kinetic during the maturation process is not expected to be similar. Histone chaperone such as CAF-1 are expected to be enriched on newly replicated chromatin compared to mature samples, mirroring the kinetic of replisome proteins. Histones on the other hand are expected to be low abundant behind replisomes compared to later time points.

Histone and histone chaperones kinetics. **A.** Relative enrichments of histones and replication coupled histone chaperones from iPOND-MS dataset. **B.** Rational of the findings. Changes in relative abundance of the proteins involved along the maturation process are shown.

Besides, in the current way that the data is analyzed and presented, there are a lot of fluctuations in protein enrichment across the 1-2 hour timepoints of chromatin maturation, which would be very interesting if real. For e.g., Fig. 1I, Triptolide treatment, most of the cluster I and cluster II proteins show medium-high enrichment on nascent, depleted in 1h, but recover in 2h. If the binding/recruitment of these proteins on newly-replicated chromatin is RNA Pol II dependent, why would they come back after 1h? If this real, this would be very interesting.

We agree with Reviewer 3, and as we have no indication that one experiment should be removed from the analysis, our data support that chromatin maturation is a dynamic process with potential transient binding events. To increase the robustness of this observation, we

propose to perform an additional biological replicate for a total of n=4. This will require 1.5 months total of lab work and analysis.

There are several additional examples of problems with quantification/normalization. As for SWI/SNF subunits, both SMARCA4 and SMARCC1 are core subunits and based on several thorough biochemical studies, cannot be expected to bind separately. However, they show different kinetics in DMSO as well as TPL and DRB.

SMARCA4 and SMARCC1 show slightly different kinetics in DMSO, but these differences are not significant. Consistent with this, both subunits belong to the same cluster, cluster 7.

SMARCA4 and SMARCC1 binding kinetics in DMSO, TPL and DRB. Mean of 3 biological replicates and standard deviation are shown.

Another problem of the assay is that it shows genome-wide average. As Reviewer 1 rightly pointed out, transcription inhibition could disproportionately affect chromatin maturation kinetics in different genomic regions. Perhaps it would be interesting to analyze sets of genomic regions separately, such as highly transcribed and lowly transcribed genes. This might be achieved by adding a purification step using pools of DNA sequence probes before or after the streptavidin enrichment.

We have used replication timing as a mean to compare chromatin dynamic in highly transcribed regions (euchromatin) and low transcribed regions (heterochromatin) (Alvarez et al. 2023, Cell reports, Figure 4). As this strategy requires cell cycle synchronisation or cell sorting, we decided to avoid exposing cells to additional stress. Re-CHIP strategies such as CHOR-seq would be the best approach to tackle this question in a future study.

Additional comment: The formaldehyde cross linking used in iPOND makes it difficult to interpret/distinguish what is actually chromatin bound versus what is enriched due to protein-protein cross linking. The authors should highlight that in the limitations section.

This is an important point, and we will include it in the limitations section of the revised version of the manuscript.

On a positive note, it is a very important and timely study and the manuscript has a lot to consider. Addition of proper controls and normalization/analysis of replicates will make it stronger

We hope that our point-by-point response and the set of new experiments proposed will satisfy Reviewer 3.

Reviewer #3 (Significance (Required)):

Overall, it is a very important and timely study, and the manuscript has a lot to consider. There are several recent papers on the kinetics of chromatin maturation behind the replication fork, and this study adds a very important dimension to this ongoing investigation, and will be of interest to a broad readership in the chromatin and transcription field. We are very happy Reviewer 3 acknowledges the relevance of our work.

Dear Constance,

Thank you for the submission of your manuscript with referee reports and proposed revision plan to EMBO reports.

I discussed your study with my colleagues here and we would like to invite you to revise it for our journal, along the lines you suggest in your proposed revision plan.

I would thus like to invite you to revise your manuscript with the understanding that the referee concerns must be fully addressed and their suggestions taken on board. Please address all referee concerns in a complete point-by-point response. Acceptance of the manuscript will depend on a positive outcome of a second round of review. It is EMBO reports policy to allow a single round of major revision only and acceptance or rejection of the manuscript will therefore depend on the completeness of your responses included in the next, final version of the manuscript.

We realize that it is difficult to revise to a specific deadline. In the interest of protecting the conceptual advance provided by the work, we recommend a revision within 3 months (28th Oct 2023). Please discuss the revision progress ahead of this time with the editor if you require more time to complete the revisions.

- 1) A data availability section providing access to data deposited in public databases is missing. If you have not deposited any data, please add a sentence to the data availability section that explains that.
- 2) Your manuscript contains statistics and error bars based on $n=2$. Please use scatter blots in these cases. No statistics should be calculated if $n=2$.

5) a complete author checklist, which you can download from our author guidelines . Please insert information in the checklist that is also reflected in the manuscript. The completed author checklist will also be part of the RPF.

6) Please note that all corresponding authors are required to supply an ORCID ID for their name upon submission of a revised manuscript (. Please find instructions on how to link your ORCID ID to your account in our manuscript tracking system in our Author guidelines

- the name of the statistical test used to generate error bars and P values,
- the number (n) of independent experiments (please specify technical or biological replicates) underlying each data point,
- the nature of the bars and error bars (s.d., s.e.m.),
- If the data are obtained from n Program fragment delivered error ``Can't locate object method "less" via package "than" (perhaps you forgot to load "than"?) at //ejpvfs23/sites23b/embor_www/letters/embor_decision_rc_revise_and_rereview.txt line 56.' 2, use scatter blots showing the individual data points.

11) All Materials and Methods need to be described in the main text. We would encourage you to use 'Structured Methods', our new Materials and Methods format. According to this format, the Materials and Methods section should include a Reagents and Tools Table (listing key reagents, experimental models, software and relevant equipment and including their sources and relevant identifiers) followed by a Methods and Protocols section in which we encourage the authors to describe their methods using a step-by-step protocol format with bullet points, to facilitate the adoption of the methodologies across labs. More information on how to adhere to this format as well as downloadable templates (.doc or .xls) for the Reagents and Tools Table can be found in our author guidelines: . An example of a Method paper with Structured Methods can be found here:

12) I would also like to alert you that EMBO Press offers a new format for a video-synopsis of work published with us, which essentially is a short, author-generated film explaining the core findings in hand drawings, and, as we believe, can be very useful to increase visibility of the work. This has proven to offer a nice opportunity for exposure i.p. for the first author(s) of the study. Please see the following link for representative examples and their integration into the article web page:

https://www.embopress.org/video_synopses
<https://www.embopress.org/doi/full/10.15252/emj.2019103932>

I look forward to seeing a revised form of your manuscript when it is ready.

Best wishes,
Esther

I. REVISION PLAN – and work done during the revision period.

Comments from reviewers are labelled as followed: **R1.1.1**: Reviewer 1, Comment 1, Section 1. Sections were created to address the multiple points raised by reviewers within one comment.

Description of the planned revisions

- 1) To validate the key findings of this study: use an orthogonal approach, PLA-EdU (R1.2.4) – **New figure EV1J and EV4A.**
- 2) To strengthen the iPOND-MS conclusions: increase the number of biological replicates (R2.7, R3.2) – **New Figure 1 (n=4)**
- 3) To distinguish the effect of TPL and DRB ahead and behind replisomes: iPOND-MS in chase experiments (24h, as in Alabert et al. 2015) (R1.2.6, R2.1) – **New Figure 4G-I (n=3)**
- 4) To measure the proportion of DNA replication independent EdU signal: Quantification of PCNA and EdU foci colocalization (+/- transcription inhibitor) (R1.1.3) – **Figure R1.1.3, p6**
- 5) Test FRAP based prediction: measure RNAPII level on chromatin after short DRB treatment by QIBC (R2.4.1) – **Figure R2.4.1, p15.**

Existing Data and information we would like to include in the revised version of the manuscript.

- 1) Effect of DMSO: Analysis of EdU incorporation, cell cycle distribution and DNA damage in +/- DMSO (R1.1.2) – **Figure R1.1.2, p5.**
- 2) EdU chase conditions: pulse chase quantification by flow cytometry (R1.1.5) – **New Figure EV1B.**
- 3) Effect of TPL/DRB on DNA repair: Quantification of γ H2AX levels in EdU positive and negative cells (R1.1.3) – **New Figure EV4B, EV4C.**
- 4) Effect of TPL/DRB on the replication program: Analysis of S phase progression, cell cycle- and S phase-distribution (QIBC) (R1.1.4) – **New Figure EV1C, EV1D.**
- 5) DNA damage on newly replicated chromatin: PLA experiment between EdU and γ H2AX +/- transcription inhibition as well as the quantification of the H2AFX peptide from iPOND-MS (n=3) (R2.1) – **New Figure 3C and Figure R2.1 p13.**
- 6) DNA damage on chromatin: Quantification of γ H2AX +/- transcription inhibition on total chromatin via QIBC and Western Blot (R1.1.2, R2.1) - **New Figure EV1E, and R1.1.3 p5.**
- 7) Assessment of the reproducibility of the dataset: **PCA with identified proteins of each individual biological replicate (R3.2) – New Fig. EV1F.**
- 8) Expand the limitation section (R3.1 and 3.4) – **New extended limitation section.**

Description of analyses that authors prefer not to carry out.

At this point, we do not consider repeating all the iPOND-MS experiments using a spike-in for external normalisation due to the reasons presented in R1.2.1 (p6).

II. POINT BY POINT RESPONSE TO REVIEWER COMMENTS

Reviewer #1 (Evidence, reproducibility and clarity (Required)):

The goal was to characterize the changes in the composition of the proteome associated with replicated DNA in conditions of genome-wide inhibition of transcription initiation or transcription elongation. They use iPOND, a MS based technique that identifies proteins specifically associated with replicated DNA labeled with EdU. They use non-synchronized foetal human lung fibroblasts and examine time points immediately after replication (after the 11 min EdU pulse) and 1 and 2 hrs after the Thymidine "chase" later when chromatin has "matured", to assess how the inhibition of transcription influences chromatin maturation and the binding of chromatin associated proteins to replicated DNA. The question is pertinent and is in line with the long-standing interest of the group in chromatin replication dynamics. They conclude that 1. RNAPII loading is necessary for the binding of some TFs, chromatin remodelers and DNA repair factors and 2. RNAPII elongation is needed for H2A.Z incorporation, H3K36me2 restoration and DNA repair factor binding. Transcription is on the other hand not needed for nucleosome assembly, histone acetylation and H3K9me3, H3K27me3 and H4K20me2 incorporation or restoration.

There are two main issues that make the interpretation of their results very difficult and make me question their conclusions:

R1.1.1 They don't provide sufficient evidence that the treatments with TPL and DRB do not interfere with replication. The distributions of EdU intensity per EDU+ cell after treatment in Figures 1D-E and S1A are not sufficient. It is not clear why EdU incorporation is so heterogeneous in the cell population (the range of intensities goes from near 0 to 50000!), which makes me wonder if the DMSO treatment also has an effect on replication. I don't think this heterogeneity can simply be explained by the fact that the cell population is asynchronous.

In the former Figure 1D-E, a linear scale was shown instead of a logarithmic scale to facilitate comparison of EdU incorporation upon DMSO, TPL and DRB treatments. As shown below, the range of EdU intensities is in good accordance with previous publications (see examples from Massimo Lopes and Luis Toledo labs). Based on reviewer 1's comment, a log scale is now used in the new Figure 1D.

Range of EdU intensities. Cells were labelled with EdU and imaged by Quantitative Image-based Cytometry (QIBC). EdU intensities (a.u., arbitrary units) are shown as a function of total DAPI intensities. Each dot represents a nucleus identified based on DAPI intensities. This study, data from Fig. 1D, EdU intensities in S phase cells range from 800 to 50.000. Berti et al. 2020, from 6 to 600. Zonderland et al. 2022, from 30 to 4000.

R1.1.2 They need to show a -DMSO control as well.

A standard concentration of DMSO was used (0.06%). When comparing DMSO and no DMSO conditions, no defects in EdU incorporation, changes in cell cycle distribution or increases in DNA damage were observed (see panels below).

Comparison of DMSO and no DMSO conditions. Left, EdU incorporation. Cells were labelled with EdU for 20 min and imaged by QIBC. Each dot represents a single cell. Middle, cell cycle distribution. EdU and DAPI were used to gate G1, S and G2 phase cells. Right, quantification of γ -H2AX fluorescence signals in individual cells. Each dot represents a single cell. Median values are shown in red. Cells were treated 3h with Camptothecin (Campto, 1 μ M) as a positive control for DNA damage. >1464 nuclei were analysed per condition.

R1.1.3 Besides since they are only using a positive EdU signal as their criteria for replicating cells, they cannot rule out that some of the EdU signal is coming from DNA repair after replication and depending on how deleterious DMSO/TPL/DRB are to replication the fraction of cells that undergo DNA repair might be significant.

The drug treatments used in this study, DMSO, TPL and DRB, do not increase γ -H2AX signal (see right panel above). γ -H2AX levels have also been measured in EdU positive and negative cells (see panels below). It seems therefore unlikely that a significant proportion of EdU signal is coming from DNA repair.

Quantification of γ -H2AX signal in EdU positive and negative cells. Quantification of γ -H2AX fluorescence signals in individual cells measured by QIBC. Cells were treated with TPL and DRB according to conditions used in the study, and labelled with EdU for the last 20 min. Camptothecin treatment is used as a positive control for DNA damage. >600 nuclei were analysed per condition.

Moreover, we have quantified by microscopy the number of EdU and PCNA foci per cells (Panel B) and their degree of colocalization in +/- transcription inhibitor conditions (Panel A). The drug treatments used in this study did not affect the number of EdU and PCNA foci per cells nor their colocalization. For the colocalization experiment, we included a HU treatment as a control, as PCNA is known to lose its colocalization with EdU in these conditions (Sirbu et

al., 2011). It seems therefore unlikely that a significant proportion of EdU signal is coming from DNA repair.

Quantification of number of PCNA and EdU foci and their colocalization. Cells were treated with TPL and DRB according to conditions used in the study and labelled with EdU for the last 20 min. **A**, the Pearson correlation between EdU and PCNA foci was measured in Volocity. >91 nuclei were analysed per condition. **B**, the number of PCNA and EdU foci were counted per cell using the “foci detector” module for QIBC developed by Jiri Lukas laboratory. >3725 nuclei were analysed per condition.

R1.1.4 More importantly, they need to show that the various treatments don't interfere with the replication program, especially since replication is coupled with new nucleosome assembly and the transcription of replication dependent histone variants is induced during S-phase. Transcription inhibition could disproportionately affect the replication of some parts of the genome more than others and since there is no evidence to the contrary the differences that they observe between the TPL/DRM treated and DMSO treated proteomes bound to replicated DNA could just be because they were isolated from different genomic loci.

We have carefully examined the impact of TPL and DRB treatments on the replication program using multiple approaches. First, inhibitor treatments minimally affect DNA replication efficiency (Fig. 1D) and do not alter S phase progression (see panel below), which is in good accordance with previous published work (Stewart-Morgan et al. 2020).

Analysis of S phase distribution upon DMSO, TPL and DRB treatment. Flow cytometry plots of asynchronous cells treated with DMSO, TPL and DRB. Cells were EdU pulsed and chased according to the conditions defined in the manuscript. Nas, Nascent sample; 2h, Mature sample.

Second, we have inspected replication foci patterns (Dimitrova & Gilbert 1999) and quantified the distribution of the cell population between early/mid and late S phase cells. We did not observe any abnormal patterns, or changes in patterns distribution upon transcription inhibition as observed after DNA damage (Seiler et al. 2007) or upon RIF1 depletion (Yamazaki et al. 2012) (see panel below). Altogether these data support that the replication program is unaffected 2h after transcription inhibition. We have included these controls in the revised version of the manuscript as new Figure EV1C-D.

Cell cycle and S phase distribution. DMSO, TPL and DRB treated cells are pulsed with EdU for 20 min and analysed by microscopy. A) Representative images of Early/Mid and Late S phase patterns. Early and Mid-S phase cells have a pattern of replication foci distributed throughout the nucleus. Late-S-phase cells would have a small number of large foci within the nucleus. B) Distribution of cells in G1 and G2 phase based on EdU and DAPI intensities. Distribution of cells in Early/Mid (top section) and late S phase (bottom section) based on the distinct EdU patterns shown in A. >1230 nuclei were analysed per condition.

R1.1.5 I am also not convinced that they are able to stop EdU incorporation after 11min with the addition of only equimolar amounts of Thymidine (20 μ M EDU and 20 μ M Thymidine). Equimolar amounts of Thymidine are not sufficient to stop EdU incorporation rapidly. They need to show the kinetics of EdU incorporation in synchronized cells +/- Thymidine.

EdU incorporation has been monitored after thymidine chase in TIG-3 cells and showed that the chase is stopping EdU incorporation (see panel below). This new panel has been added to the revised version of the manuscript (Fig. EV1B).

EdU pulse chase quantification by flow cytometry. Cells were pulsed for 11 min with EdU (20 μ M) (Nascent sample) and chased 2h by 20 μ M of thymidine (Mature sample).

R1.1.6 Without these controls it is impossible to draw any meaningful conclusions from the iPOND data.

We hope that the controls provided above and in the revised version of the paper will satisfy Reviewer 1 concerns regarding the experimental conditions of this study.

R1.2.1 The normalization of iPOND and total protein MS data is problematic. It seems that each time point from each treatment was first normalized internally to the median of all protein levels in each dataset and then the relative abundances of each protein were normalized to 100% over all treatments and time points. Internal normalization makes it impossible to directly compare time points and treatments between each other. If the enrichment of a protein goes down from one time point to the next it doesn't mean that there is less of that protein on replicated DNA in absolute terms, it just means that there is less of that protein relative to the median of the whole set of proteins at that time point.

Reviewer 1 is raising an important point. Each sample is median normalised based on the assumption that most of the proteins in the compared samples have similar abundances and only a relatively small percentage of proteins are differentially abundant. As pointed out by Reviewer 1, median normalisation potentially erases biological differences between samples. Yet, in our iPOND and NCC datasets such differences have never been observed across biological replicates, whether we use replication stress agents (HU, CPT, ATMi), translation inhibitor (CHX), or here transcription inhibitors (TPL, DRB) (Panel A). Another possible normalisation method is to use histones as protein rulers. This method was successful for NCC-MS (Alabert et al. 2014) but proven to be problematic for iPOND-MS. In NCC (20 min biotin-dUTP labelling), the amount of histone detected on nascent and mature samples moderately changes. In iPOND (11min EdU labelling) it greatly increases (Fig. EV1H). Consequently, normalising the iPOND-MS dataset using histones shifted the proteins toward nascent time points (Panel B). This method was therefore considered not appropriate for this dataset. Finally, Reviewer 1 is suggesting an external normalisation method using a “spike in” protein. EdU-labelled DNA from *Drosophila* S2 cells has been spiked in samples prior to the Click-iT reaction and used to normalize read counts in iPOND sequencing experiments (Reveron et al. 2018). However, normalising iPOND mass spectrometry dataset using a ‘spike in’ protein, to allow to correct for sample preparation variation, requires a synthetic peptide or a protein clearly distinguishable from the human homologue and that binds EdU labelled DNA. To our knowledge, there is no known protein that could be used for this purpose, including histones due to the high degree of histone sequence conservation.

Normalisation methods. **A.** Box plots showing protein intensities prior to normalisation for 9 time points in 3 biological replicates (27 total). **B.** Box plots showing the protein intensities after histone normalisation. Yellow arrowheads, nascent samples.

R1.2.2 Their claim that they are comparing iPOND enrichments to total protein abundance is misleading since the data from total protein extracts was also internally normalized so they are comparing relative enrichments in iPOND data to relative enrichments in total cell extracts, which unsurprisingly do not correlate.

In whole cell extracts, 2 hr after TPL or DRB treatments, only a small percentage of proteins are expected to be differentially abundant (Skalska et al. 2021). As for R1.2.1, analysis of the raw intensities further supports that median normalisation is an appropriate normalisation method (see panel below). Of note, this comparison is important to identify changes on replicated chromatin that may be mirrored by changes in whole cell extracts (see new Figure EV2F).

Comparison of newly replicated chromatin and total extracts. A. Box plots showing the protein intensities prior to normalisation for 3 conditions, 3 biological replicates (9 in total). Whiskers show min to max.

R1.2.3 It is impossible to make any meaningful conclusions about proteome dynamics using this kind of analysis. They should have used external normalization with a "spiked in" protein to be able to directly compare time points and treatments.

Please see R1.2.1, point 7 above.

R1.2.4 Such as it is right now, their analysis produces some puzzling conclusions that I suspect will turn out to be artefacts of their normalization procedure.

Based on Reviewer 1's comment, we have worked with a bioinformatician from the Computational Department at the University of Dundee, Marek Gierlinsky. He has independently re-analysed all the proteomic datasets presented in this study to:

- Include new tools available to correct for the batch effects often observed in TMT proteomics (HarmonizR, Voß et al., 2022)
- Use more robust statistical tools than individual t-test to identify differential abundance of proteins (limma, Ritchie et al., 2015)

This new analysis is the one presented in the revised version of the manuscript as it contains more robust statistical tools. Of note, it is in good accordance with the pre-revised version of the manuscript. Finally, we have experimentally validated several of our findings using an orthogonal approach based on high content microscopy, Proximity Ligation Assay (Serebryanny and Misteli, 2019, Fenstermaker et al. 2023), selecting a subunit of RNAPII, a TF and a DNA repair factor (Fig. EV1J, Fig. 3I, Fig. EV4A). Therefore we do not think that our conclusions are the result of normalisation procedure.

R1.2.5 It is not clear for example why the appearance of histones on replicated DNA would be delayed as they claim: in yeast nucleosomes (new and old recycled ones) are assembled on replicated DNA within minutes of the passage of the replication fork, I don't see why this would not be the case in human cells since the replication machinery is essentially the same in humans and yeast.

Yeast and human are indeed very similar with only ~300 bp of DNA nucleosome free behind each replisome in average. Consequently, even if nucleosome assembly is rapid, when pulling down nascent chromatin, we do expect to pulldown less histones than in mature samples. We, and others (Lopez-Contreras et al. 2013), have detected a similar drop in histone abundance on nascent chromatin compared to mature samples by iPOND-MS, and iPOND-western blot, whether the same amount of protein or number of cells is loaded per well (Fig. EV1H).

R1.2.6 It is also puzzling why RNAP2 is enriched in the nascent and 1hr time points but then becomes depleted in the 2hr time point in the DRB treatment since global RNAPII levels don't change in the DRB treatment compared to DMSO (Figure 1C).

This is either an interesting biological difference between chromatin and newly replicated chromatin, or due to the different sensitivity of the two technologies used (High content microscopy for chromatin, iPOND-MS for replicated chromatin). To compare the effect of transcription inhibition on steady state chromatin and newly replicated chromatin we have performed an iPOND-MS on cells EdU pulse chased for 24h, as previously done in Alabert et al. 2015 (Revised Fig. 4G and point R2.1). The systematic comparison of proteins bound to newly replicated chromatin and steady state chromatin revealed that many RNAPII dependent effects are specific to newly replicated chromatin.

R1.2.7 All the conclusions for PTM restoration/incorporation are essentially meaningless: internal normalization makes it impossible to detect whether PTM levels double at the 2hr time point compared to the Nascent time point in the DMSO treatment, as would be expected for all examined PTMs except for H4K5K12Kac which are marks of new histones.

The histone PTMs data are not internally normalized. For each biological replicate and each time point, for any given peptide identified, the proportion of each modified state of the peptide is calculated as in Alabert et al 2015. For instance, for the peptide histone H4 aa20-24 carrying K20, the proportion of peptides with K20me0, K20me1, K20me2 and K20me3 are calculated. Moreover, in human, histone methylations are not expected to double within 2h, requiring instead 4h to 20h, depending on the histone residue (Alabert et al. 2015, Reveron et al. 2018). Histone modifications expected to decrease within 2h are H4K5K12ac and are behaving accordingly in this dataset (Fig. 2D). This dataset is therefore in good accordance with published proteomic analysis of histone modifications.

R1.2.8 Right now, relative PTM levels are all over the place: only histone acetylations seem to increase, while H3K9me3 and H3K27me3 don't change even though they should also double since heterochromatin should also be restored on both sister chromatids.

H3K9me3 and H3K27me3 are not expected to double within 2h in human cells (see point above). This point has been added to the result section, p7.

R1.2.9 They will only be able to accurately assess the impact of transcription inhibition on PTM restoration when they are able to reliably measure the rate of increase of PTM levels during chromatin maturation.

This is what is shown for the first 2h after the passage of the fork. Longer time points (12 and 20h for instance) would be required to complete the histone methylation restoration analysis. However, such experiments will be affected by the known effect of prolonged transcription inhibitors treatments on genome stability (Skalska et al. 2021).

****Referees cross-commenting****

On reviewer's 2 comment on significance:

I think a thorough descriptive analysis of a biological process is extremely valuable and unlike my colleague, I think these types of studies need to be published in high impact journals with a broad readership. Biological processes need to be described first as completely as possible before we can propose meaningful models on how they function and identify the molecular mechanisms that execute and regulate them. As my colleague is surely aware, thorough descriptive studies of any poorly characterized biological process take years (i.e. at least one

grant cycle) and comprehensive follow up mechanistic studies can take even longer than the initial descriptive study and can only be done during the following grant cycle, if the authors were lucky enough to obtain funding. Funding agencies however are more likely to award grants to perform these follow up mechanistic studies if the authors (especially if they are junior PIs) have published in higher impact journals in their previous grant cycle. The kind of thinking exhibited by reviewer 2 disproportionately disadvantages junior PIs that work on understudied biological processes. It is a disservice to scientific progress to dismiss excellent descriptive studies and "downgrade" them to lower impact journals where they will be unfairly labeled as a "work of lesser importance". This kind of thinking is also a disservice to the lower impact journals that often publish works whose quality is comparable to articles published in high impact journals. I value more any comprehensive description of a biological process over what most of the time passes for mechanistic insight that is deemed worthy of publication in a high impact journal i.e. a hastily analyzed phenotype of, more often than not, one single mutant tacked on at the end of a descriptive study. This one mutant phenotype then forms the basis of a somewhat "slapdash" model that is often proven wrong by subsequent publications and that the authors would have probably dismissed themselves had they been given more time to develop and test their model in a follow up publication.

I do not think the main issue with the present study is its descriptive nature. As I said in my review, the main issues are technical: the lack of external normalization of MS data and insufficient evidence of the impact of transcription inhibitors on replication dynamics. The study should not be published in any journal (high or low impact) before those issues are resolved.

on reviewer 2's remarque 4. in major comments:

iPOND identifies proteins bound to 100-300bp fragments labeled with EdU (i.e. after replication or DNA repair). It is by definition identifying proteins bound to chromatin behind the fork, so I don't think that the isolation of RNAPoIII bound in front of the fork is a major issue

Reviewer #1 (Significance (Required)):

I am not convinced by their conclusions and I cannot recommend that the the study be published at this stage due to normalization issues and insufficient evidence that transcription inhibition does not perturb the replication program (see above). They would need to redo all the iPOND experiments using external "spike in" normalization and monitor replication genome-wide before they can make any meaningful conclusions about the transcription dependent composition of the proteome associated with replicated DNA.

Expertise keywords: Chromatin, Genomics (assay development and bioinformatics analysis) , Replication, Transcription

Reviewer #2 (Evidence, reproducibility and clarity (Required)):

In this manuscript, the authors characterized the re-establishment of chromatin after DNA replication in fibroblasts using iPOND-MS. By using a short pulse of EdU, followed by different length of thymidine-chase, the authors compare the proteome at nascent DNA (just after the EdU pulse) with the proteome on re-established chromatin (1h and 2h post EdU pulse). Moreover, by using two different transcription inhibitors, they investigate the implication of active transcription elongation and of RNAPII binding itself on the reestablishment of chromatin. They show that different transcription factors bind to newly replicated DNA with different kinetics and are affected differentially by transcription inhibition. They also show that upon transcription inhibition by DRB, certain DNA damage repair proteins are depleted, implicating transcription in the recruitment of these factors at nascent DNA. Chromatin remodelers were shown to be enriched on nascent DNA, but triptolide-transcription inhibition reduced their enrichment, implicating RNAPII in the reestablishment of chromatin structure and of steady-state chromatin accessibility. Lastly, the authors show that histone incorporation and histone modification restoration on nascent DNA is mostly uncoupled from transcription with the exceptions of H3.3K36me2 (transcription inhibition by triptolide or DRB drastically reduces restoration) and H4K16ac (DRB treatment increases its incorporation in nascent DNA). Overall, the results and the analysis of the datasets appear robust and well executed. Nevertheless, the work provided by the authors feels mainly descriptive and does not provide further mechanistic insights beyond the current state of the art. Some follow-up experiments to study the functional impact on the different enrichment patterns on nascent DNA or the function of the dependency on RNAPII for the reestablishment of steady-state enrichment on chromatin of some factors would have greatly increased the scientific impact of the manuscript. Nevertheless, the proteome of nascent DNA, its kinetic, and the effect of transcription inhibitors will provide interesting information and a useful resource for research groups in the DNA replication, chromatin, epigenetic, and DNA damage repair fields. Thus, in conclusion, I would recommend this manuscript to be published in its current state in a lower tier journal such as MBoC or PLOS ONE journals. If the authors can provide additional mechanistic insights by addressing at least a few of the specific points listed below, I think it would become a stronger candidate for a journal with higher impact.

Major comments:

R2.1. At p.7, the authors state: "Altogether, this analysis further confirms that RNAPII's binding and elongation on newly replicated chromatin are a source of genotoxic stress, and identifies dedicated repair factors handling transcription replication conflicts.". I don't think that depleted DNA repair proteins from nascent chromatin upon DRB treatment is enough to claim that the analysis confirms that transcription on nascent DNA is a source of stress. Another possibility could be that transcription helps handling prior DNA damage on nascent DNA without causing the damage. A useful experiment to clarify this point would be the direct quantification of DNA damage markers on nascent chromatin (e.g yH2AX-EdU colocalization quantification by immunofluorescence). Has the yH2AX variant been detected in the iPOND MS dataset? Another possible follow-up experiment could be to detect direct physical DNA damage on nascent DNA for example by using a TUNEL assay or similar DSB mapping method. Can the DNA damage be prevented by DRB or TRP addition?

Several DNA repair factors are reduced on nascent chromatin upon DRB and TPL treatments, but as pointed out by Reviewer 2 we have no indication that the DNA damage is reduced on nascent chromatin. Indeed, the variant H2AX (gene name H2AFX) is detected by iPOND-MS, and its abundance is not reduced upon transcription inhibition (Fig. EV2D and see panel A

below). To further clarify this point, γ H2AX level on nascent chromatin was quantified directly using another approach, by high content microscopy (QIBC) measuring PLA signal between EdU and γ H2AX. Similar to the iPOND-MS result, γ H2AX levels do not drop on newly replicated chromatin upon transcription inhibition (See panel B below).

DNA damage assessment on nascent chromatin. A. Quantification of the H2AFX peptide from iPOND-MS ($n=3$). Median and standard deviation are shown. Dots, values from each individual experiments. **B.** PLA foci from a PLA between EdU and γ H2AX in cells treated with DMSO, TPL and DRB. CPT is used as a positive control for the assay. >750 nuclei were analysed per condition.

As suggested by Reviewer 2, one possible explanation is that in unperturbed condition, DNA repair factors are recruited to replisomes due to replicative stress present ahead of the fork. We do not have tools to isolate chromatin ahead of the fork. But we have tools to isolate steady state chromatin, either by chromatin fractionation or by iPOND with a longer EdU pulse chase experiment (24h, as in Alabert et al. 2015). We have chosen the “iPOND longer chase” over “chromatin fractionation” to compare the composition of chromatin and replicated chromatin, enabling to use the same method, thereby limiting bias of comparing two different methods (New Fig. 4H). In this setup, H2AFX levels dropped upon TPL or DRB treatment compared to DMSO (see panel below), but the changes were not significant. Therefore, although these observations support that there may be less damage on chromatin upon TPL and DRB treatment compared to DMSO, we did not include this point in the result section. The full dataset available in Table S7/8.

H2AFX relative intensities in DMSO, TPL and DRB treatments in 24h chased chromatin. Each colour represents an individual biological replicate. The experimental design is described in Fig. 4H.

R2.2. Figure 1B-E: Can the authors also show quantifications of EU, RNAPII and EdU at the 1h and 2h timepoints after the chase?

Quantification of EU and RNAPII 2h after the chase are shown below (Panel A-C). Quantifications of EdU in nascent and 1 or 2hr after the chase are shown below (Panel D).

Quantification of EU, RNAPII and EdU in chase time points. **A.** Experimental design. **B.** EU incorporation quantified by QIBC after 2h and 4h of treatment. **C.** RNAPII level quantified by QIBC after 2h and 4h of treatment. **D.** EdU intensities determined by flow cytometry in EdU pulsed cells (20 μ M) and chased for 1h or 2h with thymidine (20 μ M). Left to right, 11min pulse, 1 and 2h chase. Top to bottom, DMSO, TPL and DRB. >950 nuclei were analysed per condition.

R2.3. The authors state in p.7 that "The other proteins are DNA repair proteins involved in fork quality control and HR as well as transcription replication conflicts (Berti et al., 2020)". I think this gives rise to the question if the effect of DRB treatments on the enrichment of certain proteins at nascent DNA is due to the inhibition of transcription elongation inhibition on nascent DNA or in front of replication forks, affecting the enrichment of proteins implicated in handling transcription-replication conflicts in front of the fork and not on nascent DNA itself. The authors should address the possibility that some of the proteins enriched in the iPOND-MS datasets could be there because they are enriched in front of the replisome instead of on the nascent DNA.

Because DNA fragments generated in iPOND range from 100 to 300bp, isolation of unlabelled DNA is unlikely (see below).

Comparison of genomic DNA fragment size using TapeStation. Example of electropherograms of DNA fragments generated by iPOND protocol.

R2.4.1 On this topic, transcription inhibition is performed for two hours prior to the EdU pulse and iPOND-MS procedures. For the DRB treatment, I would expect RNAPII to be

paused/stalled prior to the passage of the replication fork that will replicate the analyzed EdU-labelled nascent DNA. This would mean that replication forks during the EdU pulse will encounter paused/stalled RNAPII, generating potential problems. Such interference would most probably lead to chromatin removal of RNAPII from the chromatin. Surprisingly, the authors show enrichment of RNAPII at nascent DNA.

On nascent chromatin and on steady state chromatin, after two hours of DRB treatment, we did not detect a significant increase or decrease in RNAPII abundance (See panels A, B). As FRAP experiments suggest that a shorter DRB treatment should increase RNAPII level on chromatin (Steurer et al. 2018), we have examined the abundance of RNAPII on chromatin by QIBC after shorter DRB treatments (see Panel C). As a control we included Actinomycin D that has been shown to increase the level of RNAPII on chromatin (Brahma and Henikoff, 2023). In these conditions, we did not detect an increase of RNAPII after DRB treatment suggesting that the replication fork is unlikely faced with higher level of arrested RNAPII upon DRB treatment in these conditions.

RNAPII abundance on nascent chromatin and steady state chromatin upon transcription inhibition. A, abundance of RNAPII isolated by iPOND-TMT (Fig. 1B). Cells were treated 2hr with TPL and DRB according to conditions used in the study. B, abundance of RNAPII isolated by iPOND-TMT 24hr after chase (Fig. 4H). Cells were treated 2hr with TPL and DRB according to conditions used in the study. C, RNAPII abundance on chromatin examined by QIBC after treatment for indicated time (x axis) with DRB, TPL, Actinomycin D (Act D) and flavopiridol (Flav).

R2.4.2 How can the author differentiate from accumulation of RNAPII in front of the fork, leading to purification by iPOND, and RNAPII on nascent DNA.

As mentioned above, because of fragment size generated in the iPOND protocol, we do not expect to isolate chromatin ahead of the fork (please see R2.3).

R2.4.3 Also, if the accumulation of RNAPII is on the nascent DNA, do the authors suggest that RNAPII gets loaded more on nascent DNA while under DRB inhibition or that stalled RNAPII are mainly by-passed by replication forks, leading to their enrichment on nascent DNA?

Our data suggest that in DMSO and DRB, RNAPII is loaded to similar extent on newly replicated chromatin (Fig. 1E). In later time points, while in DMSO, RNAPII remains on replicated chromatin, in DRB, RNAPII is unloaded. This is in good accordance with DRB blocking the elongation step and not the loading step of RNAPII.

R2.5. At p.14, the authors state: "Because they share the same DNA template, transcription is known to challenge replisome progression at high frequency, from RNAPII constituting a roadblock to progressing replisomes, to generate RNA:DNA hybrids (Berti et al., 2020). It is

therefore remarkable that behind replisomes, only a handful of DNA repair factors appear to be involved in response to RNAPII binding and elongation.". How does the fact that transcription represents a roadblock in front of the forks makes it remarkable that only a handful of DNA damage repair pathways are involved behind the fork (where they are not a roadblock to any replisome)?

Because of the diverse range of issues generated by transcription, we were expecting to detect a greater variety of DNA repair proteins at the replisomes. In the revised version of the manuscript, we have estimated the effect of each drug treatments on each repair pathways by measuring the mean \log_2 fold change for the treatment, and the statistical significance was estimated by gene set enrichment using a bootstrap (see Methods). In this analysis, upon TPL treatment most DNA repair pathways were significantly depleted from replicated chromatin (New Fig. 3B). We discuss this point in the revised result section.

R2.6. At p.11, the authors states: "As di and tri-methylations require several hours to be re-established following DNA replication (Alabert et al., 2015; Reveron-Gomez et al., 2018), 11 minutes after the passage of the fork, such increase most probably reflects an increase of H4K20me2 and H3K9me3 on recycled parental histones.". Can the authors extend their interpretation of this result? Do the authors think that DRB treatments increase methylation of histones in G1, prior to replication, or specifically in front of the fork (due to conflicts? DNA damage?), and that those methylated histones get recycled on nascent DNA?

In the revised version of the manuscript, we have simplified this section. The pre-revised version showed the histone modification levels in total chromatin, in nascent chromatin and their restoration kinetics. In the revised version we only focus on the restoration kinetic and the most robust differences between DMSO, DRB and TPL treatments. The full dataset is still available for the reader in Table EV6.

R2.7. Figure 4: The authors perform the histone PTM analysis under 0h (nascent chromatin) versus 2h (re-established chromatin) timepoints. It would have been insightful to also include a 1h timepoint in this experiment. There appear to be some trends/changes but they do not show statistical significance (e.g. H4K5K12ac or H3K14ac). It might be useful to increase the number of biological replicates (including the 1h timepoint) here, which could improve the confidence in the results and/or discover additional transcription-dependent changes of histone PTM restoration.

We could not increase the number of replicates for the histone modification experiments within the time frame of the revision. Therefore, we have amended the result section to focus on the restoration kinetic following the passage of the fork and the most robust differences between DMSO, DRB and TPL treatments.

Minor comments:

1. Fig3I: It would be nice to show a TF from the "Restored within 11 min" category as a comparison point.

We agree with Reviewer 3, but we did not manage to get an antibody specific for one of these TFs that work by immunofluorescence during the revision period. Future work includes exploring TF binding kinetics by ChOR-seq, as some of the antibodies we tested work by CHIP.

2. In page 14, th authors state: "However, we did not detect significant signs of DNA damage in DRB treated cells (Fig. 2A, 2B)". Which signs the authors looked at?

We have measured γ H2AX level on chromatin by QIBC (Fig. EV4B-C), on nascent chromatin by PLA between EdU and γ H2AX (Fig. 3C) and in whole cell extracts by western blot (Fig. EV1E).

3. In the iPOND experiment, which size of DNA fragments is achieved?

DNA fragments are typically 150 to 300bp, see point R2.3.

4. At p.14, the authors state: "Because they share the same DNA template, transcription is known to challenge replisome progression at high frequency, from RNAPII constituting a roadblock to progressing replisomes, to generate RNA:DNA hybrids (Berti et al., 2020)." The paper has not addressed the role of RNA:DNA hybrids in these processes.

We used the antibody that recognize RNA:DNA hybrid (Clone S9.6), and tested by high content microscopy its specificity. So far, in our hands, this antibody was not specific and will therefore require further optimization to explore this pathway by QIBC. One route will be to test different RNases treatments (Smolka et al., 2021).

5. Fig3D: Is there enough datapoints to state a conclusion?

We agree with Reviewer 2, too few points are present in each subcategory. We therefore have decided to remove it from the revised version of the manuscript.

6. S1A: mistakes in the x axis labels ("no EU" in a EdU quantification graph, "no EdU" in a EU quantification graph).

Thank you for pointing out this mistake.

7. S1F is not sufficiently described in the legend. It took me some time and additional efforts to understand what the right panel of S1F was showing.

Thank you for pointing this out. We agree with Reviewer 3 and decided to simplify this section in the revised version of the manuscript by presenting volcano plots in Figure 1 and quantification of protein affected by DRB and TPL as bar charts in Figure EV2.

8. S2E-F: are the axis wrong? Is it supposed to be Nascent when its comparing total extracts?

Thank you for pointing this out. Total extracts are collected at the nascent and the mature time points, providing information 2, 3 and 4hr after drug treatments. For simplicity in the revised version of the manuscript we show the overlap between protein significantly affected by DRB and TPL in iPOND and in whole cell extracts as a Venn diagram (New Fig. EV2F).

9. A lot of graphs have non-precise axis labels that needs reading of the manuscript and/or legends to understand. For example: 1K-L (distribution, %), 2L (% of the max), 3B-C-D \log_2 (Nascent/2h), 3G IP/Input, 4C (Inhibitor treatment/DMSO (%)), S2E-F (TPL/DRB Nascent/DMSO Nascent), S3A (IP/Input), S4A (No Y axis label).

Thank you for pointing this out, it has been corrected in the revised version of the manuscript.

10. FigS4: Assignment of colors in bar graphs of C-J to treatments is not shown. Heatmaps in H and K do not show if these are 0 or 2h. The heatmap in H shows H3 modification and the heatmap in K shows modification in H3.3 but the labels of the modification in K (except the first one) are the names of the modifications of H3, not H3.3. In the legend, GAPDH is written GABDH.

Thank you for pointing this out. We have simplified this figure (new Fig.2, EV3). The full dataset is available in Table EV6.

Reviewer #2 (Significance (Required)):

In this manuscript, the authors characterized the re-establishment of chromatin after DNA replication in fibroblasts using iPOND-MS. As mentioned above, the work provided by the authors feels mainly descriptive and incremental and therefore does not provide further mechanistic insights beyond the current state of the art. Some follow-up experiments to study the functional impact on the different enrichment patterns on nascent DNA or the function of the dependency on RNAPII for the reestablishment of steady-state enrichment on chromatin of some factors would have greatly increased the scientific impact of the manuscript. Nevertheless, the proteome of nascent DNA, its kinetic, and the effect of transcription inhibitors will provide interesting information and a useful resource for research groups in the DNA replication, chromatin, epigenetic, and DNA damage repair fields. Thus, in conclusion, I would recommend this manuscript to be published in its current state in a lower tier journal such as MBoC or PLOS ONE journals. If the authors can provide additional mechanistic insights by addressing at least a few of the specific points, I think it would become a stronger candidate for a journal with higher impact.

Reviewer #3 (Evidence, reproducibility and clarity (Required)):

In this study, the authors used metabolic labeling of newly-replicated or nascent chromatin followed by quantitative Mass spectrometry (iPOND-MS) to characterize protein composition of nascent chromatin at time points after DNA replication: immediately after a short pulse of EdU labeling (nascent), and after 1 and 2 hours of Thymidine chase (maturing chromatin). The iPOND method was established before but in the current manuscript the authors combined this with inhibiting RNA Pol II transcription at distinct stages to determine the effects on transcription and RNA Pol II cycle on chromatin protein dynamics at the wake of DNA replication. The inhibitors they used are Triptolide, which blocks transcription initiation and induces a proteasomal degradation of all chromatin bound RNA PolII, and DRB, which blocks transcription elongation causing an enrichment of paused RNA PolII. The authors compared the relative enrichment of ~1200 proteins on nascent and maturing chromatin and the effects on transcription inhibition on these proteins.

The authors found that RNA PolII does not affect the loading or retention of most histones on nascent chromatin except for the histone variant H2A.Z, which requires RNA PolII loading. However, DRB treatment (no elongation) resulted in stabilization of all histones (which the authors do not seem to catch on). Interesting, unlike the histone, both replication-coupled and -independent histone chaperons seem to be enriched immediately behind the fork and are affected by RNA PolII to different extents. They next look at ATP-dependent remodelers and find that most remodeler families are facilitated by RNA PolII loading, while elongation affects some remodeler families and not others. They see the same trend looking at a wide variety of transcription factors. Interestingly, while RNA PolII loading is required for the establishment of some histone post translational modifications (H3K36me3), some others such as H3K9me3 and H4K20me2 are negatively affected. Finally, the authors find that RNA PolII elongation promotes binding of several DNA repair proteins, and speculate that this is because of DNA damage from replication-transcription conflict.

R3.1 My main concern about this manuscript is that the relative enrichment of most factors show variability across the time points, which make the interpretation of the data difficult. This becomes more concerning when we look at protein complexes such as the ATP-dependent remodelers. Subunits of the same complex which are expected to bind together show different patterns of enrichment. This raises the concern as to how data was normalized.

Proteins that function as part of a stoichiometric protein complex should be captured equally in an iPOND-MS experiment. However, subunits of the same complex often exhibit different enrichments, including subunits of the replisome (See below, figure from David Cortez, *Methods in Enzymology*, 2017). This has been attributed to the different abundance in tryptic peptides between different subunits. This point has been added to the limitation section.

Fig. 5 Comparison of the enrichment of selected replisome proteins at replication forks calculated in five proteomic datasets. A log₂ transformation of the mean enrichment comparing fork/chromatin (pulse/chase) is depicted. Larger positive values indicate increased enrichment at forks compared to bulk chromatin. Error bars were calculated as SEM where possible. (...)

R3.2 Furthermore, how do the replicates compare to each other? The others selected ~1200 proteins which were enriched in all three replicates, but how does their relative enrichment compare in the replicates? The authors need to show some kind of comparison across replicates to confirm that the differential relative enrichments are real and biologically meaningful.

The PCA analysis show the comparison between all the biological replicates (Fig. EV1F). The volcano plots show the adjusted p-values providing a mean to identify statistically reproducible changes across biological replicates (e.g., Fig. 1F-G). Based on Reviewer 3 comment, we also now provide the fold enrichment for each individual biological replicate in the supplementary tables. This provides a mean for the reader to assess how biological replicates compare to each other for any of his/her protein of interest. Finally, we have developed an online tool that can be used to explore all the datasets presented in this study: https://shiny.compbio.dundee.ac.uk/public/marek_ncc_prot/de/. Here, the enrichment for each individual biological replicate is shown as a coloured dot in the feature plot window as shown for PCNA below.

R3.3 Also, the TF data is very descriptive. Insightful analysis of similarities/differences between types of TFs would be interesting.

We did not include this analysis in the manuscript as each sub-type of TFs is formed of only few TFs, limiting the relevance of the findings.

R3.4 Minor comment: The formaldehyde cross linking used in iPOND makes it difficult to interpret/distinguish what is actually chromatin bound versus what is enriched due to protein-protein cross linking. The authors should highlight that in the limitations section.

This point has been added to the limitations section in the revised version of the manuscript.

****Referees cross-commenting****

I agree with most of Reviewer 1's comments about the lack of proper controls and normalization, which make the interpretations difficult. Particularly all of the controls mentioned under point 1 should not be difficult to perform, and if included, would strengthen the study and the manuscript.

All the controls required by Reviewer 1 are provided page 3-7.

Reviewer 1 makes an important point about normalization, which I totally agree with. Ideally, a spike-in approach would help obtain a much more quantitative and reliable understanding of

differential protein enrichment. However, repeating all iPOND experiments with spike-in might be a big ask.

We would like to thank Reviewer 3 for his/her understanding. As mentioned in R1.2.1 point 7, to our knowledge, while spike-in is routinely used to normalize EdU-seq experiments, no such spike-in exist to quantify EdU based proteomic dataset.

What the authors could do at minimum is show how replicates compare with each other. It looks like they pooled all three replicates for analysis, but comparing relative enrichment of all 1257 proteins across replicates would help.

Please see point R3.2.

The point about delayed histone occupancy is a critical one and difficult to rationalize. To note, histone chaperons are enriched on nascent, but histones are not.

Although histones and histone chaperones form a complex on newly replicated chromatin, their relative enrichment during the maturation process is not expected to be similar. Histone chaperone such as CAF-1 are expected to be enriched on newly replicated chromatin compared to mature samples, mirroring the kinetic of replisome proteins. Histones on the other hand are expected to be low abundant behind replisomes compared to later time points.

Histone and histone chaperones kinetics. **A.** Relative enrichments of histones and replication coupled histone chaperones from iPOND-MS dataset. **B.** Rational of the findings. Expected changes in relative enrichment of histone and replication coupled histone chaperones are shown below.

Besides, in the current way that the data is analyzed and presented, there are a lot of fluctuations in protein enrichment across the 1-2 hour timepoints of chromatin maturation, which would be very interesting if real. For e.g., Fig. 11, Triptolide treatment, most of the cluster I and cluster II proteins show medium-high enrichment on nascent, depleted in 1h, but recover in 2h. If the binding/recruitment of these proteins on newly-replicated chromatin is RNA Pol II dependent, why would they come back after 1h? If this real, this would be very interesting.

We agree with Reviewer 3, and as we have no indication that one experiment should be removed from the analysis, our data support that chromatin maturation is a dynamic process with potential transient binding events. To increase the robustness of our observations, we have performed an additional biological replicate for a total of n=4.

There are several additional examples of problems with quantification/normalization. As for SWI/SNF subunits, both SMARCA4 and SMARCC1 are core subunits and based on several thorough biochemical studies, cannot be expected to bind separately. However, they show different kinetics in DMSO as well as TPL and DRB.

SMARCA4 and SMARCC1 do show different kinetics, but these differences are not significant.

SMARCA4 and SMARCC1 binding kinetics in DMSO, TPL and DRB. Mean of 4 biological replicates and standard deviation are shown.

Another problem of the assay is that it shows genome-wide average. As Reviewer 1 rightly pointed out, transcription inhibition could disproportionately affect chromatin maturation kinetics in different genomic regions. Perhaps it would be interesting to analyze sets of genomic regions separately, such as highly transcribed and lowly transcribed genes. This might be achieved by adding a purification step using pools of DNA sequence probes before or after the streptavidin enrichment.

We have used replication timing as a mean to compare chromatin dynamic in highly transcribed regions (euchromatin) and low transcribed regions (heterochromatin) (Alvarez et al. 2023, Cell reports, Figure 4). As this strategy requires cell cycle synchronisation or cell sorting, we decided to avoid exposing cells to additional stress. Re-CHIP strategies such as CHOR-seq would be the best approach to tackle this question in a future study.

Additional comment: The formaldehyde cross linking used in iPOND makes it difficult to interpret/distinguish what is actually chromatin bound versus what is enriched due to protein-protein cross linking. The authors should highlight that in the limitations section. This is an important point, and it has been added to the limitations section.

On a positive note, it is a very important and timely study and the manuscript has a lot to consider. Addition of proper controls and normalization/analysis of replicates will make it stronger.

We hope that the revised version of the manuscript will satisfy Reviewer 3.

Reviewer #3 (Significance (Required)):

Overall, it is a very important and timely study, and the manuscript has a lot to consider. There are several recent papers on the kinetics of chromatin maturation behind the replication fork, and this study adds a very important dimension to this ongoing investigation, and will be of interest to a broad readership in the chromatin and transcription field.

We are very happy Reviewer 3 acknowledges the relevance of our work.

Dear Constance,

Thank you for the submission of your revised manuscript. We have now received the enclosed reports and cross-comments from the referees. Referee 3 still has a few concerns that I would like you to address and incorporate before we can proceed with the official acceptance of your manuscript.

Referee 3's concerns can be addressed along the lines suggested by referees 1 and 2 (see below). A PLA assay can be added, if it is straightforward. A peptide spike-in iPOND is not required. Please co-submit a point-by-point response to all last concerns and address all concerns in the ms text.

A few editorial requests will also need to be addressed:

- The ms has 4 main figures but the results and discussion sections are not combined. Please either add 2 more main figures, or combine the results and discussion sections to publish your ms as a short report. The character count should not exceed 29,000 characters (including spaces but excluding references and materials and methods) for a short report.
- Please add a "Disclosure and Competing Interest Statement".
- Please remove the author credits from the ms file. All credits need to be entered in our online system during ms submission.
- Please correct the REFERENCE FORMAT: et al should be used after 10 author names; year should be in brackets. The EMBO reports reference format can be found in EndNote.
- Please complete the statistics part on the author checklist and send us a new, completed list.
- Please enter all funding info also in our online submission system, several grants are currently missing.
- Tables EV1-EV8 should be uploaded as Datasets; they should be renamed to Dataset EV1-EV8 and the callouts should also be updated; the legends need to be removed from the ms and inserted in each CSV file as a separate tab/sheet.
- If possible, I would like to suggest to modify the synopsis image. The current image is not self-explanatory.
- I would like to suggest some changes to the short summary and bullet points:

RNAPII promotes the stabilisation of chromatin remodelers, transcription factors and histone modifiers on newly replicated chromatin, a function that is not observed in steady state chromatin.

- RNAPII promotes the re-association of hundreds of proteins with newly replicated DNA. This includes several chromatin remodelers, transcription factors and histone modifiers.
- The function of RNAPII on newly replicated chromatin is not observed on steady state chromatin.
- Blocking transcription reduces the recruitment of DNA repair proteins to nascent chromatin in human cells. [Or is it the maintenance instead of the recruitment? If it is unclear, may be use "binding"?]
- Nucleosome assembly and the re-establishment of most histone modifications are uncoupled from transcription.

- SUPPLEMENTARY FIGURE LEGENDS should be renamed to EXPANDED VIEW FIGURE LEGENDS

- The manuscript sections should be in the following order: Title page - Abstract & Keywords - Introduction - Results - Discussion - Materials & Methods - Data Availability - Acknowledgments - Disclosure Statement & Competing Interests - References - Figure Legends - Tables with legends - Expanded View Figure Legends.

- Please address these figure legends comments:

1. Please define the annotated p values ****/**/* in the legend of figure 1c; 2d-e; 3c; 4f i; EV1a, j; EV4a-c; EV5b-e, h as appropriate.
2. Please indicate the statistical test used for data analysis in the legends of figures 1f-i; 2e; 3a-d, f, i; 4f, i; EV3b-c; EV4b-d, h.
3. Please note that information related to n is missing in the legends of figures 2d-e; 3e; 4i; EV1d, g; EV2a, f-h; EV3b-c; EV5g.
4. Please note that the error bars are not defined in the legends of figures 3e; EV1d; EV2a, f-h; EV3b-c.

I would like to suggest some changes to the title and abstract. Please let me know whether you agree with the following:

RNA polymerase II promotes the organization of chromatin following DNA replication.

or
RNA polymerase II stabilizes chromatin organization following DNA replication.

Understanding how chromatin organisation is duplicated on the two daughter strands is a central question in epigenetics. In mammals, following the passage of the replisome, nucleosomes lose their defined positioning and transcription contributes to their re-organisation. However, whether transcription plays a greater role in the organization of chromatin following DNA replication remains unclear. Here we analyze protein re-association with newly replicated DNA upon inhibition of transcription using iPOND coupled to quantitative mass spectrometry. We show that nucleosome assembly and the re-establishment of most histone modifications are uncoupled from transcription. However, RNAPII acts to promote the re-association of hundreds of proteins with newly replicated chromatin via pathways [does the ms provide insight into which pathways? If not, please delete.] that are not observed in steady state chromatin. These include ATP-dependent remodellers, transcription factors and histone methyltransferases. We also identify a set of DNA repair factors that may handle transcription-replication conflicts during normal transcription in human non-transformed cells. Our study reveals that transcription plays a greater role in the organization of chromatin post-replication than previously anticipated.

With my best wishes, also for the festive season,

Esther

Referee #1:

The revised manuscript is considerably improved, and the authors have adequately addressed the reviewers' concerns with appropriate changes, new figures, and text. One comment I still have is that it will be very informative to provide a clustering of sorts of all proteins showing changes in enrichment, to get an overall idea of related/corresponding and non-related/discordant changes.

Other minor comments:

1. In the abstract, what do the authors mean by "whether transcription plays a greater role"? Greater than what?
2. Cite Fenstermaker et al 2023 (PMID: 37468626) more appropriately as they showed RNA PolII occupancy immediately after the fork using PLA assays.
3. I noticed a few typos: "pioneers" factor; iSWI should be ISWI; INO80/SWR should be INO80/SWR1.

Referee #2:

The authors have addressed most of the questions raised by three reviewers by providing proper controls, additional experiments, and reasonable explanations. The issue raised by reviewer 1 regarding the normalization method is critical and the authors have provided data showing that most of the proteins in the compared samples have similar abundances except for a small percentage of proteins. The additional spike-in control will strengthen the study, but it seems technically not feasible to achieve for now.

The question about how transcription inhibition affects the proteome of newly synthesized and mature DNA is of great interest to researchers interested in chromatin dynamics behind the replication fork. Although no further mechanistic insights are provided, the description of the proteome changes behind the fork upon global transcription inhibition is an important dataset and resource for the community that should be published as soon as possible.

Referee #3:

The authors have adequately addressed issues R1.1.1, R1.1.2 and R1.1.3. The other points I raised have only been partially addressed, as detailed below.

R1.1.4:

The graph in Figure 1D is not informative: it looks like there are more cells with an average EdU intensity just below the median in TPL and DRB treatments compared to DMSO. The authors should divide the datasets into bins of EdU intensity and compare cell densities between DMSO and TPL/DRB treatments in each bin. The t-test is not the best test for this assay as it compares

the means of normally distributed datasets. Since the distribution of the whole cell population is clearly not Gaussian, a K-S test that compares distribution shapes would be more appropriate. From visual inspection of the graphs it looks like TPL/DRB treatment decreases EdU incorporation to some extent, but it is difficult to judge the significance of the effect with the current representation and quantification.

While the graphs in Fig. EV1C do qualitatively show that DNA replication has progressed in all three cases, the graphical representation is not quantitative, which makes it difficult to assess whether TPL/DRB may have a partially adverse effect on genome replication. The scales of the y axes are different between the three graphs (the font of the numbers in both axes is too small), making it hard to compare them. It is nevertheless obvious that the 2N peak of the black curve is higher in DMSO than in DRB or TPL, suggesting that more cells have finished replication 2hrs after the EdU pulse in DMSO than in DRB/TPL and consequently suggesting that replication progression is slower in DRB/TPL. Their claim that DRB/TPL does not affect replication progression would be more convincing if they showed the following areas under the curves for each graph: 1. the overlap between the red (Nas, nascent chromatin) and black (2hr, matured chromatin) curves. 2. the area under the red curve that does not overlap with the black curve, and 3. the area under the black curve that does not overlap with the red curve. Also, if EdU incorporation had indeed been completely stopped by addition of equimolar amount of Thymidine after the EdU pulse, as they claim, the total areas under the red and black curves should be the same in each graph (i.e. the number of EdU+ cells should not increase after the chase). Is this in fact the case? A bigger area under the black curve compared to the red curve means that EdU has continued to be incorporated into cells after the chase at least to some extent, which would further complicated the interpretation of their results.

Finally, I am assuming that the microscopy experiment in Fig. EV1D was designed to address this point: "Transcription inhibition could disproportionately affect the replication of some parts of the genome more than others and since there is no evidence to the contrary the differences that they observe between the TPL/DRM treated and DMSO treated proteomes bound to replicated DNA could just be because they were isolated from different genomic loci." Unfortunately, the experimental design is not described in enough detail for me to assess if S-phase progression is indeed the same in TPL/DRB treated cells compared to DMSO cells. Were cells treated with DMSO/TPL/DRB for 2hrs before the EdU pulse as shown in Fig 1.B? Were cells fixed and imaged right after the pulse (the N time point) or 1hr or 2hrs later? Which time point is shown in the figure? All three time points should be shown, in any case: if the deleterious effect of TPL/DRB on replication worsens with time (Skalska et al., 2021), one would expect progressive accumulation of cells in S-phase and a decrease of the G2 cell fraction over time. It would therefore also be useful to show a later time point (6hrs, 8 hrs, 12hrs?) to find out at which point after TPL/DRB addition cells stop replicating. Even with those additional time points and even if they demonstrate that S-phase fractions are comparable between TPL/DRB and DMSO treatments 2hrs after the EdU pulse, this would still be a bulk measurement of S-phase progression and would not show whether replication of some genomic loci is more sensitive to TPL/DRB treatment, which was my main concern here. They need to show that all genomic loci are replicated at similar levels at the N time point in all treatments with genome-wide sequencing of EdU labelled DNA. They should also show that genome-wide replication dynamics (fork speed, replication timing) are not altered 2hrs (the N time point), 3hrs (the 1hr maturation time point) and 4hrs (the 2hrs maturation time point) after TPL/DRB treatment.

R1.1.5

The data in Fig EV1B needs to be quantified. It does look like bulk EdU intensity of EdU+ cells does not go up between the two time points, as expected if EdU incorporation slows down and stops after the pulse, but a potentially smaller increase in EdU incorporation is impossible to see with this kind of representation. The data should be represented as EdU intensity density plots (violin plots) of cell count bins at 1N, 1N to 2N and 2N for Nascent and Mature time points. If EdU incorporation truly stops after the chase, cell density should increase from 1N to 2N between Nascent and Mature time points, while their respective EdU intensities should be similar.

Also, the experiment should be repeated with more intermediate time points with and without Thymidine chase to check how much and when the Thymidine chase contributes to slowing down and/or stopping EdU incorporation as opposed to EdU being progressively used up during the time-course. This is important because it is essential that EdU incorporation be stopped as rapidly as possible in order to be able to conclude that the changes in iPOND proteome composition are a consequence of chromatin maturation.

R1.2.1

"Yet, in our iPOND and NCC datasets such differences have never been observed across biological replicates, whether we use replication stress agents (HU, CPT, ATMi), translation inhibitor (CHX), or here transcription inhibitors (TPL, DRB) (Panel A)."

This claim does not make a lot of sense. It is impossible to conclude that there are no biological differences between samples solely based on bulk protein levels, even if they are isolated from the same number of cells. As panel A, shows there is quite a bit of variability in total protein levels between samples and it is obviously impossible to tell whether the cause of the variability is experimental or biological.

"This method was successful for NCC-MS (Alabert et al. 2014) but proven to be problematic for iPOND-MS. In NCC (20 minbiotin-dUTP labelling), the amount of histone detected on nascent and mature samples moderately changes. In iPOND (11min EdU labelling) it greatly increases (Fig. EV1H). »

Isn't it worrisome to the authors that two different approaches (NCC-MS and iPOND) that should in theory give a similar result, differ so widely in the amount of histones recovered on nascent chromatin? While it may be reasonable to expect some drop in histone occupancy on nascent DNA compared to mature chromatin, it is certainly not reasonable to expect that practically no histones are recovered with iPOND in the nascent time point (Fig. EV1H). EdU is pulsed for 11 min, which means that, assuming fork speeds of 1 to 2 kbp/min, there should be stretches of 11 to 22kbps of EdU labeled DNA throughout the genome in every cell. According to their results, these 10-20kbps stretches should be devoid of nucleosomes immediately after the

passage of the fork? This goes against all previously published observations including their own (Alabert et al., 2014). The authors should consider the possibility that there is some unresolved technical issue with their iPOND protocol.

"However, normalising iPOND mass spectrometry dataset using a 'spike in' protein, to allow to correct for sample preparation variation, requires a synthetic peptide or a protein clearly distinguishable from the human homologue and that binds EdU labelled DNA. To our knowledge, there is no known protein that could be used for this purpose, including histones due to the high degree of histone sequence conservation."

Obviously, using cells of a different species as a spike-in control, like it is done for iPOND sequencing, is impossible with proteomics experiments, as the authors point out. That is however not what I was suggesting. I was actually suggesting to add a dozen or so synthetic peptides (that are not found in human cells) at a constant mass to mass ratio of total protein versus synthetic peptide to all samples before doing the mass-spec analysis. That way all samples would be normalized to the same external standard. This would make it easier to compare time points and treatments between themselves.

R1.2.4.

Why only show a PLA assay for ZNF462, whose binding is independent of RNAPoIII? Why not also show a PLA assay for a TF whose binding depends on PolII?

R1.2.5

As explained above (R1.2.1), considering the experimental design and the results from EV1H, one would expect that 10 to 20 kbps of nascent DNA will be devoid of nucleosomes genome-wide. This contradicts the authors' statement that only 300bp of DNA is nucleosome free behind the fork.

R1.2.6

"The systematic comparison of proteins bound to newly replicated chromatin and steady state chromatin revealed that many RNAPII dependent effects are specific to newly replicated chromatin."

This may be true, but it still does not answer my question: why is RNAP2 enriched in the nascent and 1hr time points but then becomes depleted in the 2hr time point in the DRB treatment since global RNAPII levels don't change in the DRB treatment compared to DMSO (Figure 1C)? Can the authors provide an explanation in the discussion beyond "that's what happens on replicated chromatin"?

Also, the results shown in Fig. 4G raise a new question: why is steady state chromatin after BRD treatment different from whole cell extract? Shouldn't they be the same if BRD does not interfere with replication? Otherwise, the observed differences between steady state and whole cell extract and the fact that DRB is added after and before EdU, respectively, in the steady state and whole cell extract samples, suggests that DRB might interfere with replication.

R1.2.7, R1.2.8 and R1.2.9

"Moreover, in human, histone methylations are not expected to double within 2h, requiring instead 4h to 20h, depending on the histone residue (Alabert et al. 2015, Reveron et al. 2018). Histone modifications expected to decrease within 2h are H4K5K12ac and are behaving accordingly in this dataset (Fig. 2D). This dataset is therefore in good accordance with published proteomic analysis of histone modifications."

While it is certainly reasonable to have certain expectation based on previously published data, I don't have to tell these authors that the iPOND experiment in this work is not exactly equivalent to the experiments in Alabert et al. and Reveron et al. In both of these works nascent chromatin was isolated after synchronization in early S-phase, so proteins were isolated from early replicating chromatin. In this work nascent chromatin is isolated from asynchronous cells so all genomic loci should be represented (early and late replicating). It is therefore reasonable to assume that there might be some differences in the dynamics of histone PTM reestablishment between the two sets of experiments, especially for H3K27me3 and H3K9me123, which tend to be in regions that replicate late. This should be pointed out and later time points should be added (4, 6 or 8hrs?) at least for the DMSO treatment, although if they do the experiment I asked for in R1.1.5, they could add a later time point for the DRB and TPL treatments as well.

Also, it is shown in Reveron et al., that H3K4me3 is restored in 50% of early replicating regions within 1 hr. Consequently, PTM recovery dynamics observed here are not quite in accordance with published results. Wouldn't it be reasonable to expect that H3K36me2/3 follows similar kinetics in the DMSO treatment since it is also a transcription dependent PTM? Instead, Figure 2E shows that H3K36me2 goes up 5 fold (??) from 5% to 25% within 2hrs and H3K36me3 only goes up a few percentages. H3K36me2/3 are surprisingly low in the N time point, suggesting that for some reason (maybe the click reaction is not very efficient on chromatin with recycled nucleosomes) iPOND enriches for newly synthesized histones and does not recover recycled histones at least in the N time point. This would explain why nascent chromatin appears depleted of nucleosomes in this time point, if new nucleosome assembly on nascent DNA happens sometime after old nucleosomes recycling. If iPOND does indeed enrich for new nucleosomes, this would also explain why H3K36me2 goes up 5 fold instead of the expected 2 fold or less.

Clearly, it is difficult to explain all these discrepancies, when it is not known if iPOND recovers old and new histones in equal proportions.

Unfortunately, even though I commend the authors for their necessary and valiant attempt at understanding how transcription influences chromatin maturation after replication, I cannot recommend the work for publication due to the lack of controls (listed above) that would have made their conclusions more convincing.

Cross-comments from referee 2:

R1.1.4 Although I agree with Reviewer #3 on Figure 1D that the t test is not appropriate based on the non-Gaussian distribution of EdU+ cells and should be replaced by a more appropriate statistical test, the two distributions do not show "major differences"

in their EdU incorporation rates, suggesting that there is no global effect on DNA replication upon TPL or DRB treatment in the given timeframe. The FACS histograms in Fig. EV1 do also show similar distributions between DMSO, DRB and TPL treated nas. vs. 2h cells so I agree with the authors' conclusion that all presented data do not show a major global impact of the TRX inhibitors on DNA replication. The reviewer is right that this does not exclude the possibility that there might be local differences at individual genes/areas of the genome but the whole paper is based on bulk analysis of the proteome changes averaging individual differences from individual cells and loci so the reported changes reflect the global average of proteomic changes concurrent with the global EdU patterns.

R1.1.5 I agree that the requested quantification would be an valuable addition to the manuscript.

R1.2.1 The reviewer makes an interesting point about the different enrichments of histones between the NCC and iPond methodologies. I do believe that this discrepancy can be explained by the different methodological approaches where NCC is based on incorporating biotin-dUTP and then direct streptavidin pulldown whereas iPond is based on an EdU Click-it reaction to conjugate biotin to nascent DNA. It is possible that the required Click-it reaction in iPond may shear the nascent DNA or affect the nucleosomal density. Nevertheless, I consider both NCC and iPond as complementary approaches to detect the proteomic makeup of nascent chromatin, both of which have been widely used in the field and gave important biological insights. Thus, I do think the authors should discuss this point in the manuscript but I do not think the reported discrepancy of both methods makes their results invalid. For the internal spike-ins, the authors could attempt to normalize on other background proteins detected in all mass spec experiments for example on ribosomal proteins as these are highly abundant proteins usually not associated with replication forks and present in all MS datasets.

R1.2.4.

The PLA assay can be added but I do not see this as a crucial and important point for the manuscript.

R1.2.5

The differences may indeed be because of the different methodology and should be discussed in the manuscript as mentioned above.

R1.2.6

DRB treatment might block RNAPII on promoters and therefore affect fork progression in the given timeframes. Indeed a study by Fenstermaker et al., 2023 reported rapid re-association of RNAPII in the wake of the replisome on immature chromatin and rapid resumption of transcription. The authors could discuss their findings and the observed differences with DRB in light of this study.

R1.2.7, R1.2.8 and R1.2.9

The reviewer raises again a good point and in the end it goes back to the potential differences in used methodologies between NCC and iPOND. I do think that the observed discrepancies need to be reported and discussed in the manuscript, but I do not agree that this would invalidate the author's data and findings. I do believe that elaborating on these issues in the discussion would be adequate and helpful to the readers.

Cross-comments from referee 1:

It appears to me that several of R3's concerns could be addressed by revising the text. The authors should consider revising the limitations section to address issues concerning EdU incorporation in TPL/DRB treatment (R1.1.4, and R1.1.5), along with performing the additional data analysis and control experiments suggested by the reviewer. Also, the PLA assay to test RNAPII and a TF's co-occupancy (R1.2.4.) should be doable within a reasonable time frame. The suggested peptide spike-in iPOND is essentially asking the authors to replicate the entire study with spike-in normalization, which demands time and resources, particularly if the authors don't end up finding major differences with/without spike-in.

In summary, the study does have limitations, some of which have been discussed, and more could be addressed with careful data analysis and control experiments and then discussed in the text. However, this should not unreasonably delay its publication. As R2 rightly commented, "the description of the proteome changes behind the fork upon global transcription inhibition is an important dataset and resource for the community". I'd stress upon being more rigorous in analyzing and discussing the limitations, which should address several of R3's concerns.

Referee 1 (former referee 3):

The revised manuscript is considerably improved, and the authors have adequately addressed the reviewers' concerns with appropriate changes, new figures, and text. One comment I still have is that it will be very informative to provide a clustering of sorts of all proteins showing changes in enrichment, to get an overall idea of related/corresponding and non-related/discordant changes.

We provide a clustering of all proteins showing significant changes in the revised version of the manuscript as Fig EV2C-D. We would like to thank Referee 1 for all his/her insightful comments.

Other minor comments:

1. In the abstract, what do the authors mean by "whether transcription plays a greater role"? Greater than what? We mean greater than its published role in repositioning nucleosomes.
2. Cite Fenstermaker et al 2023 (PMID: 37468626) more appropriately as they showed RNA PolII occupancy immediately after the fork using PLA assays. We have added this point in the introduction (page 2): "RNA polymerase II (RNAPII) is recruited to replicated DNA (Bruno et al, 2023; Fenstermaker et al, 2023; Stewart-Morgan et al., 2019)"
3. I noticed a few typos: "pioneers" factor; iSWI should be ISWI; INO80/SWR should be INO80/SWR1. Thank you for pointing out these typos. This has been corrected in the revised version of the manuscript.

Referee 2:

The authors have addressed most of the questions raised by three reviewers by providing proper controls, additional experiments, and reasonable explanations. The issue raised by reviewer 1 regarding the normalization method is critical and the authors have provided data showing that most of the proteins in the compared samples have similar abundances except for a small percentage of proteins. The additional spike-in control will strengthen the study, but it seems technically not feasible to achieve for now.

The question about how transcription inhibition affects the proteome of newly synthesized and mature DNA is of great interest to researchers interested in chromatin dynamics behind the replication fork. Although no further mechanistic insights are provided, the description of the proteome changes behind the fork upon global transcription inhibition is an important dataset and resource for the community that should be published as soon as possible.

We thank Referee 2 for all his/her insightful comments.

Referee 3:

The authors have adequately addressed issues R1.1.1, R1.1.2 and R1.1.3. The other points I raised have only been partially addressed, as detailed below.

R1.1.4:

The graph in Figure 1D is not informative: it looks like there are more cells with an average

EdU intensity just below the median in TPL and DRB treatments compared to DMSO. The authors should divide the datasets into bins of EdU intensity and compare cell densities between DMSO and TPL/DRB treatments in each bin. The t-test is not the best test for this assay as it compares the means of normally distributed datasets. Since the distribution of the whole cell population is clearly not Gaussian, a K-S test that compares distribution shapes would be more appropriate. From visual inspection of the graphs it looks like TPL/DRB treatment decreases EdU incorporation to some extent, but it is difficult to judge the significance of the effect with the current representation and quantification.

Fig. 1D reveals non-normal distribution of log EdU mean intensity, therefore a non-parametric Mann-Whitney test was used to compare the different treatments. The Mann-Whitney test is appropriate as the null hypothesis states that probabilities of $X > Y$ and $X < Y$ are equal. Of note, the null hypothesis of the Kolmogorov-Smirnov test that allows to assess changes in the shape of the distribution also shows no significant difference between DMSO and TPL/DRB. The full name of the test used has been added to the legend of Figure 1D in the revised version of the manuscript.

While the graphs in Fig. EV1C do qualitatively show that DNA replication has progressed in all three cases, the graphical representation is not quantitative, which makes it difficult to assess whether TPL/DRB may have a partially adverse effect on genome replication. The scales of the y axes are different between the three graphs (the font of the numbers in both axes is too small), making it hard to compare them.

The scale of the graphs has been updated in the revised version of the manuscript.

It is nevertheless obvious that the 2N peak of the black curve is higher in DMSO than in DRB or TPL, suggesting that more cells have finished replication 2hrs after the EdU pulse in DMSO than in DRB/TPL and consequently suggesting that replication progression is slower in DRB/TPL. Their claim that DRB/TPL does not affect replication progression would be more convincing if they showed the following areas under the curves for each graph: 1. the overlap between the red (Nas, nascent chromatin) and black (2hr, matured chromatin) curves. 2. the area under the red curve that does not overlap with the black curve, and 3. the area under the black curve that does not overlap with the red curve.

The area from the flow cytometry data were extracted and the percentage of overlap between nascent and mature curves are shown in the revised version of the manuscript (Fig. EV1C). The data revealed that nascent and mature area overlapped at 62% in DMSO, 59% in TPL, and 65% in DRB. Therefore, in all 3 conditions, cells have progressed similarly in S phase.

Also, if EdU incorporation had indeed been completely stopped by addition of equimolar amount of Thymidine after the EdU pulse, as they claim, the total areas under the red and black curves should be the same in each graph (i.e. the number of EdU+ cells should not increase after the chase). Is this in fact the case? A bigger area under the black curve compared to the red curve means that EdU has continued to be incorporated into cells after the chase at least to some extent, which would further complicated the interpretation of their results.

The efficiency of the chase has been further examined in R1.1.5 (see below).

Finally, I am assuming that the microscopy experiment in Fig. EV1D was designed to address this point: "Transcription inhibition could disproportionately affect the replication of some

parts of the genome more than others and since there is no evidence to the contrary the differences that they observe between the TPL/DRM treated and DMSO treated proteomes bound to replicated DNA could just be because they were isolated from different genomic loci." Unfortunately, the experimental design is not described in enough detail for me to assess if S-phase progression is indeed the same in TPL/DRB treated cells compared to DMSO cells. Were cells treated with DMSO/TPL/DRB for 2hrs before the EdU pulse as shown in Fig 1.B? Were cells fixed and imaged right after the pulse (the N time point) or 1hr or 2hrs later? Which time point is shown in the figure? We apologise for not providing clearer description of the experimental design. Cells were treated, as for the iPOND-mass spectrometry experiments, 2hr with DMSO/TPL/DRB before been EdU pulsed for 20 min, fixed and analysed by microscopy. The legend of the panel has been updated accordingly in the revised version of the manuscript.

All three time points should be shown, in any case: if the deleterious effect of TPL/DRB on replication worsens with time (Skalska et al., 2021), one would expect progressive accumulation of cells in S-phase and a decrease of the G2 cell fraction over time. It would therefore also be useful to show a later time point (6hrs, 8 hrs, 12hrs?) to find out at which point after TPL/DRB addition cells stop replicating. In the mass spectrometry experiment, we have treated cells with DMSO/TPL/DRB for 2hr before labelling cells with EdU. Therefore, we show the distribution of EdU patterns right after the EdU labelling to assess whether drug treatments have perturbed the DNA replication programs. Later time points do not reflect how drug treatments may affect EdU labelling.

Even with those additional time points and even if they demonstrate that S-phase fractions are comparable between TPL/DRB and DMSO treatments 2hrs after the EdU pulse, this would still be a bulk measurement of S-phase progression and would not show whether replication of some genomic loci is more sensitive to TPL/DRB treatment, which was my main concern here. They need to show that all genomic loci are replicated at similar levels at the N time point in all treatments with genome-wide sequencing of EdU labelled DNA. They should also show that genome-wide replication dynamics (fork speed, replication timing) are not altered 2hrs (the N time point), 3hrs (the 1hr maturation time point) and 4hrs (the 2hrs maturation time point) after TPL/DRB treatment. We discuss the limitation regarding the analysis of bulk chromatin and the effect of TPL and DRB on EdU labelling, in the limitation section of the revised version of the manuscript, page 12.

R1.1.5

The data in Fig EV1B needs to be quantified. It does look like bulk EdU intensity of EdU+ cells does not go up between the two time points, as expected if EdU incorporation slows down and stops after the pulse, but a potentially smaller increase in EdU incorporation is impossible to see with this kind of representation. The data should be represented as EdU intensity density plots (violin plots) of cell count bins at 1N, 1N to 2N and 2N for Nascent and Mature time points. If EdU incorporation truly stops after the chase, cell density should increase from 1N to 2N between Nascent and Mature time points, while their respective EdU intensities should be similar.

Based on above suggestion, we have plotted EdU intensities as density plots of cell bins from Early (1N) and late S phase (2N) cells. Regarding cell density, as predicted by reviewer 1 if EdU incorporation stops, cell density for the 2N population increases from 45% in nascent to

94% in mature. This observation further confirms that the thymidine chase is effective. Regarding EdU intensities, EdU intensities decrease upon maturation for 1N cells and increase for 2N cells. A flawed chase would have led to an increase in EdU intensities for 1N and 2N cells. Therefore, these variations do not suggest a continuous incorporation of EdU during the chase period. These quantifications have been added to the manuscript as new EV1B.

Quantification of cell density (expressed as %) and EdU intensities (a.u.) for 1N and 2N populations from the experiment described in EV1B.

Also, the experiment should be repeated with more intermediate time points with and without Thymidine chase to check how much and when the Thymidine chase contributes to slowing down and/or stopping EdU incorporation as opposed to EdU being progressively used up during the time-course. This is important because it is essential that EdU incorporation be stopped as rapidly as possible in order to be able to conclude that the changes in iPOND proteome composition are a consequence of chromatin maturation. We provide several controls (EV1C-D) supporting that EdU incorporation is chased by the thymidine treatment using a 1/1 ratio EdU / Thymidine concentration. It has also been used by others (Dungrawala and Cortez, 2015, *Methods Mol Biol*, Wessel et al. 2019, *Cell Reports*), including the method paper published by the lab that has developed the iPOND protocol (Sirbu et al. 2012, *Nature protocol*).

R1.2.1

"Yet, in our iPOND and NCC datasets such differences have never been observed across biological replicates, whether we use replication stress agents (HU, CPT, ATMi), translation inhibitor (CHX), or here transcription inhibitors (TPL, DRB) (Panel A)."

This claim does not make a lot of sense. It is impossible to conclude that there are no biological differences between samples solely based on bulk protein levels, even if they are isolated from the same number of cells. As panel A, shows there is quite a bit of variability in total protein levels between samples and it is obviously impossible to tell whether the cause of the variability is experimental or biological.

"This method was successful for NCC-MS (Alabert et al. 2014) but proven to be problematic for iPOND-MS. In NCC (20 min biotin-dUTP labelling), the amount of histone detected on nascent and mature samples moderately changes. In iPOND (11 min EdU labelling) it greatly increases (Fig. EV1H). »

Isn't it worrisome to the authors that two different approaches (NCC-MS and iPOND) that should in theory give a similar result, differ so widely in the amount of histones recovered on

nascent chromatin? While it may be reasonable to expect some drop in histone occupancy on nascent DNA compared to mature chromatin, it is certainly not reasonable to expect that practically no histones are recovered with iPOND in the nascent time point (Fig. EV1H). EdU is pulsed for 11 min, which means that, assuming fork speeds of 1 to 2 kbp/min, there should be stretches of 11 to 22kbps of EdU labeled DNA throughout the genome in every cell. According to their results, these 10-20kbps stretches should be devoid of nucleosomes immediately after the passage of the fork? This goes against all previously published observations including their own (Alabert et al., 2014). The authors should consider the possibility that there is some unresolved technical issue with their iPOND protocol.

It is not worrisome as these datasets are generated from experiments with different labelling durations. A section covering NCC and iPOND differences regarding histones and other proteins has been added in the discussion of the revised version of the manuscript, page 10.

"However, normalising iPOND mass spectrometry dataset using a 'spike in' protein, to allow to correct for sample preparation variation, requires a synthetic peptide or a protein clearly distinguishable from the human homologue and that binds EdU labelled DNA. To our knowledge, there is no known protein that could be used for this purpose, including histones due to the high degree of histone sequence conservation."

Obviously, using cells of a different species as a spike-in control, like it is done for iPOND sequencing, is impossible with proteomics experiments, as the authors point out. That is however not what I was suggesting. I was actually suggesting to add a dozen or so synthetic peptides (that are not found in human cells) at a constant mass to mass ratio of total protein versus synthetic peptide to all samples before doing the mass-spec analysis. That way all samples would be normalized to the same external standard. This would make it easier to compare time points and treatments between themselves.

Using synthetic peptides added just before running the mass spectrometry to normalise the dataset won't correct the potential variability introduced by all the steps taking place before the mass spectrometry analysis. The experiment suggested will therefore not in our opinion solve the normalisation points raised by Reviewer 1. On the other hand, we have worked with a bioinformatician from the Computational Department at the University of Dundee to optimise the data analysis pipeline. We also have tested different normalisation methods and as regardless of the methods used most of our conclusions stand, we do not think the conclusions of this study are normalisation artefacts.

R1.2.4.

Why only show a PLA assay for ZNF462, whose binding is independent of RNAPolIII? Why not also show a PLA assay for a TF whose binding depends on PolIII?

We have tested several monoclonal antibodies specific to TFs and only the one shown in the manuscript worked by immunofluorescence in our cell line. We therefore could not provide additional PLA assays.

R1.2.5

As explained above (R1.2.1), considering the experimental design and the results from EV1H, one would expect that 10 to 20 kbps of nascent DNA will be devoid of nucleosomes genome-wide. This contradicts the authors' statement that only 300bp of DNA is nucleosome free behind the fork. The 10 to 20kb of newly replicated DNA labelled with EdU are not devoid of histones. There is relatively less histones compared to later time points. In

longer exposed immunoblots from EV1H, histones are detected, and by mass spectrometry we can estimate a drop in histones for the nascent time point, but they are still detected.

R1.2.6

"The systematic comparison of proteins bound to newly replicated chromatin and steady state chromatin revealed that many RNAPII dependent effects are specific to newly replicated chromatin." This may be true, but it still does not answer my question: why is RNAP2 enriched in the nascent and 1hr time points but then becomes depleted in the 2hr time point in the DRB treatment since global RNAPII levels don't change in the DRB treatment compared to DMSO (Figure 1C)?. Can the authors provide an explanation in the discussion beyond "that's what happens on replicated chromatin"?

In DRB, after 2hr of treatment, RNAPII level is stable on nascent and steady state chromatin compared to DMSO. After an additional 2hr of maturation (cells have now been treated 4hr total in DRB), RNAPII decreases on replicated and steady state chromatin. For both time points there is a good correlation between steady state and newly replicated chromatin. The referee may have been confused by "2hr DRB treatment" (nascent) and "+2h maturation" (4h DRB treatment in total).

Also, the results shown in Fig. 4G raise a new question: why is steady state chromatin after BRD treatment different from whole cell extract? Shouldn't they be the same if BRD does not interfere with replication? Otherwise, the observed differences between steady state and whole cell extract and the fact that DRB is added after and before EdU, respectively, in the steady state and whole cell extract samples, suggests that DRB might interfere with replication.

Whole cell extracts allow to measure the cellular abundance of the proteins identified. The difference between whole cell extracts and steady state chromatin reveals that upon DRB treatment there is an unloading of DNA repair proteins from steady state chromatin, but no change in the overall cellular abundance of these proteins.

R1.2.7, R1.2.8 and R1.2.9

"Moreover, in human, histone methylations are not expected to double within 2h, requiring instead 4h to 20h, depending on the histone residue (Alabert et al. 2015, Reveron et al. 2018). Histone modifications expected to decrease within 2h are H4K5K12ac and are behaving accordingly in this dataset (Fig. 2D). This dataset is therefore in good accordance with published proteomic analysis of histone modifications." While it is certainly reasonable to have certain expectation based on previously published data, I don't have to tell these authors that the iPOND experiment in this work is not exactly equivalent to the experiments in Alabert et al. and Reveron et al. In both of these works nascent chromatin was isolated after synchronization in early S-phase, so proteins were isolated from early replicating chromatin.

No, in Alabert et al 2015 HeLa S3 cells are in mid-S phase. Moreover, in Alabert et al. 2020, asynchronous mES cells are used. These references are included in the manuscript.

In this work nascent chromatin is isolated from asynchronous cells so all genomic loci should be represented (early and late replicating). It is therefore reasonable to assume that there might be some differences in the dynamics of histone PTM reestablishment between the two

sets of experiments, especially for H3K27me3 and H3K9me123, which tend to be in regions that replicate late.

Yes, it is reasonable to assume that there may be differences, given that it is also another cell line (TIG-3) and another technology (iPOND). Yet, we did not notice major differences.

This should be pointed out and later time points should be added (4, 6 or 8hrs?) at least for the DMSO treatment, although if they do the experiment I asked for in R1.1.5, they could add a later time point for the DRB and TPL treatments as well.

This is feasible in DMSO but not upon TPL/DRB treatment as prolonged treatment with these drugs lead to DNA damage (Skalska et al. 2021). Since the aim of this study is to monitor the effect of transcription on histone modification restoration, later time point using DMSO alone is beyond the scope of this paper.

Also, it is shown in Reveron et al., that H3K4me3 is restored in 50% of early replicating regions within 1 hr. Consequently, PTM recovery dynamics observed here are not quite in accordance with published results.

By ChOR-seq (Reveron et al.) the restoration of H3K4me3 can be reliably monitored. By mass spectrometry it cannot. Indeed, this modification is very low abundant (less than 1% of the peptide H3_aa3-8) and therefore the levels detected by mass spectrometry are within the noise of the experiments. Therefore, in Alabert et al. 2015 (NCC) and in this study (iPOND), we cannot comment on H3K4me3 levels. The data for this peptide, and for all peptides detected, are fully available in Dataset EV6 (Row 7) and available in Chorus.

Wouldn't it be reasonable to expect that H3K36me2/3 follows similar kinetics in the DMSO treatment since it is also a transcription dependent PTM? Instead, Figure 2E shows that H3K36me2 goes up 5 fold (??) from 5% to 25% within 2hrs and H3K36me3 only goes up a few percentages. H3K36me2/3 are surprisingly low in the N time point, suggesting that for some reason (maybe the click reaction is not very efficient on chromatin with recycled nucleosomes) iPOND enriches for newly synthesized histones and does not recover recycled histones at least in the N time point. This would explain why nascent chromatin appears depleted of nucleosomes in this time point, if new nucleosome assembly on nascent DNA happens sometime after old nucleosomes recycling. If iPOND does indeed enrich for new nucleosomes, this would also explain why H3K36me2 goes up 5 fold instead of the expected 2 fold or less. Clearly, it is difficult to explain all these discrepancies, when it is not known if iPOND recovers old and new histones in equal proportions.

In this setup, we cannot distinguish new and old histones, but we can follow the increase or decrease of histone modifications on replicated chromatin. Not having the distinction between new and old histones is not preventing to identify differences upon drug treatment. We have used SILAC coupled to iPOND to differentiate new and old histones, and we can see that similar to NCC, iPOND recovers almost 50/50 of new and old histones (see figure below). The slight overrepresentation of old histones likely reflects a contribution of recycled light amino acids during histone biosynthesis (Scharf et al. 2009; Xu et al. 2012; Zee et al. 2012).

Proportion of new and old histones on nascent chromatin detected by SILAC-iPOND. Cells were labelled as in Alabert et al. 2015 to distinguish the two populations. The dotted line indicates theoretical values. Two individual biological replicates are shown (labelled 1 and 2).

Unfortunately, even though I commend the authors for their necessary and valiant attempt at understanding how transcription influences chromatin maturation after replication, I cannot recommend the work for publication due to the lack of controls (listed above) that would have made their conclusions more convincing.

We hope that the revised version of the manuscript will please reviewer 1.

Cross-comments from referee 2:

R1.1.4 Although I agree with Reviewer #3 on Figure 1D that the t test is not appropriate based on the non-Gaussian distribution of EdU+ cells and should be replaced by a more appropriate statistical test, the two distributions do not show "major differences" in their EdU incorporation rates, suggesting that there is no global effect on DNA replication upon TPL or DRB treatment in the given timeframe.

The name of the statistical test used is provided in the revised version of the manuscript.

Please see also response to R1.1.4.

The FACS histograms in Fig. EV1 do also show similar distributions between DMSO, DRB and TPL treated nas. vs. 2h cells so I agree with the authors' conclusion that all presented data do not show a major global impact of the TRX inhibitors on DNA replication. The reviewer is right that this does not exclude the possibility that there might be local differences at individual genes/areas of the genome but the whole paper is based on bulk analysis of the proteome changes averaging individual differences from individual cells and loci so the reported changes reflect the global average of proteomic changes concurrent with the global EdU patterns.

R1.1.5 I agree that the requested quantification would be an valuable addition to the manuscript. We now provide further quantifications of the EdU intensities upon chase using density plots (See response to R1.1.5, and new EV1B-C).

R1.2.1 The reviewer makes an interesting point about the different enrichments of histones between the NCC and iPOND methodologies. I do believe that this discrepancy can be explained by the different methodological approaches where NCC is based on incorporating

biotin-dUTP and then direct streptavidin pulldown whereas iPond is based on an EdU Click-it reaction to conjugate biotin to nascent DNA. It is possible that the required Click-it reaction in iPond may shear the nascent DNA or affect the nucleosomal density. Nevertheless, I consider both NCC and iPond as complementary approaches to detect the proteomic makeup of nascent chromatin, both of which have been widely used in the field and gave important biological insights. Thus, I do think the authors should discuss this point in the manuscript but I do not think the reported discrepancy of both methods makes their results invalid. A section covering NCC and iPOND differences has been added in the discussion of the revised version of the manuscript, page 10.

For the internal spike-ins, the authors could attempt to normalize on other background proteins detected in all mass spec experiments for example on ribosomal proteins as these are highly abundant proteins usually not associated with replication forks and present in all MS datasets. In our experience, normalising to background proteins introduces more noise to the data, as these proteins also vary from sample to sample. Also, due to batch effects, some proteins vary dramatically between batches, so we cannot exclude that, for example, ribosomal proteins remain unchanged. We looked at ribosomal proteins (see figure below) and some of them exhibit batch effects (that is marked changes between batches), while others do not. Instead, we apply HarmonizR to mitigate batch effects, which adds a second layer of normalisation.

Abundance of two ribosomal proteins identified by iPOND-TMT. Values for each biological replicate and for each corresponding treatment is shown.

R1.2.4.

The PLA assay can be added but I do not see this as a crucial and important point for the manuscript. We have tested several monoclonal antibodies specific to TFs and only the one shown in the manuscript worked by immunofluorescence in our cell line. We therefore could not provide additional PLA assays.

R1.2.5

The differences may indeed be because of the different methodology and should be discussed in the manuscript as mentioned above. A section covering NCC and iPOND differences has been added in the discussion of the revised version of the manuscript, page 10.

R1.2.6

DRB treatment might block RNAPII on promoters and therefore affect fork progression in the given timeframes. Indeed a study by Fenstermaker et al., 2023 reported rapid re-association of RNAPII in the wake of the replisome on immature chromatin and rapid resumption of transcription. The authors could discuss their findings and the observed differences with DRB in light of this study.

We now highlight this point in the result section page 5: "Upon DRB treatment, although the level of phosphorylated RNAPII decreases on nascent chromatin (Fenstermaker et al., 2023), RNAPII abundance is only moderately affected (Fig. 1E, 1G)."

R1.2.7, R1.2.8 and R1.2.9

The reviewer raises again a good point and in the end it goes back to the potential differences in used methodologies between NCC and iPOND. I do think that the observed discrepancies need to be reported and discussed in the manuscript, but I do not agree that this would invalidate the author's data and findings. I do believe that elaborating on these issues in the discussion would be adequate and helpful to the readers.

Based on referees' suggestions, we have included additional controls and expanded the discussion and limitation section in the revised version of the manuscript.

Cross-comments from referee 1:

It appears to me that several of R3's concerns could be addressed by revising the text. The authors should consider revising the limitations section to address issues concerning EdU incorporation in TPL/DRB treatment (R1.1.4, and R1.1.5), along with performing the additional data analysis and control experiments suggested by the reviewer. Also, the PLA assay to test RNAPII and a TF's co-occupancy (R1.2.4.) should be doable within a reasonable time frame. The suggested peptide spike-in iPOND is essentially asking the authors to replicate the entire study with spike-in normalization, which demands time and resources, particularly if the authors don't end up finding major differences with/without spike-in.

In summary, the study does have limitations, some of which have been discussed, and more could be addressed with careful data analysis and control experiments and then discussed in the text. However, this should not unreasonably delay its publication. As R2 rightly commented, "the description of the proteome changes behind the fork upon global transcription inhibition is an important dataset and resource for the community". I'd stress upon being more rigorous in analyzing and discussing the limitations, which should address several of R3's concerns.

Based on referees' suggestions, we have included additional controls and expanded the discussion and limitation section in the revised version of the manuscript.

Dr. Constance Alabert
MCDB, University of Dundee
School of Life Sciences
Dow street
Dundee, Scotland DD15EH
United Kingdom

Dear Constance,

I am very pleased to accept your manuscript for publication in the next available issue of EMBO reports. Thank you for your contribution to our journal.

Best wishes,
Esther
